# Brain-wide dynamics linking sensation to action during decision-making

Andrei Khilkevich[1,2✉], Michael Lohse[1,2✉], Ryan Low[1], Ivana Orsolic[1], Tadej Bozic[1], Paige Windmill[1] & Thomas D. Mrsic-Flogel[1✉]

Perceptual decisions rely on learned associations between sensory evidence and appropriate actions, involving the filtering and integration of relevant inputs to prepare and execute timely responses[1,2]. Despite the distributed nature of task-relevant representations[3–10], it remains unclear how transformations between sensory input, evidence integration, motor planning and execution are orchestrated across brain areas and dimensions of neural activity. Here we addressed this question by recording brain-wide neural activity in mice learning to report changes in ambiguous visual input. After learning, evidence integration emerged across most brain areas in sparse neural populations that drive movement-preparatory activity. Visual responses evolved from transient activations in sensory areas to sustained representations in frontal-motor cortex, thalamus, basal ganglia, midbrain and cerebellum, enabling parallel evidence accumulation. In areas that accumulate evidence, shared population activity patterns encode visual evidence and movement preparation, distinct from movement-execution dynamics. Activity in movement-preparatory subspace is driven by neurons integrating evidence, which collapses at movement onset, allowing the integration process to reset. Across premotor regions, evidence-integration timescales were independent of intrinsic regional dynamics, and thus depended on task experience. In summary, learning aligns evidence accumulation to action preparation in activity dynamics across dozens of brain regions. This leads to highly distributed and parallelized sensorimotor transformations during decision-making. Our work unifies concepts from decision-making and motor control fields into a brain-wide framework for understanding how sensory evidence controls actions.

To link external events to beneficial actions, the brain must learn to transform relevant sensory input to drive the neural dynamics that underlie movement preparation and execution[1,11]. Where and how these transformations occur in the brain remain unclear.

When individuals make decisions based on ambiguous sensory information over time, the brain is thought to gradually accumulate the relevant input into an integrated neural representation that determines the upcoming choice[1]. Neural activity reflecting the integration of sensory evidence has been reported in several brain areas[1,8,12–22], most prominently in cortical areas such as frontal-premotor cortex[8,13,14,22] and posterior parietal cortex[15–18], and their immediate downstream targets such as the striatum[19–21]. However, recent studies have uncovered a broader encoding of sensory inputs, choice and actions throughout the brains of trained animals[3,5,9], raising questions about where sensory input is transformed into integrated task-relevant representations that guide action, and how widely distributed these representations are. It also remains unclear whether specific brain areas specialize in integration of sensory evidence owing to their inherent properties[8,23–26], or whether learning shapes the nature of this computation.

Here we address how integrated sensory evidence is converted to a choice and ultimately action. Action initiation is preceded by a build-up of preparatory activity that is observed in many brain areas[4,14,27–32] (also referred as choice-related activity), which in motor and premotor regions appears distinct from and orthogonal to the pattern of population activity that drives movement execution[4,33–35] (see ref. 36 for debate). Although evidence integration has been reported to modulate the preparatory activity of individual neurons in certain brain regions[14,18,37–41], the effect of evidence integration on the evolving neural dynamics surrounding movement[33,34], as well as the brain regions involved[42–44], remain to be understood on a brain-wide scale. In particular, it is unclear how segregated or parallelized the transformations between evidence integration, movement preparation and execution are across brain areas as well as across dimensions of neural activity.

To understand the brain-wide transformation of sensory input into choice and action, it is necessary to use tasks that can distinguish sensory and decision-related processes from action signals that dominate global brain activity[3,5,6,9]. Such tasks, pioneered in non-human primates[1,16,45–47], have recently been adapted for rodents[3,14,48–50], enabling greater access to interrogate the underlying circuit mechanisms

[1]Sainsbury Wellcome Centre for Neural Circuits and Behaviour, University College London, London, UK. [2]These authors contributed equally: Andrei Khilkevich, Michael Lohse. ✉e-mail: a.khilkevich@ucl.ac.uk; m.lohse@ucl.ac.uk; t.mrsic-flogel@ucl.ac.uk

as well as unbiased, brain-wide measurements with dense electrode recordings[3,5,51].

In this study, we describe how sensory evidence propagates and is transformed across the brain as mice engage in a task that requires temporal integration of visual input, designed to separate the influence of sensory evidence and movement on neural responses[14]. Our results reveal that ambiguous sensory input becomes integrated within widely distributed multi-regional premotor circuits in a learning-dependent manner, driving the preparatory phase of movement-related neural dynamics that eventually trigger the initiation of appropriate actions.

To study how relevant sensory input is transformed across the brain prior to a decision, we trained food-restricted, head-fixed mice on a visual change detection task designed to dissociate ongoing visual evidence observation from movement-related activity[14]. Mice were trained to be stationary on a running wheel while observing a drifting grating stimulus, whose speed fluctuated noisily every 50 ms around a geometric mean temporal frequency (TF) of 1 Hz ($\sigma = 0.25$ octaves), and to report a sustained increase in its speed by licking a reward spout (Fig. 1a). The mice were motivated to react promptly upon detecting a change by limiting the time in which the reward was accessible (Methods). Since changes in speed were often ambiguous, their timing unpredictable and the change in magnitude was randomized, mice had to continuously track the sensory stimulus for a prolonged duration (3–15.5 s) prior to the change. To ensure mice remained still during this time, any licking or movement on the running wheel prior to the stimulus change caused the trial to be aborted (Methods).

The detection performance of the mice improved with the size of the change in stimulus TF (Fig. 1b). At the same time, their reaction times were hundreds of milliseconds faster for large stimulus changes (Fig. 1b), similar to other reaction-time tasks requiring temporal integration[1]. Furthermore, the average stimulus speed preceding 'early licks' (Fig. 1c), which occasionally occur during the baseline stimulus prior to change, was increased during approximately 0.3 to 1 s before early lick (Fig. 1c). This suggests that at least some early licks are triggered by fluctuations in the baseline stimulus and that sensory information influences the mouse's judgments on the timescale of hundreds of milliseconds.

Thus, by encouraging mice to continuously monitor ambiguous sensory evidence while controlling for their movement, this task enables us to examine how the brain processes sensory evidence and transforms it into action commands.

## Brain-wide encoding of sensory input

To understand how the brain of trained mice transforms visual stimulus speed into goal-directed licking in this task, we performed dense silicon electrode recordings (Neuropixels probes[51]) from 15,406 units spanning 51 brain regions (that is, 12,772 units from regions with more than 40 manually curated, good and stable units; Extended Data Fig. 1, Supplementary Table 1 and Methods) distributed across the cortex, basal ganglia, hippocampus, thalamus, midbrain, cerebellum and hindbrain (Fig. 1d–f, 15 mice, 114 recording sessions, 167 probe insertions and 50,997 trials), while capturing high-speed videos of the face and pupil as well as movements of the running wheel (Fig. 1f).

To identify which neurons encode visual evidence (stimulus TF), lick preparation and lick execution, we utilized single-cell Poisson generalized linear models (GLMs) that fit trial-to-trial neural activity from task-related events, stimuli and behaviour (Fig. 1g and Extended Data Fig. 2). By using a cross-validated nested test (that is, holding out a predictor of interest to assess its contribution to neural activity), we identified the neurons that significantly encode different variables of interest while accounting for variance captured by other predictors (Methods).

In agreement with the prevalence of motor-related signals in the brain[3–6,9], lick execution was encoded globally with the activity of at least 50% of neurons recorded encoding this action (Fig. 1k,l and Extended

Data Fig. 3a). Using videography to establish the onset of lick execution, we also identified a smaller, yet substantial fraction of neurons encoding lick preparatory activity (that is, modulation of activity within 1.25 s leading up to a lick), also distributed globally (Fig. 1h,k,l). A sparser fraction of neurons encoded subtle fluctuations in stimulus TF during the baseline period on trials devoid of mouse movements (5–45%; referred to as TF-responsive units; Methods). These neurons were distributed across the majority of brain areas. Although the largest contingent of TF-responsive units were found in the visual system (visual cortex, visual thalamus and superficial superior colliculus), significant fractions (5–25%) were also observed in most areas outside the visual system, including regions of the frontal cortex (secondary motor cortex (MOs), anterior cingulate cortex (ACA), medial prefrontal cortex (mPFC), frontal pole (FRP), orbitofrontal cortex (ORB) and primary motor cortex (MOp)), basal ganglia (striatum (caudoputamen; CP), globus pallidus external segment (GPe) and sibstantia nigra reticular part (SNr)), hippocampus (dentate gyrus (DG), CA1, CA3 and subiculum (SUB)), midbrain (midbrain reticular nucleus (MRN), anterior pretectal nucleus (APN), multimodal and motor superior colliculus (SCm) and nucleus of the posterior commisure (NPC)) and cerebellum (lobules 4/5 (Lob4/5), simplex lobule (SIM), central lobule 3 (CENT3), CRUS1/2 and deep cerebellar nuclei (DCN)). Of note, these multi-regional responses to visual input could not be explained by other variables that might correlate with fluctuations in stimulus TF because fast or slow TF pulses did not trigger consistent movements of the face or running wheel (Fig. 1i), there was an absence of TF-responsive cells in the medulla and orofacial motor/premotor nuclei whose activity reflects movements of the mouth and tongue (Fig. 1h,k,l), and the GLM was unable to predict responses to TF fluctuations without the stimulus TF as predictor (Extended Data Fig. 2f,g).

Together, these results show that sensory evidence representations are surprisingly widespread, with a sparse subpopulation of neurons tracking behaviourally subthreshold fluctuations of relevant sensory input in almost all brain areas, but excluding the nuclei controlling orofacial movements which become engaged when mice report their decision. These sparse, distributed representations of visual evidence ultimately give rise to the initiation of movement which itself recruits activity in more than half of neurons across the brain.

## Timescales of sensory responses across the brain

To determine how sensory evidence propagates in activity across the brain, we quantified neural responses to momentary samples of stimulus TF during baseline period when mice did not lick or move. We aligned neural responses to fast TF pulses (50 ms stimulus samples 1× s.d. above baseline TF of 1 Hz; Fig. 2a–c and Methods), and quantified their peak time (Fig. 2d) and duration (full width at half peak value; Fig. 2f), which closely matched those estimated by the GLM (Fig. 2e,g and Extended Data Fig. 5a–d). As expected, brain regions in early visual system (dorsal lateral geniculate complex (LGd), primary visual cortex (VISp) and superficial superior colliculus (SCs)) responded earliest to fast TF pulses with brief responses that faithfully tracked the stimulus TF (Fig. 2b,d–i). By contrast, brain regions outside the visual system containing TF-responsive units responded significantly more delayed to fast TF pulses (Fig. 2b–e,h) and exhibited more prolonged responses than neurons in visual areas (Fig. 2b,c,f,g,i and Extended Data Fig. 4). Specifically, neurons in frontal motor cortex, basal ganglia, cerebellum and some regions of the midbrain and thalamus maintained the representation of sensory evidence for several hundred milliseconds beyond the duration of the stimulus sample that triggered the response (Fig. 2b,c,f).

## Parallel sensory integration in premotor areas

The longer timescales of neural responses to fast TF pulses outside the visual system suggests that these areas can integrate multiple

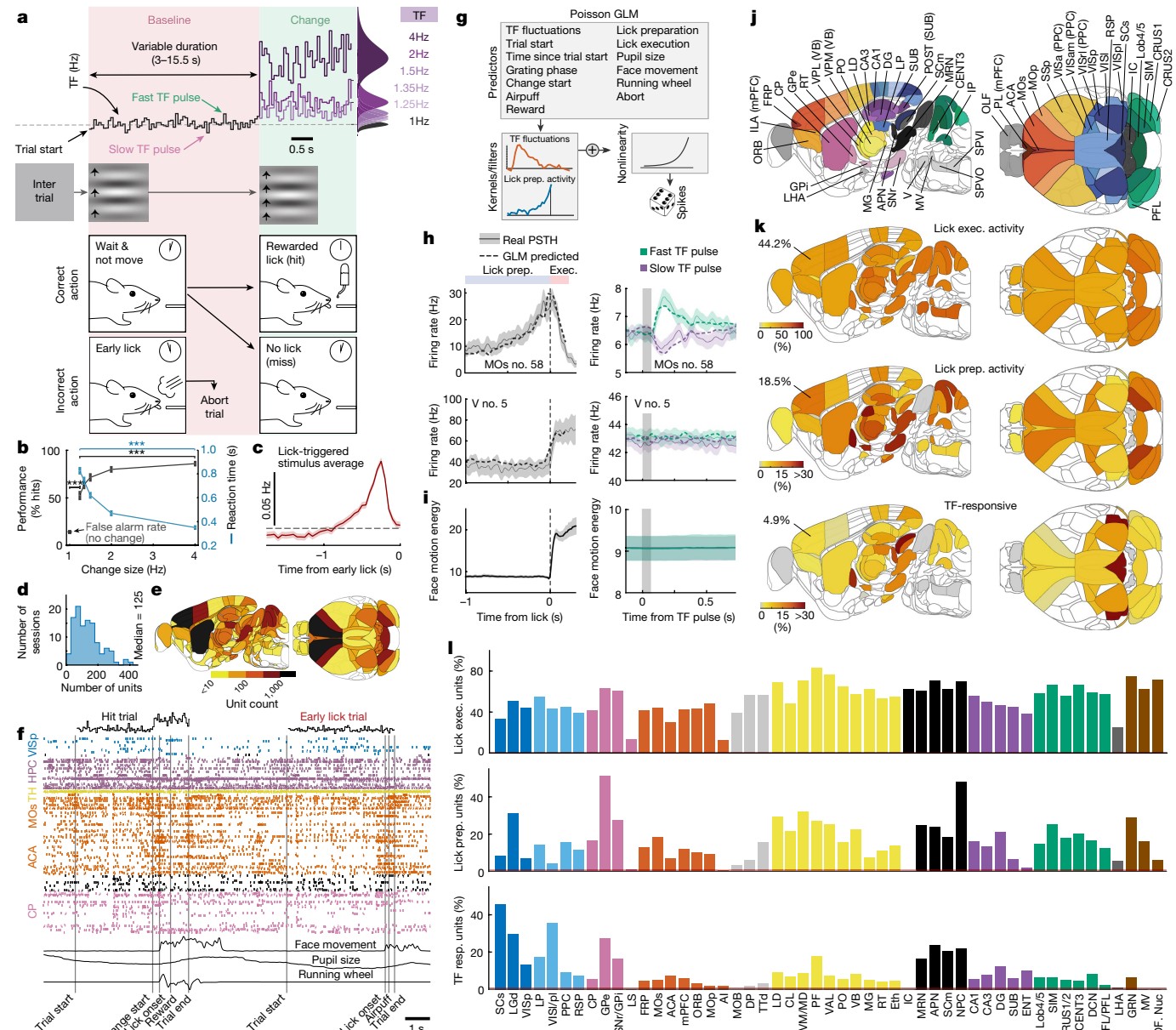

**Fig. 1 | Widespread representation of sensory evidence, lick preparation and lick execution across the mouse brain during noisy visual change detection. a**, Schematic of the visual change detection task for head-fixed mice. **b**, Psychometric and reaction-time curves (mean and 95% confidence interval; two-sided Student's *t*-test; *n* = 114 sessions, 15 mice). **c**, Mean stimulus TF (with 95% confidence interval) preceding early licks during the baseline period. Dashed lines indicate linear mean (1.016 Hz) of baseline stimulus TF. **d**, Number of units recorded per recording session. **e**, Brain map of number of units recorded per area across all recording sessions of trained mice. **f**, Example time series across two trials (a rewarded trial and an early lick trial) of stimulus TF, spike times across simultaneously recorded neurons (two probes), face motion energy (from videography), pupil size and running wheel movement. HPC, hippocampus; TH, thalamus. **g**, Schematic of single-trial Poisson GLM. Prep., preparation. **h**, Mean firing rate around early licks (left), and mean response to fast and slow TF pulses during baseline period (right) for an

example neuron in MOs and trigeminal motor nucleus (V), together with GLM predicted (on 10% held-out data) mean activity (dashed lines, with 95% confidence interval). Exec., execution; PSTH, peristimulus time histogram. **i**, Mean (with 95% confidence interval) face motion energy (from videography (Methods)) around early licks, and around fast and slow TF pulses. **j**, Brain maps with labelled brain regions. See Supplementary Table 2 for definitions of abbreviations. **k**, Brain maps of percentage of units encoding lick execution (top row), lick preparation (middle row) and stimulus TF fluctuations during the baseline period in the absence of movement (bottom row). **l**, Percentage of units encoding lick execution, lick preparation and stimulus TF fluctuations during baseline across all brain regions with more than 40 units recorded. Resp., response. See Supplementary Table 1 for number of units recorded in each brain area and Supplementary Table 2 for definitions of brain region abbreviations. *P < 0.05, **P < 0.01, ***P < 0.001.

samples of behaviourally relevant visual input. Indeed, previous modelling of mouse behaviour in this task shows that mice are guided by TF fluctuations unfolding over several hundred milliseconds[14]. Although this suggests that mice use temporal integration of stimulus TF to detect changes, they may also respond to outliers in stimulus to guide their lick responses. To disambiguate between these

behavioural strategies (integration versus outlier detection), we applied a combination of analytical and modelling approaches to mouse behaviour to show that mice indeed do use integration of evidence over a timescale of around 0.25 s. First, the decay time (τ) of the early lick-triggered stimulus average (psychophysical kernel; see ref. 52) is 0.27 s, a time course significantly longer than predicted

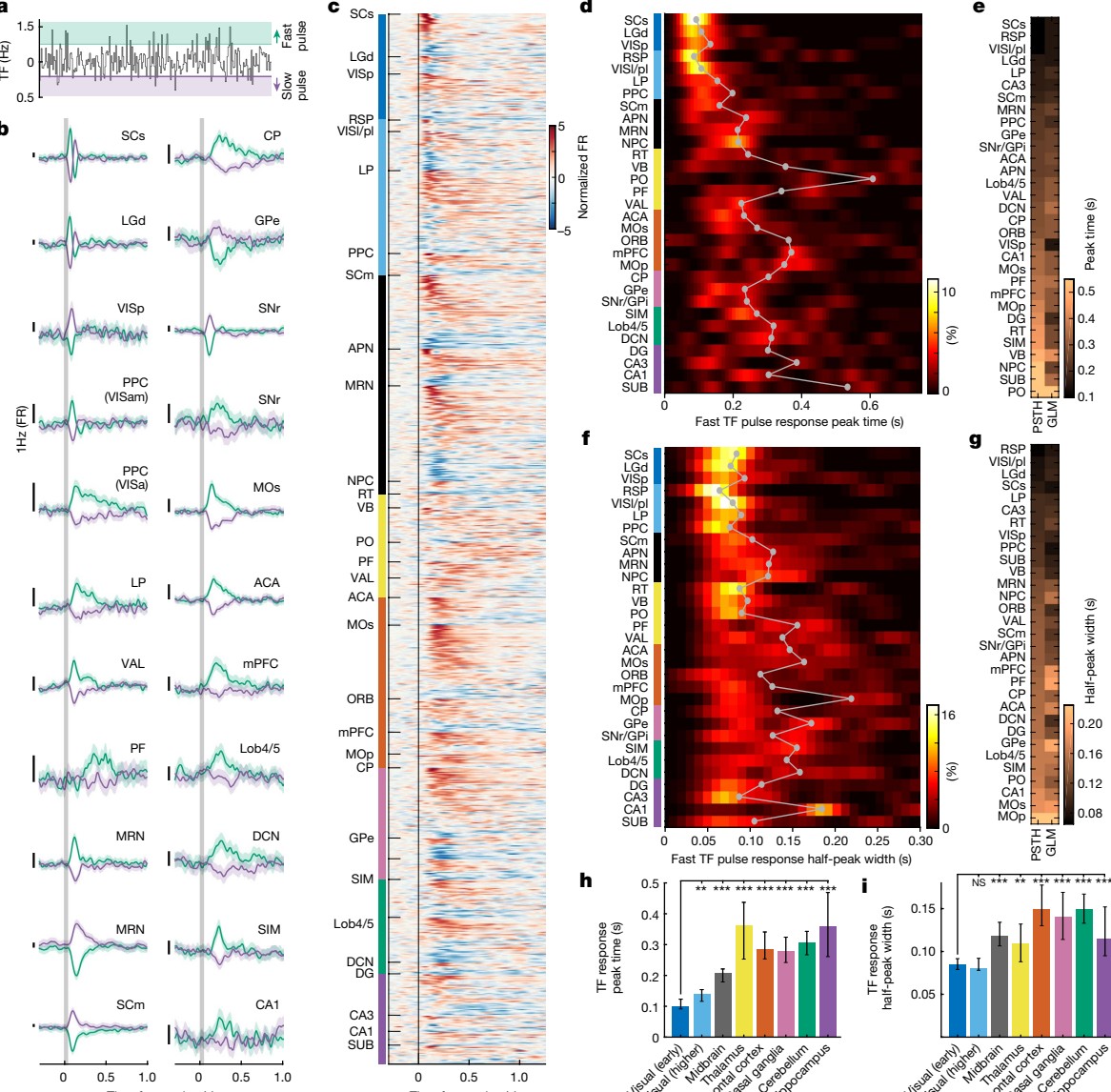

**Fig. 2 | Propagation and widening of fast TF pulse responses across the brain.**
**a**, Schematic of identification of fast (TF pulse > 1 s.d.) and slow (TF pulse < −1 s.d.) TF pulses fluctuating around the mean baseline stimulus TF. **b**, Single-neuron examples of fast and slow TF pulse responses from selected areas across the brain (mean with 95% confidence interval). FR, firing rate. **c**, Fast TF pulse responses of all TF-responsive neurons in all brain areas with ten or more TF-responsive units. **d**, Distribution of response peak times estimated from fast TF pulse responses for each brain area with ten or more TF-responsive units (grey line and circles indicate median peak time per area). **e**, Comparison of median peak times estimated from fast TF pulse responses (left column) and GLM weights tracking TF fluctuations (GLM TF kernels; see Extended Data Fig. 2 for example kernels; Methods) for each area (right column). **f**, Distribution

of fast TF pulse response half-peak widths (estimated from fast TF pulse responses) for each area with ten or more TF-responsive units (grey line and circles indicate median peak time per area). **g**, Median fast TF pulse response half-peak widths compared with half-peak widths of the GLM TF kernel. **h**, Fast TF pulse response peak times across major brain area groupings (median and 95% confidence interval; brain areas in each group are listed in Supplementary Table 1). **i**, Fast TF pulse response half-peak widths across major brain area groupings (median and 95% confidence interval). Wilcoxon rank sum test. Values of $n$ for each brain area grouping are presented in Supplementary Table 1 and definitions of brain area abbreviations can be found in Supplementary Table 2. NS, not significant.

by an artificial agent relying solely on an outlier detection strategy (Fig 3a,b and Methods). Second, mice are more likely to lick when two fast pulses occur within 0.25 s of each other than would be predicted by the joint independent effect of two fast pulses (Fig 3d and Extended Data Fig. 6e–i). Moreover, the independent effect of two fast pulses fully explained the data of the outlier-detection agent (Extended Data Fig. 6h,i). Finally, a simple leaky-integrator model with a 0.25 s decay time ($\tau$) better predicts early lick times and single-trial hit reaction times than when this model is not allowed to integrate evidence (Extended Data Fig. 7b–h).

Given that lick responses depend on integrating the stimulus TF over several hundred milliseconds, we next determined the neural correlates of this integration process. We reasoned that a prolonged response to a fast TF pulse serves as a neural substrate for temporal integration of multiple fast TF pulses, by allowing responses to successive fast TF pulses to build on each other. By finding instances during the baseline period when two fast TF pulses occurred at a given delay from each other (Fig. 3e and Methods), we calculated the average response across all TF-responsive units in a brain region to those pulses, and measured the amount of response facilitation to the second fast pulse relative

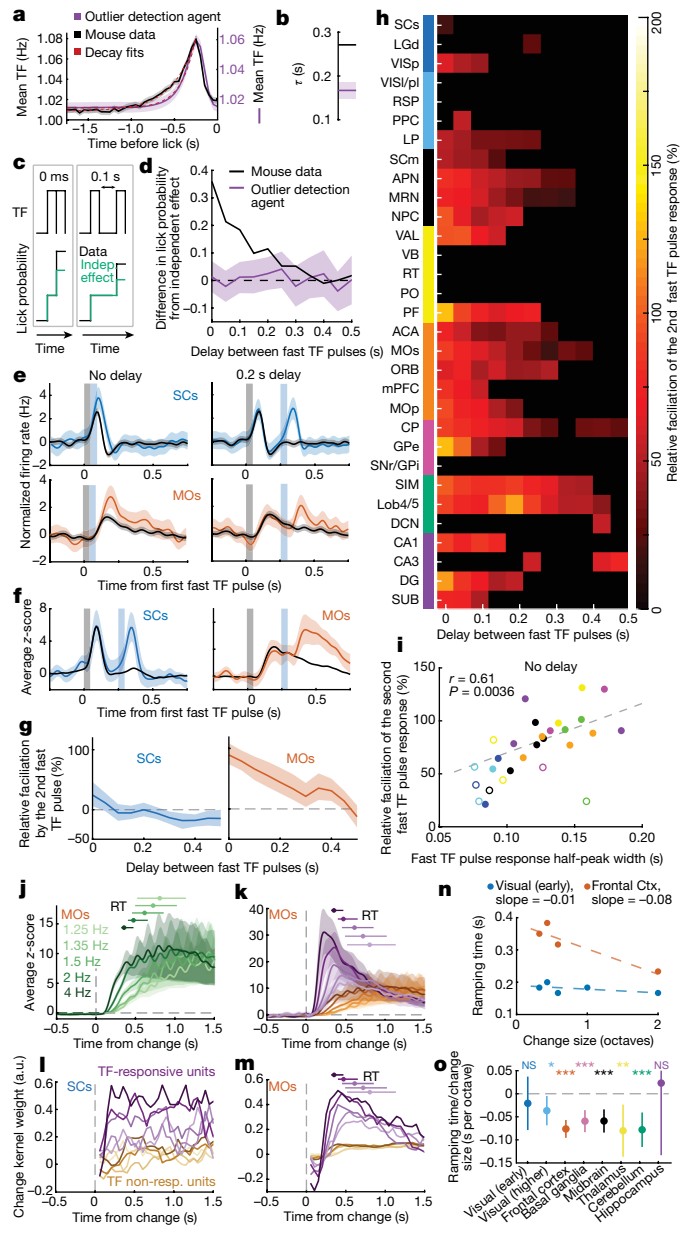

**Fig. 3 | Accumulation of visual evidence as a behavioural strategy and its neural implementation across the brain. a**, Mean stimulus TF preceding early licks in mouse data and outlier-detection agent. Red dashed lines show exponential decay fits. **b**, Decay time of the exponential fits in **a**. **c**, Schematic showing how lick probability is affected by two fast TF pulses that either integrate temporally (black) or act independently (indep.; green). **d**, Difference between observed early lick probability after two sequential fast TF pulses and the one predicted from their independent effect (Extended Data Fig. 6e–g), normalized by the probability from independent effect, shown as a function of delay between pulses. Data are mean with 95% confidence intervals. **e**, Responses to a single fast TF pulse (black) or a sequence of two fast pulses separated by 0 s (left) or 0.2 s (right) in example neurons from SCs and MOs. **f**, Average response to a sequence of two fast TF pulses separated by 0.2 s delay from all TF-responsive neurons in SCs (left) and MOs (right). **g**, Facilitation of response to the second fast TF pulse as a function of delay between two pulses for TF-responsive units in SCs and MOs. **h**, Same as **g**, but for all brain regions with at least ten TF-responsive units. Only time points with 95% confidence interval above zero (bootstrap test) are shown. **i**, Pearson correlation between second fast TF pulse facilitation and the median half-peak width of response to fast TF pulse across brain regions (*P* value based on *t*-statistic). Correlation excludes brain regions without significant facilitation, shown as open circles. **j**, Average activity of MOs units aligned to TF change onset on hit trials, split by change magnitude. Reaction times (RTs) per magnitude are shown as median (dots) with ranges between 25th and 75th percentiles. **k**, Same as **j**, but with the MOs population split into TF-responsive (shades of purple) and TF non-responsive (shades of orange) units. Darker colours correspond to larger change magnitudes. **l,m**, Mean GLM weights tracking activity after change (change kernels) from SCs (**l**) and MOs (**m**) units, derived from activity during change periods. Kernels shown for TF-responsive and non-responsive units, across different change magnitudes. Colour coding as in **k**. Reaction times shown as in **j**. a.u., arbitrary units. **n**, Each dot is the time to 50% of the peak value (ramping time) of the average change kernel across TF-responsive units in early visual areas and frontal cortex (Ctx), shown per change magnitude. **o**, Scaling of ramping time in activity with change size: each point represents a slope (in seconds per octave) of the linear fit to the dependence shown in **m**, for each group of brain regions. Bootstrap test. Values of *n* for each brain region and brain region group are presented in Supplementary Table 1 and definitions of brain area abbreviations can be found in Supplementary Table 2. In all panels, shaded regions or error bars indicate 95% confidence intervals.

to the first fast pulse response (Fig. 3f,g, and Fig. 3e for single-neuron examples; Methods). The majority of early and higher-order visual areas did not show facilitated responses to the second fast pulse even at a 0.1 s interval between the pulses (Fig. 3g,h and Extended Data Fig. 5e), whereas thalamic lateral posterior nucleus (LP) and hippocampal regions showed facilitation of up to 0.2–0.3 s inter-pulse delay. Across non-visual thalamus, facilitation was observed only in ventral anterior-lateral complex (VAL) and parafascicular nucleus (PF), the key nodes in cortico-cerebellar and cortico-basal-ganglia loops, respectively[29,53,54]. Most regions in frontal cortex, basal ganglia, cerebellum and midbrain exhibited significant facilitation around 0.2–0.4 s from the first fast pulse (Fig. 3g,h), resembling the behavioural integration timescales (Fig. 3b,d). The amount of relative facilitation to the second fast TF pulse correlated with response duration to a single fast TF pulse across brain regions (Fig. 3i), highlighting that one is a prerequisite for the other.

Thus far, we had isolated the sensory evidence representations by studying them in the absence of movement (that is, baseline period of the trial). Typically, however, neural representations of sensory integration are studied by examining neural responses during presentation

of stimuli that trigger the learned response, when there is an overlap of multiple correlated signals related to sensory integration, movement preparation and execution[1,37,55]. There, evidence integration is inferred by the ramping of neural responses that scale with stimulus strength[37]. Similarly, we found that in regions that integrate pulses of sensory evidence during the baseline period (Fig. 3e–h), such as MOs, the slopes of ramping activity in the change period scaled with the magnitude of the TF change (Fig. 3j). Notably, the TF-responsive subpopulation responded more strongly than the rest (Fig. 3k), with its ramping activity starting and peaking considerably earlier.

To account for the influence of the mouse's movement on these response profiles, we used the visual response components of the GLM fitted separately to neural responses for each change magnitude (Fig. 3l,m and Methods). In most areas outside of the visual system and hippocampus, the visual response components of TF-responsive neurons showed ramp-like activity that steepened with increasing change magnitudes, suggesting that these neural populations implement temporal integration of sensory evidence as mice report the change (Fig. 3l–o, Extended Data Fig. 7n,o and Methods). Moreover, for comparison, early visual areas, such as SCs, exhibited step-like, sustained responses to different change magnitudes (Fig. 3l–o and Extended Data Fig. 7n,o), thus signalling the change in stimulus TF, but without integration. This is consistent with the early visual system faithfully tracking the fluctuations in sensory input, whereas downstream structures have the capacity to integrate the stimulus stream, essentially denoising it, thus making sensory change detection easier (Extended Data Fig. 7k–m).

These results reveal that temporal integration of sensory evidence is a parallel, distributed, multi-regional computation—implemented by transforming transient responses to sensory input in visual areas into prolonged representations of integrated sensory evidence in frontal cortex, basal ganglia, cerebellum, thalamus and midbrain structures—which does not propagate to motor execution nuclei in the medulla.

## Learning enables widespread sensory integration

We next tested whether the encoding of sensory evidence outside the visual system is intrinsic to the brain regions themselves or a result of learning the relevant stimulus–reward associations. We recorded neural activity in untrained mice (6,215 units, 45 sessions, 6 mice) that had been exposed to the same stimuli but given random rewards (Fig. 4a,b and Methods), thus never associated changes in stimulus TF with reward. As expected, we found significant fractions of neurons encoding fluctuations in stimulus TF in the visual system (SCs, LGd, LP and VISp) and parts of the midbrain (APN and SCm) in untrained mice. However, we did not find cells with prominent TF responses in frontal-motor cortex, cerebellum, striatum or MRN—regions that in trained mice respond to TF fluctuations (Fig. 4c–e and Extended Data Fig. 8a). This demonstrates that encoding of sensory evidence in regions outside the visual system—where the sensory evidence is integrated—to a large degree, emerges with learning.

To test whether the integrative properties of neurons in non-visual areas are shaped by learning, we assessed whether stimulus integration can be predicted from intrinsic timescales of neural firing of each area. Intrinsic timescales of activity in cortical areas in non-human primates and rodents, defined as the time constant of autocorrelation function of each neuron's activity, have been suggested to determine duration of task-relevant responses[8,23]. However, we did not find intrinsic timescales of neural activity (measured in the inter-trial periods devoid of visual stimuli and movement) to correlate with the duration of fast TF pulse responses across different brain regions (Fig. 4f,g) or in individual neurons (Extended Data Fig. 8b–e). Notably, the intrinsic timescales of individual brain regions were similar in trained and untrained mice, indicating that they are an intrinsic property of each area that is unaffected by learning (Fig. 4h,i). Together, these results imply that representation and integration of sensory evidence emerge with learning in most association and premotor areas outside of the visual system.

## Evidence-encoding cells initiate preparatory activity

We next explored how the integrated sensory evidence is transformed into preparation of an action that reports the decision. Preparatory activity before action initiation has been observed in multiple brain areas during motor planning and in decision-making tasks[4,15,28,30], including our task (Figs. 1j and 5a). Given that neurons downstream of the visual system encode both sensory evidence and lick preparation (Fig. 1j and Extended Data Fig. 3d), we tested whether evidence integration and preparatory activity engage similar patterns of activity in these brain regions. We computed the alignment of population vectors between responses to a single fast TF pulse (Fig. 5a,b, left) and preparatory activity before the early lick onset (Fig. 5a,b, right) of TF-responsive subpopulations in different brain regions. In MOs (Fig. 5c) and other areas outside of the visual system capable of integrating sensory evidence—including frontal cortex, cerebellum, midbrain and basal ganglia—these population vectors were significantly aligned (Fig. 5d), whereby neurons that increase their firing to fast TF pulses also increase their activity prior to lick initiation, and vice versa (Fig. 5c). By contrast, no such relationship was observed in areas that do not integrate sensory evidence (Fig. 5d), such as SCs (Fig. 5c). These results imply a widespread coupling between integration of sensory evidence and movement preparation, as previously observed in monkey lateral

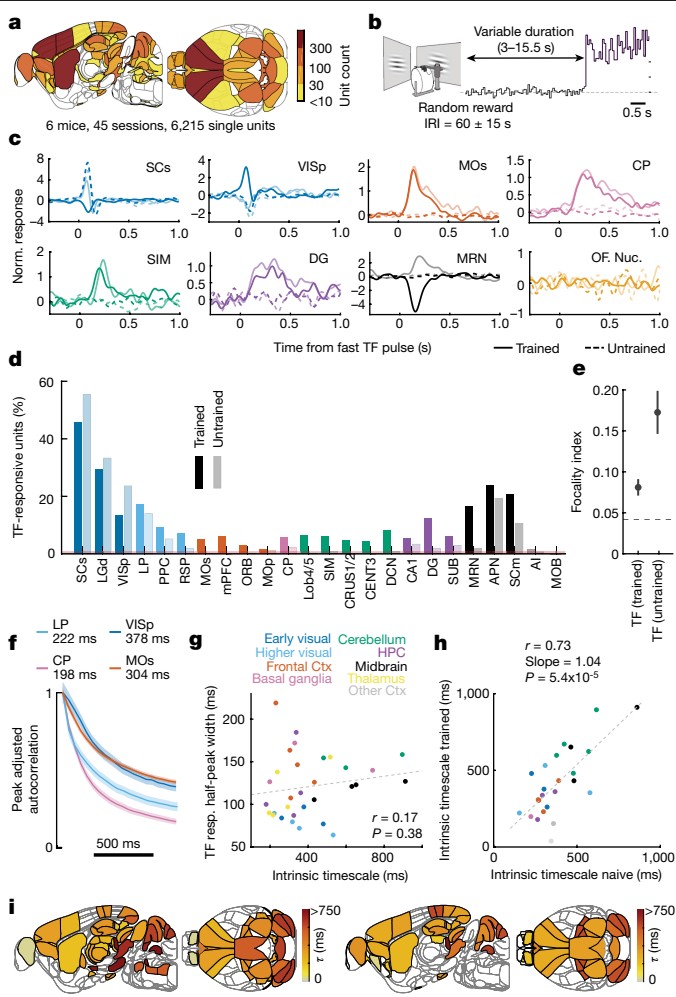

**Fig. 4 | Representation and integration of visual evidence in association and premotor areas emerges with learning. a**, Schematic of stimulus presentation with random reward delivery used for recordings in untrained mice (Methods). **b**, Brain maps of unit counts recorded from untrained mice. IRI, inter-reward interval. **c**, Examples of top two (lowest *P* value) fast TF-responsive neurons in trained mice (solid lines) or untrained mice (dashed lines) in SCs, VISp, MOs, CP, SIM, DG, MRN and in the orofacial motor nucleus. Norm., normalized. **d**, Percentage TF-responsive units in all brain areas with more than 40 neurons recorded in both trained and untrained mice. **e**, Focality index of distribution of TF-responsive units across areas with more than 40 neurons recorded in both untrained and trained mice. In untrained mice, TF-responsive units were confined to a much more limited set of brain regions, compared to trained mice, leading to a significantly higher focality index (*n* = 24 overlapping brain regions; *P* < 0.001, bootstrap test (Methods)). Error bars show 95% confidence intervals (Methods). **f**, Examples of autocorrelation functions from which intrinsic timescales are estimated (that is, τ of decay of autocorrelation function). Error bars are 95% bootstrapped confidence intervals. **g**, Pearson correlation (*P* value based on *t*-statistic) between intrinsic timescales and median half-peak width of responses to a fast TF pulse for all TF-responsive neurons across the brain of trained mice. **h**, Pearson correlation (*P* value based on *t*-statistic) between intrinsic timescales in untrained mice and trained mice. **i**, Brain maps of intrinsic timescales of trained mice (left) and untrained mice (right). See Supplementary Table 2 for definitions of brain region abbreviations.

intraparietal area (LIP) and frontal cortex[22,37], but which we find to be far more widespread across sparse subpopulations of frontal cortex, basal ganglia, cerebellum, thalamus and midbrain.

If accumulation of evidence contributes to the build-up of preparatory activity, we would expect the neural subpopulations that integrate

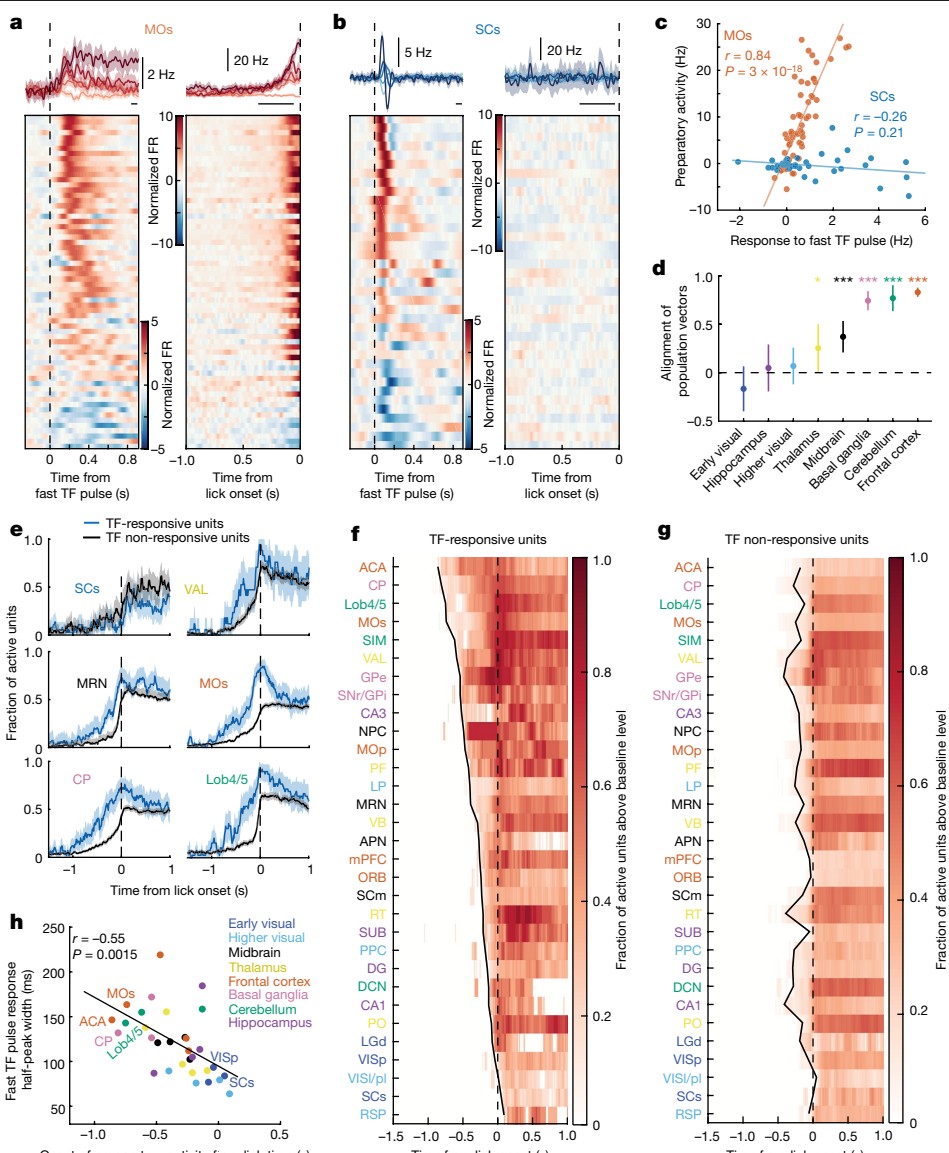

**Fig. 5 | Preparatory activity is led by TF-responsive subpopulations. a**, Left, mean responses to a fast TF pulse of five example TF-responsive units in MOs (top) and responses to a fast TF pulse for all TF-responsive units in MOs (z-scored firing rate) (bottom). Right, activity of the same neurons aligned to early lick onset. **b**, Same as **a**, but for TF-responsive units in SCs. Horizontal black lines indicate windows of activity used to calculate the alignment of population vectors in **c**. **c**, Alignment (Pearson correlation; P value based on t-statistic) between responses (baseline subtracted) of TF-responsive MOs or SCs units to a fast TF pulse and their preparatory activity before the early lick. **d**, Mean alignment of population vectors (correlation in **c**) for each group of brain regions (bootstrap test). See Supplementary Table 1 for n of each brain region group. **e**, Fraction of significantly active units (P < 0.01, z-test) as a function of

time, shown separately for TF-responsive and TF non-responsive units for six example brain regions. Values of n for each brain region are presented in Supplementary Table 1. **f**, Fraction of active TF-responsive units (thresholded by lower 95% confidence interval greater than zero, bootstrap test) as a function of time from the hit-lick onset, shown for each brain region. Brain regions are sorted according to the time of the earliest, significantly active fraction (black line; Methods). **g**, Same as **f**, but for the TF non-responsive subpopulation. **h**, Relationship between the onset of preparatory activity in TF-responsive units and their median response duration to a fast TF pulse across brain regions. Pearson correlation and corresponding P value from t-statistic are shown on top. In all panels, shaded regions and error bars indicate 95% confidence interval. See Supplementary Table 2 for definitions of brain region abbreviations.

evidence to be recruited first prior to a decision to a lick, and that brain regions with longer timescales of integration would have an earlier onset of preparatory activity. Indeed, prior to hit-lick onset during the change period, the TF-responsive populations were recruited significantly earlier than the TF non-responsive populations in areas integrating sensory evidence, including the frontal cortex, basal ganglia, cerebellum and midbrain (Fig. 5e–g, Extended Data Fig. 9b,c and Methods). The earliest differences in activation were observed across several brain subdivisions, including ACA, MOs, striatum (CP) and Lob4/5 (Extended Data Fig. 9b,c). Moreover, the onset of preparatory activity of the TF-responsive subpopulation scaled with the duration of

response to a fast TF pulse (Fig. 5h and Extended Data Fig. 9d), revealing that the longer timescales of integration lead to an earlier onset of preparatory activity. Together, these results demonstrate that accumulation of evidence contributes to the build-up of preparatory activity in multiple brain regions downstream of the visual system.

## Brain-wide orthogonal dynamics surrounding action

Previous studies have found that population activity in motor cortex transitions between orthogonal sets of dimensions (subspaces) before and after movement onset[33,34]. Following movement onset,

activity occupies a 'movement' subspace, in which projections of activity closely resemble the muscle activity during movement execution. Prior to movement onset, the patterns of activity are different and confined to an orthogonal subspace ('movement-null'), wherein activity builds up or persists, but does not drive the movement itself. To understand the neural dynamics during the transition between movement preparation and execution in our task, we applied the same analysis framework to each brain region population activity on hit-lick trials, by decomposing population activity into projections onto movement and movement-null dimensions (Methods). We defined the movement dimensions as those that captured the best similarity with the activity of orofacial motor and premotor nuclei that drive licking[56,57] (Extended Data Fig. 10b,c), and a set of movement-null dimensions orthogonal to them, wherein activity can reside without directly affecting licking.

We first tested whether the preparatory activity occupies a movement subspace or is orthogonal to it, as previously demonstrated in primary and premotor cortex[33–35] (Fig. 6a, orthogonal modes hypothesis). Figure 6b–d shows MOs activity aligned to hit-lick onset and projected onto the first movement and movement-null dimensions (see also Extended Data Fig. 10b–d). Relative occupancy of these subspaces around lick onset (Fig. 6e,f and Methods) revealed that pre-lick activity in MOs predominantly resided within the movement-null subspace (Fig. 6e, and was largely one-dimensional (Extended Data Fig. 10c)), and then transitioned into the movement subspace after the lick onset. Of note, preparatory activity was confined to the movement-null subspace across all other brain regions (Fig. 6f and Extended Data Fig. 11a,b).

Shortly following lick onset, population activity transitioned from movement-null into the movement subspace, almost concurrently throughout the brain. This state transition could result only from an increase in activity within movement subspace (Extended Data Fig. 11a) or also from a decrease in activity within the moment-null subspace following lick onset. Consistent with the latter, activity within movement-null subspace peaked and then sharply decreased immediately after the lick onset in most brain regions that had preparatory activity (Fig. 6f, green line, and Extended Data Fig. 11b, c).

Together, these results reveal that the abrupt transitions in neural dynamics between orthogonal movement-null and movement subspaces at movement onset is a general computational feature observed in most association and premotor brain areas.

## Linking evidence integration and motor dynamics

If accumulation of visual evidence drives preparatory activity, which resides in movement-null subspace, one would expect TF-responsive units to have a disproportionate contribution to activity in movement-null subspace. To test this, we decomposed projections onto movement and movement-null dimensions into a sum of contributions from TF-responsive units and the rest of the population (see Methods). For example, in MOs, we observed a disproportional contribution from TF-responsive subpopulation to the preparatory activity within the movement-null subspace (Fig. 6c,g). Applying this analysis across all brain regions, we found that the TF-responsive subpopulation contributed disproportionately to the preparatory activity in a more restricted subset of areas (Fig. 6h and Extended Data Fig. 11d,e): frontal cortex (ACA, MOs, MOp, ORB and mPFC), cerebellum (Lob4/5, SIM and DCN), basal ganglia (CP, SNr/globus pallidus internal segment (GPi) and GPe), as well as some regions of the midbrain (MRN, NPC and SCm) and thalamus (VAL and ventrobasal complex (VB)). Notably, these predominantly premotor areas integrated evidence over longer timescales (Extended Data Fig. 11f; see also Fig. 5h), emphasizing the link between evidence accumulation and preparatory activity.

Sensory evidence should no longer be informative of choice once the animal has committed to its decision. Accordingly, the contribution of TF-responsive units to preparatory activity in movement-null subspace collapsed to chance level after lick onset in most premotor areas in which TF-responsive units disproportionately drove preparatory activity (Fig. 6h; see Extended Data Fig. 11g for a comparable analysis in movement subspace). This collapse is consistent with the cessation of evidence accumulation despite the continuous presence of the change stimulus (see also Fig. 3j–l).

Consistent with the observations that preparatory activity and responses to pulses of sensory evidence are aligned within TF-responsive population of neurons (Fig. 5c,d) and that the preparatory activity of the entire population is confined to the movement-null subspace (Fig. 6f), we found that a response to TF pulse is aligned with the dimension that captures the most variance of the preparatory activity (first movement-null dimension) in most regions beyond the early visual system (Fig. 6i,j, top, k and Extended Data Fig. 12a). By contrast, responses to fast TF pulses were not positively aligned with the first movement dimension in any brain region group (Fig. 6i, j, bottom, k). Consequently, outside of the early visual system, we find that the integration of sequential pulses of evidence primarily takes place along the first movement-null dimension (Fig. 6k–m and Extended Data Fig. 12b). This provides an explanation for how sensory evidence can recruit activity across the majority of brain regions without directly driving the movement.

## Discussion

Here we describe the brain-wide neural implementation of evidence integration, movement preparation and execution—the key processes underpinning decision-making—revealing a global mechanism for transforming ambiguous sensory evidence into goal-directed actions. We show that evidence integration is a widespread phenomenon that emerges with learning and is implemented in a sparse population of neurons across most premotor areas. In these neurons, the timescales of integration are independent of intrinsic regional dynamics, suggesting that they are shaped by task experience. Notably, evidence integration and movement preparation are encoded in the same subspace of population activity across the brain, orthogonal to movement-related dynamics. Activity in this subspace was driven by neurons integrating evidence and collapsed at movement onset, allowing the integration process to reset, whereupon activity transitioned into a different subspace for movement execution concurrently across the brain. Our work links evidence accumulation onto motor dynamics on a brain-wide scale, unifying concepts from motor control and decision-making fields into a common framework for understanding how sensory evidence controls actions through global neural mechanisms.

Our finding that only expert mice exhibited robust encoding of visual input in almost all brain areas outside the visual system is consistent with previous reports of learning increasing the connectivity and correlations between cortical and subcortical regions[58–60], which may explain the distributed encoding of task variables across cortical and subcortical structures in trained animals[3,4,14]. We now show that these learning-induced multi-regional representations of task-relevant stimuli are not simply a distributed echo of the sensory input, but a transformed and integrated representation explicitly used to guide decisions. In association and premotor areas, such as frontal-premotor cortex, basal ganglia, cerebellum, parts of midbrain and thalamus, the prolonged responses to individual samples of evidence enabled their integration on a timescale of several hundred milliseconds, consistent with timescales of behavioural integration (Fig. 3 and Extended Data Fig. 6). This is a key distinction from visual areas, such as VISp and SCs (and primate middle temporal visual area[37] (MT)), where neurons do not integrate evidence (Fig. 3). Consequently, the integration of ambiguous task-relevant stimuli becomes a multi-regional distributed process implemented in a sparse population of neurons, and one that emerges with training as mice learn the value of the relevant stimulus

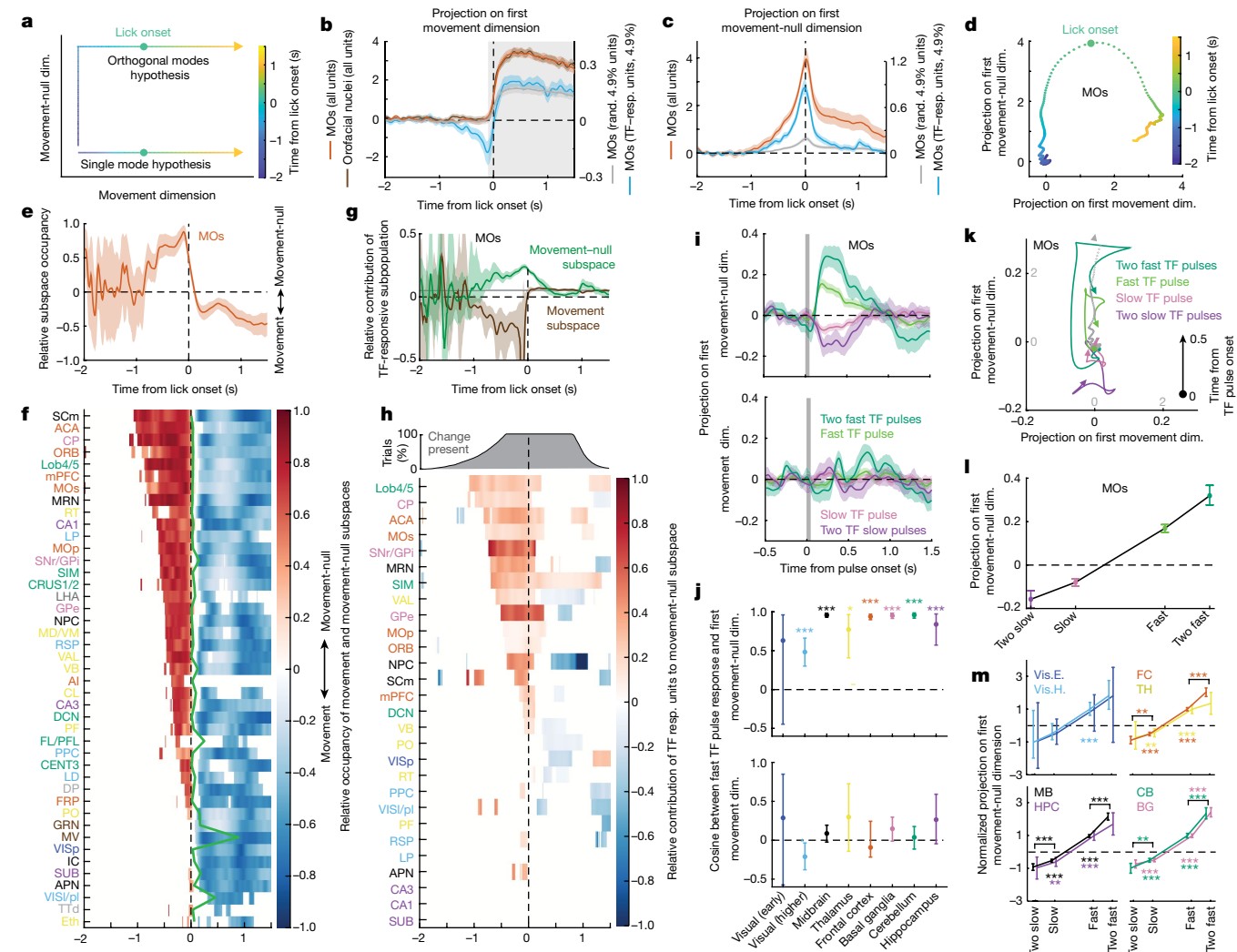

**Fig. 6 | Preparatory activity occupies movement-null subspace, is dominated by TF-responsive subpopulation and is aligned with responses to pulses of sensory evidence. a**, Schematic of two hypothetical ways population activity can transition from movement preparation to execution. Preparatory activity and action execution proceed either along the same mode of activity (single mode hypothesis) or are orthogonal to each other (orthogonal modes hypothesis). Dim., dimension. **b**, Mean projection of all MOs neuron activities around lick on hit trials onto the first movement dimension, defined by activity in orofacial nuclei in the time window around lick (grey; see Methods). Projection of activity of TF-responsive subpopulation of MOs is shown in blue (Methods; scale on the right); projection from a random (rand.) sample of MOs neurons (grey; matched to number of TF-responsive neurons; scale on the right). **c**, Projection of MOs activity onto the first movement-null dimension during hit trials. **d**, Same as **b**,**c**, but shown in a state-space formed from first movement and movement-null dimensions. Dots correspond to the state of MOs activity in 10-ms bins. Time relative to lick onset is indicated by colour. **e**, Relative occupancy of MOs activity in movement versus movement-null subspaces as a function of time (Methods). **f**, Same as **e**, but across brain regions (excluding brain regions with poor goodness of fit ($R^2 < 0.8$) to activity in orofacial nuclei; Extended Data Fig. 10d). Only time points with relative occupancy significantly different from zero ($P < 0.05$, bootstrap test) are shown (also for **h**). Brain regions are sorted according to the earliest latency of significant relative occupancy. Time of peak occupancy in movement-null subspace is shown by the green line.

**g**, Relative contribution of TF-responsive subpopulation to movement-null and movement subspaces. The grey line indicates the value expected from a random sample of neurons from MOs (matched to number of TF-responsive neurons). **h**, Same as **g**, but shown across brain regions sorted by latency of significant contribution of TF-responsive subpopulation. Top, fraction of trials with ongoing change epoch. **i**, Projections of MOs population responses to pulses of sensory evidence onto the first movement-null (top) and movement (bottom) dimensions. **j**, Cosine of the angle between population response to a fast TF pulse and first movement-null (top) and movement (bottom) dimensions. Data pooled across grouped brain regions (mean ± 95% confidence interval; bootstrap test). **k**, MOs population responses to pulses of sensory evidence (0–0.5 s after the pulse onset), shown in state-space formed by first movement and movement-null dimensions. Overlaid, MOs preparatory activity (grey) up to 100 ms before hit-lick onset (note the different scale). **l**, Peak value of projections of MOs responses to a slow or fast TF pulse, or two sequential fast or two sequential slow TF pulses, onto the first movement-null dimension. **m**, Same as **l**, but for groups of brain regions (bootstrap test). BG, basal ganglia; CB, cerebellum; FC, frontal cortex; MB, midbrain; Vis.E., visual (early); Vis.H., visual (higher). In all panels, shaded regions or error bars indicate bootstrapped 95% confidence intervals (Methods). Values of *n* for each brain region or brain region group are presented in Supplementary Table 1 and definitions of brain area abbreviations can be found in Supplementary Table 2.

feature. Notably, in our task both neural and behavioural evidence integration is 'leaky', consistent with the idea that in dynamic sensory environments perfect integration is not an optimal behavioural strategy[52]. Instead, leaky integration of a noisy stimulus stream is beneficial as it increases the signal relative to noise by temporally smoothing the input (Extended Data Fig. 7k–m).

We found that the timescales of integration are as diverse across the entire brain as has been shown across cortex[8,14]. However, evidence

integration times were not explained by the intrinsic timescales within each area, previously suggested to be predictive of response duration and ability to integrate stimuli in cortex of non-human primates and mice, respectively[8,23–26]. A possible reason for this discrepancy may be that our task allows estimation of both the intrinsic timescales and stimulus integration times in the absence of potentially confounding movement signals. In this study, we found that intrinsic timescales remain stable with learning, confirming they are an inherent property of each area. In fact, decoupling of intrinsic timescales from integration times may be advantageous because it allows task demands to sculpt the timescales of integration[26,61]. This decoupling may be implemented by learning mechanisms[59,62] that shape the activity propagating in multi-regional long-range loops involving cortex, basal ganglia, cerebellum, thalamus and midbrain, as observed during motor planning[28,29,35,53,54].

To understand how evidence integration leads to action, we adopted a framework developed for understanding the neural dynamics of movement generation, which identifies the relationship between modes of population activity that precede and follow action onset[33,34,63]. Using this framework, we demonstrate that neural dynamics of lick preparation and lick execution occupy distinct, orthogonal subspaces in most subdivisions of the brain, as previously shown in primate primary and premotor cortex during arm movements[33,34] and more recently in the mouse brain during memory-guided movements[4]. Of note, the subpopulations of neurons capable of integrating sensory evidence initiated and dominated preparatory activity in movement-null subspace. We found preparatory activity to originate earliest in regions with the longest integration timescales, such as frontal cortex, basal ganglia and cerebellum, and then transition abruptly into an orthogonal subspace upon movement initiation almost instantaneously in all brain regions investigated. This demonstrates that the transformation of accumulated evidence into movement planning and execution takes place within and across subspaces of neural activity that are shared across multi-regional circuits, rather than proceeding successively across a subset of specialized brain areas. Future research should determine the degree to which the principles of brain-wide neural dynamics observed in our study generalize to tasks involving multiple sensorimotor contingencies.

A clear advantage of orthogonalizing neural dynamics during decision-making is that it allows computations such as evidence accumulation, movement preparation or movement execution to proceed within the same population of neurons[64]. Our results highlight a particular advantage of occupying the movement-null subspace as it allows evidence integration to take place without directly causing movement. Accordingly, the lack of responses to visual evidence in the orofacial nuclei in medulla, which become active only upon lick initiation, demonstrates that brain-wide preparatory activity patterns driven by sensory evidence are incapable of driving the activity in motor circuits that control mouth and tongue movements.

The transition of population activity from movement-null to movement subspace is thought to proceed via a brief release of activity occupying movement-null subspace as an input to the movement subspace[34], which triggers the action. In a delayed response task using an explicit auditory Go cue, a trigger signal in premotor cortex depends on a pedunculopontine nucleus (PPN)/MRN–thalamic circuit[35]. Our task, however, requires an internally generated trigger when sufficient evidence is accumulated. Future work is needed to elucidate the regions that generate the trigger signal, with likely candidates receiving information from areas with early onsets of preparatory activity such as ACA, MOs, CP and Lob4/5. Conversely, an action initiation signal may propagate to the movement-null subspace, since the contribution of evidence-accumulating neurons to the movement-null subspace collapsed shortly following action onset, even though the change stimulus was still present, thus allowing the integration process to reset. This observation suggests that evidence-integrating neurons perform this

function only when it is relevant and before the mouse has committed to an action. These findings imply that activity in one orthogonal subspace can influence the activity in the other subspace, highlighting the dynamic interplay between movement-null and movement-related neural dynamics.

In summary, we demonstrate that learning recruits a neural subpopulation that is widely distributed across the brain, which concurrently integrates evidence and drives movement preparation, allowing sensory evidence to control global neural dynamics required for generation of behavioural responses.

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

# Methods

## Animals

All experiments were performed under the UK Animals (Scientific Procedures) Act of 1986 (PPL: PD867676F) following local ethical approval by the Sainsbury Wellcome Centre Animal Welfare Ethical Review Body. A total of 21 C57BL/6 J male mice (age = 34.5 ± 15.8 weeks (mean ± s.d.)) were used for electrophysiological recordings. Fifteen mice first underwent head-fixed behavioural training prior to acute electrophysiological recordings (see 'Task and training stages'), and six mice (untrained mice) only underwent habituation to the recording setup prior to acute electrophysiological recording.

Prior to behavioural training and recordings, all mice were implanted with a head-fixation bar under approximately 1.5% isoflurane and administration of Meloxicam (5 mg kg$^{-1}$) to allow for head-fixation during behavioural training and electrophysiological recordings.

During training, mice were co-housed with littermates in individually vented cages. After implantation of the recording chamber, mice were singly housed to protect the implant. Mice were housed in reversed day–night cycle lighting conditions, with the ambient temperature and humidity set to 23 °C and 56% relative humidity, respectively.

## Behavioural task

The design of the behavioural task was as previously described in ref. 14. In brief, mice were head-fixed and placed on a polystyrene wheel. Two monitors (21.5 inch, 1,920 × 1,080, 60 Hz) were placed on each side of the mouse at approximately 20 cm from the mouse head. The monitors were gamma corrected to 40 cd m$^{-2}$ of maximum luminance using custom MATLAB scripts utilizing PsychToolbox-3. The stimulus presentation was controlled by custom written software in MATLAB utilizing PsychToolbox-3. The visual stimulus was a sinusoidal grating with the spatial frequency of 0.04 cycles per degree resulting in 3 grating periods shown on a screen. Each trial began with a presentation of a grey texture covering both screens. After a randomized delay (at least 3 s plus a random sample from an exponential distribution with the mean of 0.5 s), the baseline stimulus appeared. The TF of the grating was drawn every 50 ms (3 monitor frames) from a lognormal distribution, such that $\log_2$-transformed TF had the mean of 0 and s.d. of 0.25 octaves and the geometric mean of 1 Hz. The direction of drift was randomized trial to trial between upward or downward drift. The sustained increase in TF, referred to in the text as change period, occurred after a randomized delay (3–15.5 s) from the start of baseline period and lasted for 2.15 s. For early and late blocks training (stage 8), change period times were sampled between [3, 8] s and [10.5, 15.5] s, respectively, with the delay from the earliest allowed change period sampled from an exponential distribution with a mean of 4 s. Random 15% of trials were assigned as no-change trials and did not have a change period. For stage 8 training, 10% of trials were designated to be probe trials and had a change time drown from the distribution of the other block type. Because there were no qualitative differences in neural TF pulse response between early and late blocks (data not shown) we have combined data from both block types for analyses throughout this manuscript. Findings related to stage 8 (early and late blocks) will be presented in an upcoming paper.

Mice were trained to report sustained increases in TF by licking the spout to trigger reward delivery (drop of soy milk). Licks that occurred outside of the change period are referred in the text as early licks. If mice moved on the wheel (movement exceeding 2.5 mm in a 50-ms window) in either direction, the trial was aborted (stages 7 and 8). If mice did not lick within 2.15 s from the change onset, the trial was considered a miss trial.

**Training stages.** Following the implantation of the headplate, mice were allowed to recover for a week. After that, mice went through several stages of training:

(1) Mice were handled for 3 to 7 days, until mice were comfortable with being handled by the experimenter. During this stage mice were also habituated to being restrained by being placed into a soft cloth for a short period of time. After the brief restraints they were given a small amount of soy milk as reward.

(2) Next, mice were put on food restriction. Mouse weight was monitored daily with the amount of food given adjusted per mouse to keep them sufficiently motivated for getting rewards and keep their weight no lower than 85% of the original weight prior to food restriction.

(3) Next, mice were head-fixed and placed on the running wheel of the behavioural training setup with the monitors turned off. Mice were allowed to freely run on the wheel, but not encouraged to. Typically, there were 3 habituation sessions, with the duration progressive increasing from 15 to 45 min.

(4) Next, mice were introduced to the visual stimuli used in the task. Mice were initially shown only trials with two largest changes of TF (2 and 4 Hz, lasting 2.15 s), followed by a reward auto-delivery 1.5 s after the change onset. After mice started to robustly make licks during the change period that preceded the reward auto-delivery, they were transitioned to the next stage.

(5) Here only hit trials were rewarded, early licks and running did not result in termination of the trial.

(6) After mice robustly detected strong changes in the previous step, we introduced trials with weaker changes in TF (1.25 Hz, 1.35 Hz and 1.5 Hz). Additionally, a consequence of an early lick outside of the change period was a mild air-puff to the mouse's right cheek and a termination of the trial.

(7) After mice detected weaker changes as well (assessed as higher hit rate compared to no-change trials), they were transitioned to the next stage where in order to initiate the trial start (start of the baseline stimulus), mice were required to remain stationary on the running wheel for at least 3 s plus a random sample from an exponential distribution with the mean of 0.5 s. Additionally, after the trial start, a trial was aborted as a consequence of a movement on the wheel.

(8) Finally, after mice reached sufficient proficiency at the previous stage, early and late blocks were introduced. During the session start, a block type was randomly chosen. A block was defined as a period of the session during which a mouse completed 30 hit trials. After completion of a block of trials, the block type was switched to the other block type (early to late or vice versa).

Six mice that were used in the untrained control experiment (Fig. 4e–h) went through training stages 1–3 above. Following that, they were shown the same stimuli as the trained mice, with the difference that their movements on the wheel or licking the spout did not terminate a trial nor trigger reward. Instead, they were given rewards at random times with inter-reward intervals drawn from the uniform distribution of 60 ± 15 s.

**Behavioural setup and data acquisition.** Reward delivery (soya milk) was controlled by a solenoid pinch valve (161P011, NResearch) and delivered to the mouse via a spout positioned in front of it. Mouse licking the spout was measured by a piezo element (TDK PS1550L40N) coupled to the spout and amplified with a custom-made amplifier system. Running wheel movement was measured with a rotary encoder (model Kübler) that was connected to the wheel axle. All behavioural data and events, such as piezo signal voltage trace, valve or change period on/off state, etc., were acquired via analogue and digital channels of PXIe-6341 acquisition card (National Instruments) with SpikeGLX (https://github.com/billkarsh/SpikeGLX) at 8,474 Hz.

## Behavioural data analysis

**Psychometric performance, reaction times and lick-triggered stimulus average.** Psychometric curves were calculated per session by counting the amount of hits relative to all trials where mice did not

early lick nor abort. Mean hit rates (performance) and parametric 95% confidence intervals (s.e.m. × 1.96) of hit rates were calculated across sessions ($n$ = 114) per change size. Mean reaction times and parametric 95% confidence intervals were calculated across sessions ($n$ = 114) per change size, and $p$-values were estimated from $t$-tests.

Lick-triggered stimulus average was estimated by extracting the TF pulses from −1.5 to 0 s preceding early licks and averaged across all trials, revealing mean stimulus TF prior to early licks. Parametric 95% confidence intervals were estimated by calculating the s.e.m. of TF values at each 50 ms bin (TF pulse resolution) prior to an early lick and multiplying the s.e.m. by 1.96.

**Simple behavioural leaky-integrator model.** In order to formally test if mice behaviourally integrated stimulus evidence (TF pulses) over time in our task, we constructed a simple behavioural leaky-integrator model with two adjustable parameters: decay time ($\tau$), and threshold. We fitted these two parameters by estimating which decay time and threshold predicted most early lick times (from 2 s after trial start, to exclude trial onset licks) correctly for each mouse and then determined the average best-fit decay time and threshold values across mice. For each early lick trial, we calculated the integrated log-scaled TF with decay across the entire trial up until the early lick.

For each early lick trial, we then estimated whether a threshold crossing of the integrated TF had been predicted within a second preceding an actual early lick onset. If this was the case, we considered the model to have predicted the early lick time. If not, we considered the model to not have correctly predicted that trial. We did this for all early lick trials, using a 58 × 151 parameter space: 58 possible decay times spanning from 0.05 s decay time (that is, no integration) to 1,000 s decay time (that is, perfect integration): (50 log-spaced decay times spanning 0.050–3 s, as well as 8 additional very long decay times: 4, 5, 6, 7, 8, 9, 20, 1,000 s), and 151 linearly spaced thresholds spanning [0.01–0.16]. Significance testing of best decay time across mice (that is, larger than no integration (0.05 s)) was done with a $t$-test.

We also tested if the best-fit decay/integration time parameter estimated from predicting early lick times also outperformed a model with no integration when predicting single-trial hit reaction times (that is, a trial type which the parameters were not optimized on). We did this by comparing actual and predicted reaction times per change size, and calculated Pearson's correlation between actual reaction times and predicted reaction times per change size. We calculated this by either looking at all reaction times, or only including a subset of trials with reaction times under a defined value (that is, reaction-time cut-off). This was done to better detect if any of the models specifically struggled to predict very late reaction times which may be modulated by non-sensory factors such as such as inattention or lack of engagement. Finally, for significance testing (that is, paired $t$-test) of whether a model with no integration (decay time = 0.05 s) versus a model with the best-fit decay/integration time (estimated from early lick trials as described above), were significantly different at predicting single-trial reaction times, we z-scored actual and predicted reaction times per change size (to account for change size mean reaction-time differences), and calculated the correlation between all actual reaction times (1 s reaction-time cut-off) and all predicted reaction times of a model with or without integration per mouse, and performed a paired $t$-test (across mice) of the correlation values from integration versus no integration models.

**Outlier detection agent.** To test whether mice accumulate evidence over time or merely respond to the instantaneous stimulus, we formulated a null model where behavioural responses are produced via a stochastic outlier detection strategy. Here, an internal decision occurs when a noisy sensory representation of the stimulus crosses a decision boundary, and a response occurs after a stochastic delay. The response is triggered by a single, instantaneous value of the stimulus. However,

owing to the stochastic delay, responses may show a gradually decaying statistical dependence on the stimulus history, and may even mimic evidence accumulation strategies such as integration[42].

*Model.* According to the outlier detection model, behavioural responses are generated independently for each trial as follows. Let $s_i$ be the stimulus amplitude (log TF) at each time point $t_i$. We chose time points to correspond with video frames of the stimulus, which were presented at 60 Hz (3 frames per TF pulse). At each time point, a noisy sensory representation $Z_i$ is formed as the sum of the stimulus amplitude and independent and identically distributed (i.i.d.) Gaussian sensory noise $\varepsilon_i$ (with mean zero and variance $\sigma^2$):

$$Z_i = s_i + \varepsilon_i$$

$$\varepsilon_i \widetilde{\text{i.i.d.}} \mathcal{N}(0, \sigma^2)$$

An internal decision to respond occurs at time $D$, given by the first time point where the sensory representation exceeds a decision bound $b$ (or $\infty$ if the bound is not crossed before the stimulus ends):

$$D = \min\{t_i | Z_i > b\} \cup \{\infty\}$$

The hazard function of the decision time is thus:

$$H_D(d) = \prod_{i | t_i \leq d} p(Z_i \leq b) = \prod_{i | t_i \leq d} \Phi\left(\frac{b - s_i}{\sigma}\right)$$

where $\Phi$ is the standard normal cumulative density function (CDF).

A motor response begins at time $R$, given by the decision time plus an independent, nonnegative stochastic delay $\Delta$ representing the duration of nondecision processes (for example, decision to motor delays):

$$R = D + \Delta$$

The delay has a shifted log-logistic distribution with location $\alpha$, scale $\beta$ and shape $\gamma$, and can be obtained by exponentiating a logistic random variable and then adding a constant. We constrained the location ($\alpha > 0$) and shape ($\gamma > 1$) to give the distribution nonnegative support and a bump-like density that decreases on both sides of the mode. The delay time probability density function (PDF) and CDF are:

$$p_\Delta(\Delta) = \frac{\gamma \left(\frac{\Delta - \alpha}{\beta}\right)^{\gamma - 1}}{\beta \left(1 + \left(\frac{\Delta - \alpha}{\beta}\right)^\gamma\right)^2}$$

$$F_\Delta(\Delta) = \frac{1}{1 + \left(\frac{\Delta - \alpha}{\beta}\right)^{-\gamma}}$$

Because the decision and delay times are independent, the marginal response time distribution is given by the convolution of the decision and delay time distributions. The marginal PDF and CDF of the response time are:

$$p_R(r) = \sum_d p_D(d) p_\Delta(r - d)$$

$$F_R(r) = \sum_d p_D(d) F_\Delta(r - d)$$

where the decision time probability mass function (PMF) $p_D$ can be computed from the hazard function $H_D$ above. Because delays are nonnegative, $p_\Delta(r - d) = F_\Delta(r - d) = 0$ for all $d > r$, so the above sums need only be computed over time steps up to the given response time.

The outlier detection model was implemented using custom Python software using the NumPy, SciPy, and PyTorch libraries. All computations involving probabilities were performed in log space, using functions designed to avoid numerical under/overflow.

*Fitting.* A separate model was fit for each mouse in two stages. We first fit the delay time distribution using only trials with the largest change magnitude, then fit the remaining decision parameters using the entire dataset (excluding the abort trials). This two-stage approach relies on the assumption that delays are identically distributed across trials. In return, it allows more direct estimation of the delay time distribution, providing better ability to distinguish between outlier detection and longer-timescale strategies such as integration.

For each trial $i$, let $n^{(i)}$ be the number of time points, $s^{(i)} = \{s_1^{(i)}, ..., s_{n^{(i)}}^{(i)}\}$ be the stimulus amplitudes, and $c^{(i)}$ be the time of the change point. For trials where a response occurred, let $r^{(i)}$ be the response time, measured as the onset of facial movement (see 'Motion onset time estimation' section) and $\ell^{(i)}$ be the subsequent lick time (measured at the reward spout).

*Fitting the delay time distribution.* We assumed that the greatest change magnitude (geometric mean TF 4 Hz) was large enough to trigger an immediate decision at or near the change point. Under this assumption, the delay time on large-change hit trials can be approximated by the reaction time, which can be directly measured as the time elapsed between the change point and the onset of facial movement. Thus, we fit the delay time distribution (shifted log-logistic distribution) to reaction times on large-change hit trials (denoted $\mathcal{T}_{\text{bighit}}$) by maximum likelihood, subject to the constraints described above:

$$\max_{\alpha > 0, \beta > 0, \gamma > 1} \sum_{i \in \mathcal{T}_{\text{bighit}}} \log p_{\Delta}(r^{(i)} - c^{(i)})$$

This approach is conservative for our use of outlier detection as a null model. If the largest changes were not immediately followed by a decision, then delays would tend to be overestimated, causing the fitted outlier detection model to display longer-timescale dependencies that are typically associated with evidence accumulation strategies such as integration. Thus, the risk of falsely rejecting this null model would not increase.

For the largest change magnitude, miss trials predominantly reflected task disengagement rather than typical sensory/motor delays, and were therefore excluded when fitting the delay time distribution. According to a hidden Markov model, disengagement was the a posteriori most probable state for the majority of large-change miss trials (95.2% of large-change misses were during a disengaged state).

*Fitting decision parameters.* The decision parameters (sensory noise variance and decision threshold) were subsequently fit using the entire dataset, holding the delay time distribution fixed. Here in the general case, the decision and delay times cannot be directly observed, and were marginalized out as latent variables. The decision parameters were chosen to maximize the log marginal likelihood of the observed response data:

$$\max_{\sigma^2, b} \sum_{i \in \mathcal{T}_{\text{nonmiss}}} \log p_R(r^{(i)}) + \sum_{i \in \mathcal{T}_{\text{miss}}} \log(1 - F_R(t_{n^{(i)}}))$$

For hit and early lick trials (denoted $\mathcal{T}_{\text{nonmiss}}$), the likelihood is given by the marginal probability density of a response at the observed movement onset time. For miss trials (denoted $\mathcal{T}_{\text{miss}}$), the response time is treated as right-censored; its precise value is unknown, but is known to exceed the last time point in the trial. The likelihood for miss trials is thus given by the marginal probability mass lying beyond this point.

*Sampling.* To statistically compare mouse behaviour to the outlier detection null model, we sampled 10,000 synthetic datasets from the model fitted for each mouse. For every quantity of interest, the value computed from the real data was compared to values computed from

each synthetic dataset, comprising 10,000 samples from the null distribution. Synthetic datasets were generated for each mouse as follows.

Each trial used the same change point and stimulus amplitudes presented in the real data. The real stimulus ended after the lick on trials where mice responded, leaving unknown future values that would have been presented had a lick not occurred. Such missing stimulus values were filled in by sampling from the same distribution used to produce the original stimuli (independently for each synthetic dataset).

Given the stimulus, a decision time and delay time were sampled from the distributions $p_D$ and $p_{\Delta}$ described above. The sum of these quantities yielded a synthetic response time, representing movement onset.

To generate synthetic lick times, we assumed that the additional delay between movement onset and licking was i.i.d. across trials. We therefore sampled with replacement from the measured movement-to-lick delays in the real data. Synthetic lick times were obtained by adding sampled movement-to-lick delays to synthetic movement onset times.

Synthetic lick times were used to determine trial outcomes (hit, early lick, miss). Each trial was classified as a: hit if the lick occurred during the change period; early lick if the lick occurred before the change point; or miss if no lick occurred before the end of the change period.

**Effect of magnitude and timing of TF pulses on probability of early licks.** For analyses of the effect of TF pulses on probability of early licks we used the training data of the same 15 mice used for Neuropixels recordings. Here we only used sessions where mice reached robust proficiency of the task and were at the final training protocol (mean of 77.5 sessions per mouse). Note that here the time of lick onset was measured from the registration of lick by the spout as opposed to the videography analysis on Neuropixels recording sessions elsewhere in the manuscript. We used only trials where early licks happened at least 2 s after the baseline onset to decrease the influence of impulsive licks on results.

To empirically validate that mice use multiple pulses of sensory evidence to influence their decision to lick during the baseline period, we analysed how early lick probability is influenced by magnitudes and timing of preceding TF pulses. First, we tested whether the deviation of a single TF pulse relative to the mean baseline 1 Hz makes mice correspondingly more or less likely to make an early lick within the subsequent 0.2–1.0 s. For that we separated TF pulses by magnitude (in octaves) into 15 bins such that each bin contained approximately equal number of TF pulses. To calculate the conditional probability of early lick at a certain time after a TF pulse of a given magnitude, we found instances of such events (pulled across all sessions with robust performance for each mouse) and divided them by the total amount of early licks (Extended Data Fig. 6c). To calculate an overall influence of a TF pulse on early lick probability, we summed conditional probabilities within a [−1, −0.2] s window relative to early lick onset (Extended Data Fig. 6d):

$$P(\text{L}|\text{TF}) = \sum_{t=-1}^{-0.2} P(\text{L}|\text{TF}(t))$$

which can also be written as: $P(\text{L}|\text{TF}) = P_0 + \Delta P(\text{L}|\text{TF})$, where

$$P_0 = \sum_{t=-1}^{-0.2} P(\text{L}|\text{TF}(t) = 1 \text{ Hz})$$

And can be thought as a chance level of making a lick without a deviation of stimulus TF from the mean baseline TF value.

The empirical effect of two TF pulses on lick probability was calculated from behavioural data in a similar way. To compare the measured effect of two TF pulses with their expected effect if they influenced the lick probability independently, we calculated their cumulative independent effect on early lick probability based on empirically measured

effect of a single TF pulse on early lick probability. The independent effect of two TF pulses with a delay of $\Delta t$ s between them can then be written as follows:

$$P_{Ind}(L|TF_1, TF_2) = P_0 + \Delta P(L|\Delta TF_1) + \Delta P(L|\Delta TF_2) - \Delta P(L|\Delta TF_1) \times \Delta P(L|\Delta TF_2)$$

where:

$$\Delta P(L|\Delta TF_1) = \sum_{t=-1-\Delta t}^{-0.2-\Delta t} P(L|TF(t)) - P_0$$

$$\Delta P(L|\Delta TF_2) = \sum_{t=-1}^{-0.2} P(L|TF(t)) - P_0$$

A deviation of lick probability after two TF pulses from the probability predicted by the independent effect of two TF pulses would indicate an interactive effect between pulses, which should be expected if mice utilize integration of sensory evidence. To measure the relative difference between the behavioural result and the expected independent effect of two fast TF pulses (Fig. 3d and Extended Data Fig. 6i), we calculated:

$$I = \frac{P(L|TF_{fast}, TF_{fast}) - P_{Ind}(L|TF_{fast}, TF_{fast})}{P_{Ind}(L|TF_{fast}, TF_{fast})}$$

When applying this analysis to the outlier detection agent data, we used data only from trials that resulted in early licks, meaning that the model made a decision to initiate a lick during the baseline period and before the TF change epoch. For outlier detection agent model that was fitted to a particular mouse data, we sampled the same number of early lick trials across 4,000 synthetic datasets (see section above) as there were present across all behavioural sessions of that mouse. The data was then pulled across all models corresponding to different mice and analysis steps were applied to the combined dataset as described above for the mice data. This procedure was repeated 4,000 times to estimate non-parametric 95% confidence intervals of results from the outlier detection agent.

## Electrophysiological recordings

Prior to acute electrophysiological recordings, we habituated mice to the electrophysiological recording setup for 2–7 days (depending on the performance of the mouse in the electrophysiological recording setup), to allow mice to perform optimally during electrophysiological recording sessions.

**Surgery.** Once mice were habituated to the recording setup, we implanted a recording chamber with one or two 3 mm craniotomies inside, together with a stainless-steel grounding wire in the contralateral hemisphere, under 1.5% isoflurane together with administration of meloxicam (5 mg kg$^{-1}$) and dexamethasone (2–3 mg kg$^{-1}$). During surgery a kapton disk (Laser Micromachining Limited) was placed on top of the dura inside each craniotomy. The disk had 19 holes with 0.5 mm diameter, arranged in a honeycomb shape, for keeping track of probe insertions. The craniotomy and disk were covered with DuraGel (Cambridge NeuroTech) to protect the brain. A 1–2 mm tall plastic enclosure was then positioned around craniotomies and sealed around the edges with bone cement. Finally, we covered the plastic enclosure with a removable plastic cover, to create a rigid physical barrier over the DuraGel sealed craniotomy, to provide robust protection of the recording preparation between recording sessions. The mice were allowed to recover for 24 h before the first recording session took place.

**Recordings.** Electrophysiological data collection was done using Neuropixels 1.0 probes (IMEC, Belgium) and collected with a PXI based system (National Instruments), and saved using SpikeGLX (https://github.com/billkarsh/SpikeGLX). For trained mice, we recorded up to 13 sessions per mouse (167 probe insertion from 114 sessions total (15 mice)). For untrained mice, we recorded up to 9 sessions per mouse (89 probe insertions from 45 sessions total (6 mice)). Probes were dipped in CM-DiI (Sigma-Aldrich) prior to insertion. In each session, we inserted up to 2 probes at a time. The probes were always inserted at the same angle within the coronal plane (10° and −15° relative to the vertical axis) to aid subsequent histological probe tract tracing.

At the beginning of each session, we removed the plastic lid above the recording chamber exposing the DuraGel covered craniotomy, and inserted the probe(s) through the DuraGel using microcontrollers (Sensapex) at 5–10 μm s$^{-1}$. The probe(s) was allowed to settle for 20 min, to increase stability throughout the recording session. At the end of the session probes were removed (at 15 μm s$^{-1}$) and the plastic cover over the recording chamber was reattached for protection of recording preparation.

The setup for presenting stimuli and monitoring behaviour were identical to the setups in which mice had been trained (see 'Behavioural task').

**Pre-processing and spike sorting of electrophysiological data.** Electrophysiological data was first filtered using CatGT (https://billkarsh.github.io/SpikeGLX/#catgt) with modified form of common average referencing (-dlbdmx flag).

*Spike sorting.* We spike-sorted electrophysiological data from each probe in each session using KiloSort2.0[65] (https://github.com/MouseLand/Kilosort). For initial selection of units undergoing further curation, we only selected units designated as 'good' (based on cross-correlogram contamination) by KiloSort2.0.

*Quality checks.* For our electrophysiological recordings of trained mice, we manually inspected and curated, in Phy2.0 (https://github.com/kwikteam/phy), every unit which KiloSort2.0 had designated as 'good'. For our recordings in trained mice this left 44,288 units to be manually inspected and curated, and 15,406 units were kept for analysis after manual curation. Based on the manual curation data from trained mouse recordings (see 'Manual curation of spike-sorted units from trained mice'), we established a series of heuristics for creating automatic curation of units (see 'Automatic curation of spike-sorted units from untrained mice') and used these for recordings from untrained mice.

Manual curation of spike-sorted units from trained mice. We manually inspected and curated all units which KiloSort2.0 had designated as good, based on cross-correlogram contamination. In Phy2.0, we first inspected and merged units that clearly belonged to the same cluster, but had been split by KiloSort2.0, or split the noise from signal in units with clearly separable noise contamination. We then designated each unit into one of five categories:

(1) Perfect, or almost perfect, with no/very minimal noise, drifting, cutting in/out for the full duration of recording.
(2) Usable and good signal with some noise that cannot be extracted that lasts for the full duration of the recording.
(3) Some drift, but possible physiological change in signal. Clear signal for most of duration of the recording.
(4) Drifting/sudden loss, but otherwise usable/close to perfect. Clear signal for over 50% of the duration of the recording but requires only using a subset of the session.
(5) Noise/useless. Spike shape is not physiological.

Our goal was to remove from analyses units that had large contamination with multi-unit activity, were not recorded throughout the full duration of a session, or were a result of artifacts in recorded signals. We therefore used units designated as category 1–3 above for all further analysis from trained mice.

Automatic curation of spike-sorted units from untrained mice. We next used the manual designations of units to establish a set of criteria for automatic detection of units we would include with manual curation.

Based on the manual curation data above we established the following 7 criteria for considering a unit good for analysis:

Firing rate criteria:

(1) Mean firing rate must be above 0.5 Hz.

(2) Rolling 20-min average firing rate cannot drop below 30% (that is, 70% drop from mean) of its mean firing rate.

(3) Rolling 10-min average firing rate cannot drop below 20% (that is, 80% drop from mean) of its mean firing rate.

(4) Rolling 5-min average firing rate cannot drop below 10% (that is, 90% drop from mean) of its mean firing rate.

(5) Inter-spike interval (ISI) violations. Absolute refractory period needs to have <20% estimated contamination rate from other neurons (this is what Kilosort2.0 calls 'good').

(6) If there are some spikes in the refractory period, the ISI peak in the first 5 ms cannot be within the first 2 ms.

(7) ISI histograms cannot have sudden large spikes in their shape (that is, peak of ISI cannot be 4 times larger than the second highest peak— that is usually its immediate neighbour).

These criteria selected approximately 90% of units we would have designated with categories 1 (perfect, or almost perfect) or 2 (usable and good signal with some noise) with manual curation, and excluded approximately 85% we would have designated as 4 (drifting/sudden loss) or 5 (noise) with manual curation.

This automatic selection of units was used to select units for analysis from untrained mice recordings and yielded 6,215 units out of 20,292 'good' KiloSort2.0 units.

*Clock-drift correction.* A shared 1 Hz square wave signal was recorded on the clock of each headstage and National Instruments (NI) acquisition card using a SYNC option in SpikeGLX. Clock drift between spike times from different probes and behavioural events extracted from NI acquisition card recording was corrected post-hoc via TPrime (https://billkarsh.github.io/SpikeGLX/#tprime) using the shared square wave signal.

### Videography

**Acquisition.** High-speed videography of front (100 frames s$^{-1}$, 640 × 512 pixels) and side view (50 frames s$^{-1}$, 976 × 1,024 pixels) of the mouse face was acquired using two Chameleon3 cameras (CM3-U3-13Y3M-CS, FLIR) with infrared illumination. The videos were acquired in an 8 bit greyscale format. Cameras were configured to send a TTL signal to the National Instruments PXIe board at the start of exposure of every acquired frame. These TTL signals were used to align frame times to the time of behavioural events and spike times.

**Pupil size.** In order to estimate the pupil size, we trained DeepLab-Cut[66] to track the pupil size and position using videos acquired with the side camera. The model was trained to track 12 points surrounding the mouse pupil. In order to assess the model performance, after the training the model was tested on videos from sessions not used for training. Pupil size was estimated as an area of an ellipsoidal best fit to the tracked 12 points surrounding the pupil.

**Motion energy.** For calculation of motion energy, we primarily used videos acquired with the front camera to access a finer temporal resolution (with the exception of 2 sessions where for technical reasons we used a lower jaw ROI from side camera video). To estimate motion onset times, we used ROI centred around the mouse's face, though nearly identical results were obtained with lower jaw or whisker pad ROIs from the side camera (data not shown). Motion energy was defined as a square root of the sum of squared frame-to-frame pixel value differences, divided by the number of pixels within the ROI.

**Movement onset time estimation.** In order to find the onset times of orofacial movements, we wanted to estimate the typical noise level of the motion energy signal and find the time points where the signal significantly deviated from the noise-band level. As a first step, we calculated the distribution of motion energy values in a 2-s window centred around the lick registration times. We next fitted a mixture of Gaussian distributions with the goal to capture both contribution of the variance of motion energy values during the lick as well as due to noise. The mixture of three Gaussian distributions worked well to fit the data across all sessions and mice. The threshold for the presence of movement was defined as the mean plus two standard deviations of the Gaussian with the lowest value of the mean from the Gaussian mixture.

Finally, to find the time of motion onset time, we looked backwards in time from the time of lick registration by the piezo signal. The time point preceding the first instance of motion energy going below the threshold value defined above was considered the onset time of the orofacial movement.

### Histology

For histological identification of the location of the recording probes and allocation of unit location in the mouse brain, we followed a protocol similar to ref. 67.

**Serial 2-photon tomography for Neuropixels probe tract tracing.** Following a terminal administration of pentobarbital, mice were perfused with a phosphate buffer solution (PBS) followed by 4% paraformaldehyde (PFA) solution. We post-fixed the brain in the 4% paraformaldehyde for a minimum of 24 h at approximately 5 C. Following fixation, brains were moved to PBS for a minimum of 12 h prior to imaging. For imaging, brains were embedded in 5% agarose gel and mounted onto a vibratome cutting stage under the microscope objective. The brains were imaged using serial section two-photon microscopy[68]. The microscope was controlled with ScanImage Basic (Vidrio Technologies), and custom software (BakingTray (https://github.com/SainsburyWellcomeCentre/BakingTray)). Images were stitched into a full 3D rendering of the brain using custom software (StitchIt (https://github.com/SainsburyWellcomeCentre/StitchIt)). We imaged the entire brain (from the olfactory bulb to the beginning of the spinal cord) with a resolution of *x*: approximately 2 µm, *y*: approximately 2 µm, *z*: 20 µm, with a 920 nm two-photon laser (100–150 mW power at sample). We sliced the brain in 40-µm sections, and imaged 2 *z*-planes (around 25 µm and around 45 µm from the tissue surface) into the remaining tissue following each 40-µm section. Two PMTs, one for capturing green (bandpass filter ET525/50 m) and red (bandpass filter ET570lp) fluorescence acquired the 2 channels of data subsequently used for analysis.

**Neuropixels probe tract alignment to the Allen Common Coordinate Framework atlas and estimation of unit location.** Prior to image processing, we downsampled microscopy images to 10-µm voxels and registered the brain to the standardized Allen Common Coordinate Framework (Allen CCF[69]) using custom software (BrainRegister (https://github.com/stevenjwest/brainregister)). We then manually traced each neuropixels probe tract through the brain in 3D using custom software (Lasagna (https://github.com/SainsburyWellcomeCentre/lasagna)). Finally, we assessed the overall firing rates and LFP spectra of individual Neuropixels channels and compared it to atlas positions. Where needed, we manually adjusted the scaling of brain regions along the probe track to align responses on channels with features associated with anatomical locations using custom software (Ephys alignment tool (https://github.com/int-brain-lab/iblapps/tree/master/atlaselectrophysiology)[70]). Unit location was estimated from the location of the channel that had the largest absolute peak value of the mean waveform. For all analyses, we combined units across all subdivisions of a brain region (layers of cerebral cortex, dorsal and ventral divisions as ACAd and ACAv and in some cases functionally similar brain regions—see Supplementary Tables 1 and 2).

## Neural data analysis

Only brain regions with at least 40 units were analysed. Analyses specific to TF-responsive units were done only for brain regions with ≥10 such units. No further sample size calculations were performed. Manual curation of units' quality and stability was done without the knowledge of brain regions from which recordings were made. The subsequent analyses pipeline was applied in the same manner to data from all applicable brain regions, but the custom nature of analyses prevented investigators to remain blind to the identity of brain regions or dataset type (trained versus naive mice).

**GLM of neural activity.** *Model.* We binned neural activity in 50-ms bins (matching the duration of each TF pulse) aligned to trial start. We then fitted a Poisson generalized linear model to predict trial-to-trial neural activity as a function of a set of temporally unfolded task-related predictors that were present during a trial. Each predictor was extended temporally prior and/or post the timing of the predictor in 50-ms discretized steps (matching neural activity binning), with an independent weight estimated for each time step around the predictor. We predicted neural activity using 19 task-related predictors:

(1) TF fluctuations during baseline period (kernel length: 0–1.5 s); (2) Trial start (0–1 s); (3) Time since baseline start (from 1 s from trial start to change onset); (4–9) Six change onsets (a separate predictor for each change size (0–2 s)); (10) Lick preparation (−1.25–0 s prior to lick); (11) Lick execution (0–0.5 s post lick); (12) Air-puff (0–0.25 s); (13) Reward (0–0.4); (14) Abort (−1.25–0.25); (15) Phase of grating for upwards drift (12 phase bins from 0–360°); (16) Phase of grating for downwards drift (12 phase bins from 0–360°); (17) Video motion energy (−0.05–0.8 s); (18) Running wheel movement (−0.05–0.8 s); (19) Pupil diameter (−0.75–0.75 s).

We fit the model with L2 (ridge) regularization, optimized with cyclical coordinate descent as implemented in GLMnet[71] ($\alpha = 0$). We trained a model for each neuron on 90% of the data, and cross-validated on 10% of the data, and iterated the predictions over a tenfold cross-validation. Within the training dataset we tuned the L2 regularization term using tenfold cross-validation.

*Identification of units encoding TF, lick preparatory activity and/or lick execution activity.* To identify which cells significantly responded to a predictor of interest (that is TF fluctuations during baseline, lick preparation epoch, or lick execution epoch), we first re-fitted reduced models similar to the full model on 90% of the data, with 10-fold cross-validation, except we removed a predictor(s) of interest: (1) For identification of TF-responsive units, we estimated a model where we removed the predictor estimating the responses to TF fluctuations during baseline. (2) For identification of units with lick preparation activity, we estimated a model where we removed the predictor estimating the activity leading up to a lick. (3) For identification of units responding to lick execution, we estimated a model where we removed the predictor estimating activity during lick execution, the predictor estimating activity modulation by motion energy captured by videography, and the predictor estimating activity modulation by running wheel movement.

For each 10% test set, for each neuron we then calculated the mean actual peri-event time histogram (PETH) as well as the mean predicted PETH of both the full model and the reduced model for the following types of events: (1) −0.15 to 0.75 s around fast and slow TF pulses (that is, TF values 0.5 s.d. from the mean TF during baseline); (2) −1.5 to 0 s prior to early lick onsets; and (3) 0 to 0.4 s post lick onset.

A unit was considered significantly encoding TF pulses during the baseline period if two criteria were satisfied: (1) The mean Pearson's correlation prediction of the full model (across k-folds) from the combined mean fast and slow TF pulse response (that is, mean fast TF pulse and mean slow TF pulse responses subtracted from each other) was >0.2; and (2) if the cross-validated prediction of the TF response after

subtracting the predicted TF response of the reduced model with no TF fluctuation predictor—that is, residual prediction—was significant ($P < 0.01$ (*t*-test), $n = 10$ independent cross-validations). A unit was considered significantly encoding lick preparation if (1) the mean Pearson's correlation prediction of the full model (across *k*-folds) of the mean activity leading up to a lick (−1.25 to 0 s) was >0.2; and (2) if the cross-validated prediction of the mean activity after subtracting the predicted mean activity of the reduced model with no lick preparation kernel—that is, residual prediction—was significant ($P < 0.01$ (*t*-test), $n = 10$ independent cross-validations). Finally, a unit was considered significantly encoding lick execution if (1) the mean Pearson's correlation prediction of the full model (across *k*-folds) of the mean activity following a lick (0 to 0.25 s) was >0.2; and (2) if the cross-validated prediction of the mean activity after subtracting the predicted mean activity of the reduced model with no lick preparation kernel—that is, residual prediction—was significant ($P < 0.01$ (*t*-test), $n = 10$ independent cross-validations).

*Focality index.* To assess how distributed TF encoding was across brain areas, before and after learning, we computed a focality index ($F$) (similar to Steinmetz et al.[3]) of the TF encoding:

$$F = \frac{\sum (p_a^2)}{\left(\sum p_a\right)^2}$$

where $p_a$ is the proportion of neurons in an area that is encoding stimulus TF during the baseline period. If all TF encoding neurons were confined to a single area, this measure would take on the value of 1. If encoding was perfectly distributed across all areas recorded this measure would take on the value $1/N_{\text{areas}}$. In order to compare between untrained and trained mice, we identified the common areas which had more than 40 units recorded in both trained and untrained mice. This left $N = 24$ areas from which to estimate the focality index. We estimated 95% confidence intervals and $P$ values by bootstrapping the neurons included in the estimation 10,000 times with replacement.

*Peak time and width of GLM estimated TF kernels for TF-responsive neurons.* To investigate the peak time and width of the GLM estimated TF kernel for assessing how sustained responses to TF fluctuations were based on GLM weights, we first identified the absolute peak value of the TF kernel; because the GLM was based on 50 ms binning of spike counts, peak times for the GLM TF kernel was in 50 ms resolution. In cases where the absolute peak position within 1 s was a negative weight, we flipped the kernel in order to calculate the width. We then estimated the full width at half maximum (FWHM) of each TF kernel around its peak using findpeaks in MATLAB. For each area, we calculated the median peak time and median FWHM across all TF-responsive units.

*Ramping differences in GLM change kernels.* To test how neurons accumulated evidence when they were presented with a rewarding sustained change in stimulus speed, we tested how the slope of the visual evoked ramping activity following a change onset was dependent on the amount of evidence (change size) being presented. To isolate the visual component of the activity following change onset, we used the GLM kernel which fits the activity following change onset until change offset, while linearly taking into account other variables which may contribute to activity such as pupil size, preparatory activity and movement-related activity (see Model).

We estimated the mean change kernel for each change size for TF-responsive and non-TF-responsive units separately for each area. In cases where responses to fast TF pulses were negative, we flipped the change kernel so every unit had responses aligned to positive fast TF pulses—this allowed the mean to capture the visual evidence activity ramp irrespective of sign. We then identified the time point for each change size where the change kernel reached 50% of its maximum weight (To control for noise fluctuations in kernel weights, we approximated the 50th percentile crossing by taking the mean time point of the

33.33rd percentile, 50th percentile and 66.66th percentile crossing). We then calculated the degree to which activity ramping time scaled with change size, by regressing the 50th percentile crossing against change size. We estimated the non-parametric 95% confidence intervals and $P$ values of the relationship between change size and 50th percentile crossing (that is, ramping time/change size) by bootstrapping with replacement (10,000 times) the neurons went into the mean change kernels, and then estimating the slope of the regression for each bootstrapped mean change kernels.

**Propagation and widening of TF pulse evoked activity.** *Identification of TF pulse outlier events.* Fast TF pulse was defined as TF fluctuations larger than 1 s.d. of baseline TF fluctuations (in $\log_2$ scale) above the mean TF value (TF > 1.19 Hz). Similarly, slow TF pulse was defined as TF fluctuations below 0.84 Hz.

For calculation of average response to TF outlier events, we considered only TF outlier events satisfying the following criteria:
(1) Later than 1 s from the baseline onset.
(2) Earlier than 2 s + post pulse analysis window from the motion onset time on early lick or abort trials.
(3) Excluding the change period plus a post pulse analysis window.

The aim of these criteria was to exclude the influence of baseline onset, movement, or preparatory activity on the response to TF pulses.
*Estimation of peak time and width of TF pulse evoked activity.* For each unit defined as TF-responsive by the GLM analysis described above, we calculated a mean response to a fast pulse using outlier events that occurred during the baseline period and satisfied the criteria outlined above. Additionally, we calculated a mean response to TF pulses within [−0.5, 0.5] s.d. of the baseline TF fluctuations. The goal of this procedure was to capture continuous ramps of activity that some units exhibited and exclude their influence on the shape of response to a TF pulse. We applied the subtraction of this baseline response for all TF pulse response analysis unless explicitly stated.

Next, for the baseline subtracted mean response to a fast TF pulse, we calculated its peak time, as the time of the largest absolute change in firing rate within 1 s from the pulse onset, and a corresponding half-peak width.
*Integration of multiple TF pulses.* Because the noise in TF fluctuations is random, by chance there are occurrences of two fast pulses separated by a certain delay. To study the integration of TF pulses, we found such instances of events where two fast pulses occurred at a given delay between the offset of the first and the onset of the second, additionally also satisfying the exclusion criteria outlined above. The mean response aligned to such events was considered a response to a sequence of two fast pulses.

For computing the mean response across all TF-responsive units within a brain region, in order to avoid averaging across responses with different signs, we flipped the sign of response for units that showed decreases in activity after a single fast pulse. For computing a $z$-score of response, the mean and s.d. were estimated from 0.5 s preceding the first pulse onset.
*Facilitation by the second fast pulse.* First, we measured an average of $z$-scored responses across the population of TF-responsive units within a brain region to a single fast TF pulse. We then computed the peak value of that response ($r_{1\text{fast}}$), and a corresponding peak time. To find the size of response to a sequence of two fast pulses ($r_{2\text{fast}}$), we found a time point at the same delay from the onset of the second fast pulse as the peak time of response to a single fast pulse and found a peak value of response within 100 ms centred around that time point. The relative facilitation to a sequence of fast pulses was defined as $\Delta = \frac{r_{2\text{fast}} - r_{1\text{fast}}}{r_{1\text{fast}}}$.
To determine the confidence intervals for the results of this analysis, we bootstrapped with replacement (2,000 times) across TF-responsive neurons and repeated the analysis described above for each sample

of neurons. Shaded regions indicate 2.5 and 97.5 percentiles of the resulting distribution.

**Preparatory activity before the lick onset.** To study change-aligned (Fig. 3) or hit lick-aligned (Fig. 5) activity, we computed $z$-score of mean PETH for each unit. $z$-Scoring was done using the mean and s.d. estimated from activity during 2 s before the change onset.

For analysis shown on Fig. 5, for each brain region the fraction of significantly active units within a group (that is, TF-responsive) was measured by calculating at every time point a fraction of units with the absolute value of $z$-score larger than the significance threshold of 2.576 (corresponding to $P < 0.01$). Additionally, we subtracted the 'baseline' level of activity calculated within [−2, −1.8] s before hit-lick onset, which for a few brain regions was larger than chance level likely due to non-normal distribution of firing rates or a small number of events used for estimation of the mean and s.d. The confidence intervals were estimated by bootstrapping with replacement (5,000 times) across TF-responsive (or TF non-responsive) neurons and repeating the estimation of fraction of significantly active neurons for each sample of neurons.

The latency of activation of TF-responsive or TF non-responsive populations was defined as the earliest time point following which within a 100-ms window for at least 80 ms: (1) the lower 95% confidence interval of fraction of active units was above zero; and (2) the mean fraction of active units was above 0.1.

The latency of significant difference in activation between TF-responsive and TF non-responsive populations was estimated as the first time point where within a 100-ms window for at least 80 ms the confidence intervals of the difference in activation were above zero.

The latency of significant difference in activation across all units in each brain region (Extended Data Fig. 9a) was estimated as the first time point where within a 100-ms window: (1) the lower 95% confidence interval of fraction of active units was above zero; and (2) the mean fraction of active units was above 0.05.

**Intrinsic timescales.** We binned the neural activity into 50-ms bins (same binning was used in ref. 23). We then calculated the temporal autocorrelation (20 lags = 1 s) of spike counts using Pearson's correlation in the inter-trial intervals between −2.5 s to −0.5 s prior to trial onset for each neuron (in this period mice were seeing a grey screen, and trained mice had to remain stationary for at least 3 s for the trial to begin).

To determine the intrinsic timescale for each area, we fit an exponential decay function to the mean autocorrelation function of all the units recorded in the area. For single-neuron analysis of relationship between intrinsic timescales and TF width, we estimated the autocorrelation for each TF-responsive neuron separately. For areas or neurons with autocorrelation functions with non-monotonic decay, we fit the exponential decay from the part of the autocorrelation where monotonic decay was happening (in a subset of areas this would mean offsetting the fit 1–3 time bins). Finally, we calculated the $\tau$ (that is, the intrinsic timescale value) of the exponential decay (accounting for offset where necessary).

## Population analysis

**Similarity of TF pulse responses and lick preparation activity in TF-responsive populations.** We assessed the similarity of TF responses and lick preparation activity across TF-responsive populations in each area by estimating the Pearson's correlation of mean firing rates (within a 50-ms window around the mean activity peak time across neurons within the each area) following fast TF pulses (that is, >1 s.d. TF value) and their mean activity prior to early lick [−0.3 to 0 s] (after normalizing firing rates by subtracting baseline firing rates from both TF responses and lick preparation activity). We estimated the non-parametric 95% confidence intervals and $P$ values by bootstrapping with replacement (10,000 times) the neurons going into the correlation.

**Pre-processing steps.** For all units located within a given brain region, but not necessary simultaneously recorded, we first computed the mean neural responses across a given trial type (for results shown on Fig. 6b–h: hit trials during weak TF changes (1.25 and 1.35 Hz) aligned to the lick onset times, [−2, 1.5] s time window). Only trials with hit-lick onset times larger or equal 0.4 s from change onset were used. Neurons from sessions with less than 10 trials of a given type were excluded from this analysis. Firing rates were calculated as spike counts averaged in 10 ms bins and smoothened by convolution with two-sided Gaussian with 30 ms s.d. The mean neural responses were combined into a firing rate matrix (but also see cross-validation section) with dimensions of Neurons × Time.

Neural data was pre-processed in the following way: first, to limit the dominant influence of high-firing units, we applied soft-normalization to each neuron's firing rate, such that the neurons with strong responses had close to unity range of responses $r^{/} = \frac{r}{7 + (\max(r) - \min(r))}$. The constant 7 was chosen as the roughly 20th percentile value of the firing rate range across all units. Second, the neural responses were mean-centred by subtracting the mean of each neuron's activity across time and the mean activity across all neurons at every time point.

**Definitions of movement and movement-null subspaces.** We used the approach first utilized in ref. 33. There, the authors formalized a method to find a linear mapping between low-dimensional representation of activity in PMd/M1 and the muscles EMG data, which defines a movement subspace. A null-space relative to that subspace forms an orthogonal set of dimensions which activity can occupy without directly affecting the movement execution. To extend this analysis on our data, we used combined recordings of orofacial motor and premotor nuclei (V, IRN, SPVI and SPVO) as a proxy for activity of orofacial muscles involved in execution of a lick. While recordings from GRN could have also been included into this group, we kept it separate to allow the population analysis to be applied to that region because (1) we had a large number of units recorded from that region alone; and (2) it was the only nucleus in medulla with above-chance number of TF-responsive units, warranting a separate analysis.

We considered a possible mapping onto the movement subspace for each brain region. Our rationale was the following: there exist several parallel neural pathways that can drive the activity of orofacial nuclei neurons–from primary motor cortex, basal ganglia, cerebellar or midbrain output regions[56,57,72]. Thus, the modes of activity within these regions that map onto the movement subspace may have a causal role for the execution of licks. In general, however, these signals can also be caused by movement afference that is broadcasted globally[3,5,9] (Fig. 1k,l). It is impossible to differentiate between these two possibilities from our data alone and thus the existence of mapping of activity onto the movement subspace does not necessarily imply that the brain region is causally involved in execution of the lick. With that said, we did not find a good mapping onto a movement subspace for most of the early visual areas, olfactory regions and hippocampal input regions (Extended Data Fig. 10a), suggesting that existence of mapping onto the movement subspace is not possible across all brain regions.

The mapping onto movement subspace was defined as:

$$\widetilde{M} = W\widetilde{N} \tag{1}$$

where $\widetilde{M}$ and $\widetilde{N}$ are low-dimensional representations of activity (projections onto main the principal components, the latter found via svd Matlab function) of neurons within the orofacial nuclei group and the target brain region, respectively, and $W$ is a linear mapping operator onto the movement subspace.

Before finding a linear mapping, we also zeroed the initial state across projections on principal components by subtracting from each projection the mean value within [−2, −1.5] s from lick onset. This step avoided the need for using intercept in the linear fit and simplified the visualization of projections on principal components and movement/movement-null dimensions. Linear mapping was found using only the time-period containing movement-related activity of orofacial nuclei [−0.1, 1.5] s around lick onset. This way we did not preclude the presence of preparatory activity on movement dimensions from the definition of the linear fit itself. A linear mapping to movement dimensions was found using linear regression with the Matlab function lsqnonlin.

Correspondingly, $W_{\text{null}}$ was a null-space of $W$ and was found using the Matlab function null. We used two top principal components of orofacial nuclei activity (which captured 61% of the total cross-validated variance; Extended Data Fig. 10a,b) and 4 top principal components of activity in a target brain region to find $W$ and $W_{\text{null}}$ operators (see Extended Data Fig. 10b–d). This choice resulted in both movement and movement-null subspaces being two-dimensional. We additionally ensured that norms of these operator are equal $\|W_{\text{null}}\| = \|W\|$ in order to make the comparison between the movement and movement-null subspaces fair.

Since the definition of specific dimensions in movement-null subspace is to a degree arbitrary, we defined the first movement-null dimension by finding a rotation within the movement-null subspace that maximized the amount of variance captured by that dimension prior to lick onset. The second movement-null dimension was then simply orthogonal to the first dimension in movement-null subspace. This was used mainly to simplify visualization, with all subspace-related analyses done using both dimensions in each subspace.

The positive direction of movement dimensions was chosen such that the mean value of projection of orofacial nuclei activity within [−2, 0.5] s around lick onset was positive. The positive direction for movement-null dimensions was chosen such that the mean value of projection of activity within [−2, 0] s around lick onset was positive.

**Subspace occupancy.** Relative subspace occupancy at a moment of time $t$ was defined as

$$O_{\text{R}}(t) = \frac{E_{\text{null}}(t) - E_{\text{m}}(t)}{E_{\text{null}}(t) + E_{\text{m}}(t)}$$

where $E_{\text{null}}(t)$ and $E_{\text{m}}(t)$ are Euclidean distances within movement-null and movement subspaces, measured between the neural state at the current moment of time t and the initial time point (the mean across 2 and 1.5 s before the lick onset). Values close to zero signify equal occupancy between subspaces and positive values indicate a preferential occupancy of the movement-null subspace. The peak-normalized occupancy (Extended Data Fig. 11a,b) was defined as $O(t) = \frac{E(t)}{\max(E)}$.

**Decomposition of projections onto contributions from TF-responsive and TF non-responsive units.** We decomposed the projections on main principal components into a sum of contributions from TF-responsive and the TF non-responsive units. For that, we used the knowledge of identity of each unit as TF-responsive or TF non-responsive and wrote down the principal components $U$ (from the singular value decomposition (SVD) of the firing rate matrix $N = USV^T$) as a sum of two parts as:

$$U = U_{\text{TF}}\begin{pmatrix} U_i, i \in \text{TF unit} \\ 0, i \in \text{non TF units} \end{pmatrix} + U_{\text{non TF}}\begin{pmatrix} 0, i \in \text{TF unit} \\ U_i, i \in \text{non TF units} \end{pmatrix} \tag{2}$$

where $U_i$ is a loading of the $i$th unit.

With that, projections on principal components can be written as:

$$\widetilde{N} = U^T N = U_{\text{TF}}^T N + U_{\text{non TF}}^T N = \widetilde{N}_{\text{TF}} + \widetilde{N}_{\text{non TF}} \tag{3}$$

Substituting equation (3) into equation (1) gives projections onto movement dimensions as:

$$\widetilde{N}^{\text{mov}} = W\widetilde{N} = \widetilde{N}_{\text{TF}}^{\text{mov}} + \widetilde{N}_{\text{non}}\,\text{TF}^{\text{mov}}$$

and, correspondingly, projections on movement-null dimensions are written as:

$$\widetilde{N}^{\text{null}} = W_{\text{null}}\widetilde{N} = \widetilde{N}_{\text{TF}}^{\text{null}} + \widetilde{N}_{\text{non}}\,\text{TF}^{\text{null}}$$

The relative contribution of TF-responsive units within movement and movement-null subspaces at the moment of time $t$ was then defined as following:

$$w_{\text{mov}}(t) = \frac{\widetilde{N}_{\text{TF}}^{\text{mov}}(t) \times \widetilde{N}^{\text{mov}}(t)}{\|\widetilde{N}^{\text{mov}}(t)\|^2} \times \frac{\widetilde{N}^{\text{mov}}(t)}{|\widetilde{N}^{\text{mov}}(t)|}$$

$$w_{\text{null}}(t) = \frac{\widetilde{N}_{\text{TF}}^{\text{null}}(t) \times \widetilde{N}^{\text{null}}(t)}{\|\widetilde{N}^{\text{null}}(t)\|^2} \times \frac{\widetilde{N}^{\text{null}}(t)}{|\widetilde{N}^{\text{null}}(t)|}$$

where the second multiplicative term ensures that the sign of contribution is relative to the defined positive direction (see above) of dimensions within each subspace.

In order to test whether the contribution of TF-responsive units is larger than what is expected from a uniform contribution of the full population, we repeatedly randomly selected (2,000 times) the same number of units as there were TF-responsive ones from the whole population and computed their contribution to projections on movement and movement-null dimensions as described above.

In addition to the analysis described above, we have also checked whether the above-chance contribution of TF-responsive units is a consequence of their level of activity, despite the normalization method that we used, or does it reflect a better correspondence of their activity to the population modes of activity within the movement-null subspace. For that we looked at the distribution of loadings along the first movement-null dimension–that captured the majority of preparatory activity there. We found that the majority of brain regions where TF-responsive units had above-chance contribution to the preparatory activity also had larger absolute values of loadings along that dimension than the rest of the population (Extended Data Fig. 11d,e).

**Cross-validation.** Since our analyses were focused on characterizing the mean neural responses, the cross-validation procedure that we used was designed to test the stability of the mean neural responses and their corresponding low-dimensional representations across trials. For that, we split trials into two randomly assigned and equally sized groups (fit and test trials) and calculated the mean neural response per unit across each group of trials. We next combined firing rates of neurons from the same brain region(s) (but not necessarily simultaneously recorded) into a joint matrix. After applying the pre-processing steps outlined above, we had two firing rate matrices from fit and test trials.

For cross-validated PCA (Extended Data Fig. 10a), we applied SVD on the first (fit) matrix and measured how well the remaining (test) matrix is predicted by the reconstruction from SVD components found from the first matrix. Similarly, the projections of activity on main principal components (Extended Data Fig. 13) were done using the test data, projected onto principal components found from the fit data.

For further analyses utilizing movement and movement-null subspaces, we applied SVD separately on each matrix and found their projections on first four main principal components. We then used low-dimensional representation of fit trials data to find linear mapping $W$ and $W_{\text{null}}$ onto the movement and moment-null subspaces. Finally, we applied $W$ and $W_{\text{null}}$ found from the fit data to the low-dimensional representation of the test data. This procedure was repeated 2,000 times, the 95% confidence intervals shown in Fig. 6 illustrate the 2.5 and 97.5 percentiles across projections of the test data. Because the sign of projection is arbitrary defined, we additionally applied a potential flipping of the sign of eigenvectors from each draw based on which direction had better alignment with the eigenvectors computed from the full firing rate matrix without the split into fit and test trials.

**Responses to TF pulses.** For each brain region, we constructed a firing rate matrix of all units responses to a fast TF pulse (or concatenating in time responses of each unit to different types of TF pulses for analysis shown in Fig. 6i,k–m), and used the same pre-processing steps as described above. The projections onto the movement and movement-null dimensions were done using loadings found from the analysis of hit licks activity described above (using the full firing rate matrix of hit-lick responses without the split into fit and test trials). Cross-validation of consistency of projections was done by randomly selecting half of TF outlier events, computing the mean firing rate across those events for each unit, applying the steps above to find the projections, and repeating this procedure 2,000 times. For analyses where different brain regions were combined into a common group, all units from those brain regions were combined into a joined firing rate matrix and the steps described above were applied.

Alignment of fast TF pulse response with a given dimension in movement or movement-null subspace was calculated as a cosine of an angle between the projection onto a target dimension and a 4-dimensional vector of TF pulse response (2 movement and 2 movement-null dimensions) at a time of the maximum Euclidean distance from the initial state across 4 dimensions within a 0.75-s window from the pulse onset. Similarly, for calculating the scaling of responses to different TF pulses along the first movement-null dimension, we found the sizes of projections at times of maximal Euclidean distance from the initial state within a 0.75-s window from the first TF pulse onset.

### Reporting summary

Further information on research design is available in the Nature Portfolio Reporting Summary linked to this article.

### Data availability

The data that support the findings of this study are available from the corresponding authors upon reasonable request. Source data are provided with this paper.

### Code availability

Custom acquisition, post-processing and analysis code is available at https://github.com/BaselLaserMouse/Khilkevich_Lohse_2024.

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

**Acknowledgements** The authors thank M. Hamada, D. Gupta and S. Hofer for comments on the manuscript; M. Sahani for helpful discussions regarding population analyses; R. Campbell for help with histology, microscopy and image processing; M. Faulkner for providing access to ephys-atlas alignment tools implemented by the International Brain Laboratory; the staff at the SWC Neurobiological Research Facility for mouse husbandry; the SWC Fabrication Laboratory for help with machining; M. Skretowska for help with training mice; members of the

Mrsic-Flogel, Hofer and other SWC laboratories for insightful discussion and advice. This work was supported by Wellcome awards to T.D.M.-F. (217211/Z/19/Z) and M.L. (224121/Z/21/Z), by the Sainsbury Wellcome Centre's core provided by Wellcome (219627/Z/19/Z) and the Gatsby Charitable Foundation (GAT3755).

**Author contributions** A.K., M.L., I.O. and T.D.M.-F. conceived the project. A.K., M.L. and T.D.M.-F. designed the experiments. A.K. and M.L. collected and analysed the data with supervision from T.D.M.-F. T.B. trained the mice with guidance from A.K., M.L. and I.O. R.L. developed the outlier detection agent for decision-making. P.W. manually curated spike-sorted extracellular data with guidance from A.K. and M.L. A.K., M.L. and T.D.M.-F. wrote the paper with input from all authors.

**Competing interests** The authors declare no competing interests

**Additional information**
**Correspondence and requests for materials** should be addressed to Andrei Khilkevich, Michael Lohse or Thomas D. Mrsic-Flogel.

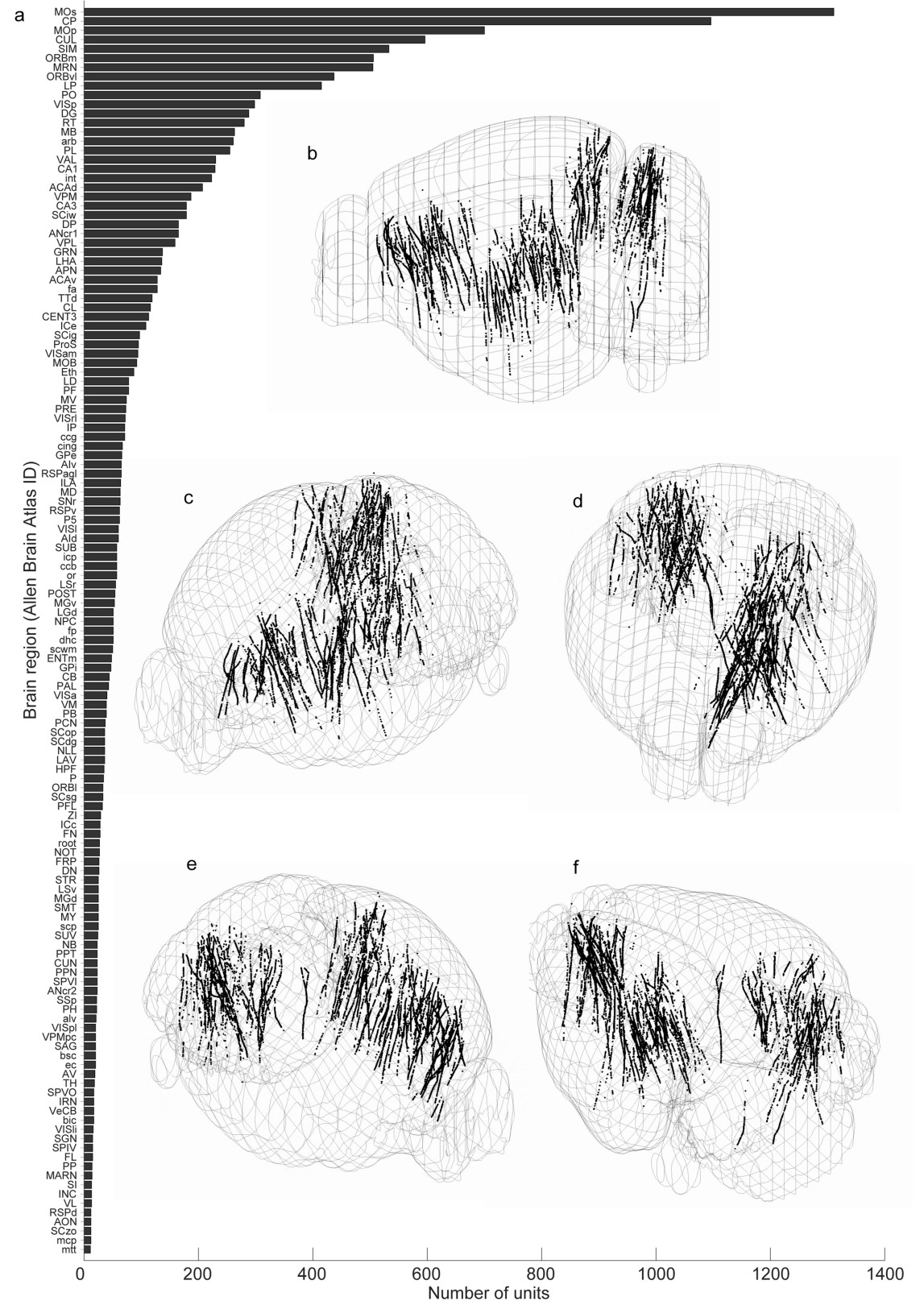

**Extended Data Fig. 1 | Summary of recordings in trained mice. a**, Number of cells recorded from trained mice in each Allen Brain Atlas designated region. **b-f**, Locations of all well-isolated and stable units, shown within a 3D rendering of Allen Common Coordinate Framework from five perspectives.

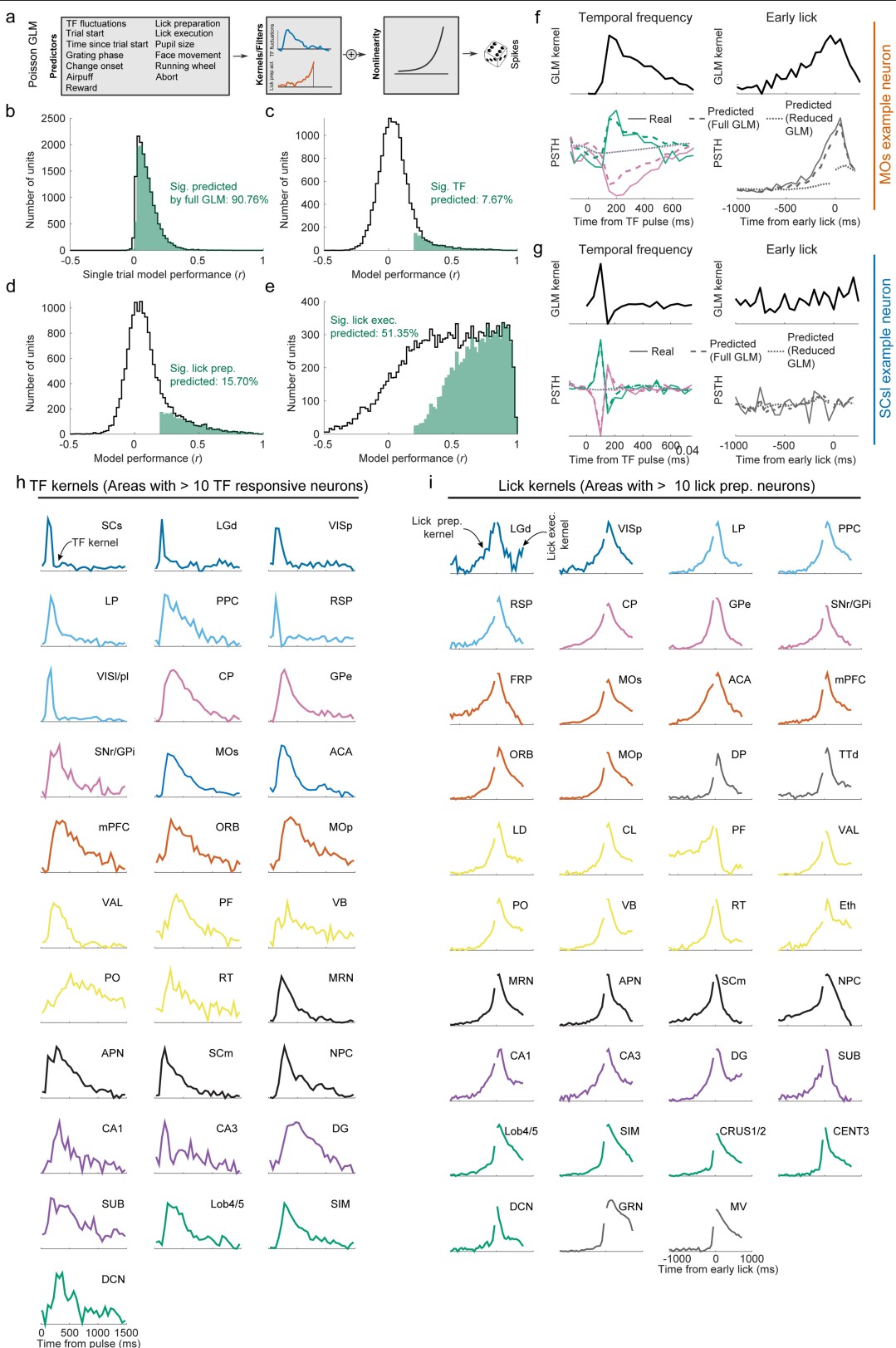

**Extended Data Fig. 2 | GLM Performance. a**, Schematic of Poisson GLM. **b**, Cross-validated model prediction performance of single trial spike counts with full GLM model (*r*). **c**, Cross-validated model prediction performance of mean PSTH following fast and slow pulses (*r*). **d**, Cross-validated model prediction performance of mean PSTH leading up to an early lick (Lick preparation) (*r*). **e**, Cross-validated model prediction performance of mean PSTH after early lick (Lick execution) (*r*). **f**, GLM predictions on example neuron recorded in MOs. *Top:* GLM kernels which the predictions are made from. *Bottom:* Real vs full GLM predicted vs reduced GLM (without key predictor in model) PSTHs.

**g**, GLM predictions on example neuron recorded in SCs. *Top:* GLM kernels which the predictions are made from. *Bottom:* Real vs full GLM predicted vs reduced GLM (without key predictor in model) PSTHs. **h**, Mean TF kernels across all areas with 10 or more TF-responsive units recorded (for averaging kernels are flipped when needed to always have a positive response). **i**, Mean lick preparation and lick execution kernels across all areas with 10 or more lick preparation neurons responsive units recorded (for averaging kernels are flipped when needed to always have a positive response).

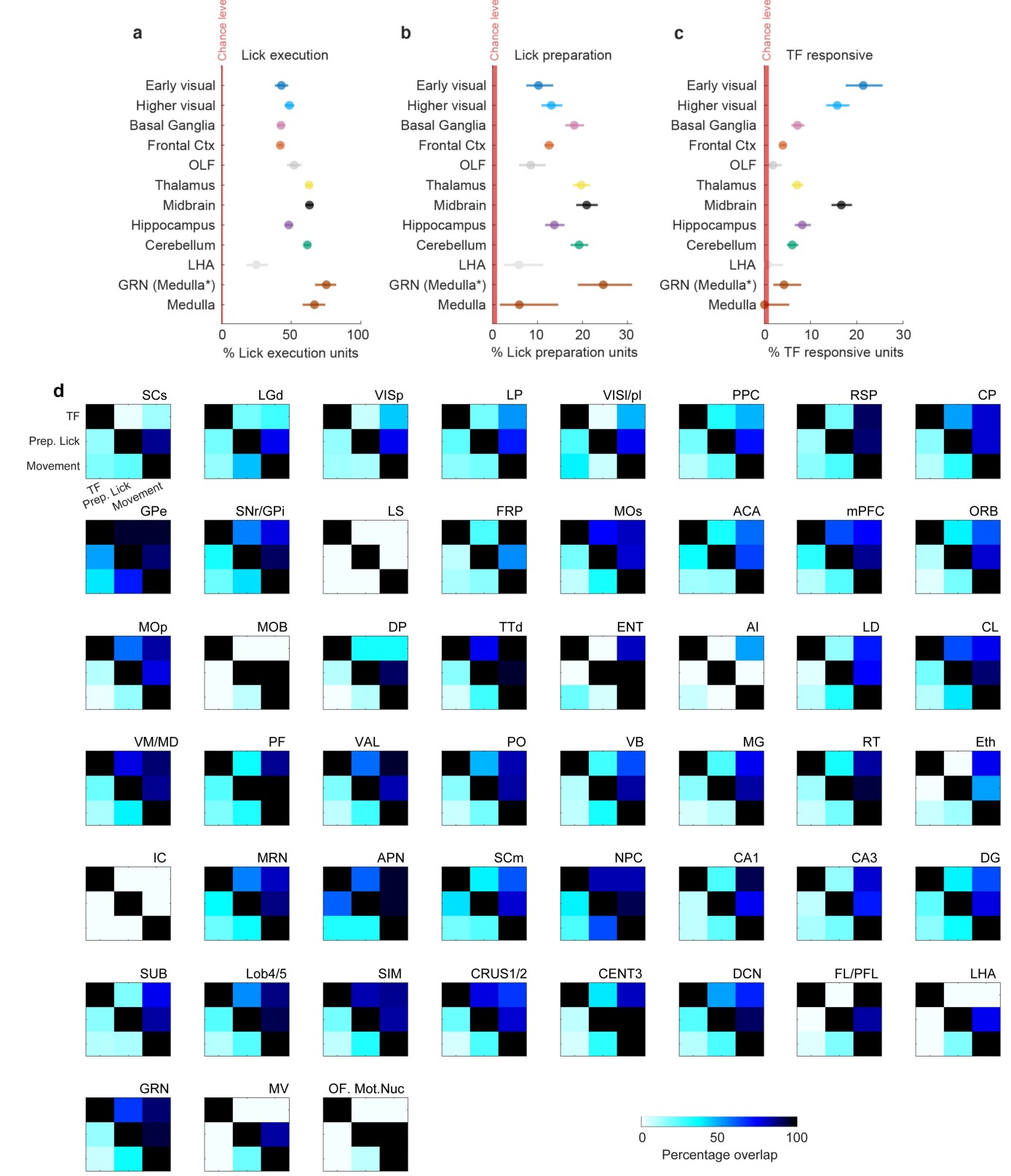

**Extended Data Fig. 3 |** See next page for caption.

**Extended Data Fig. 3 | Encoding of temporal frequency fluctuations, lick preparation and lick execution across brain areas. a-c**, Percentage of units encoding temporal frequency fluctuations during baseline, lick preparation, or lick execution in major area groupings with 95% binomial confidence intervals. **a**, Percentage lick execution units: All areas: $p < 0.001$ (Binomial test). **b**, Percentage lick preparation units: Early visual, Higher visual, Basal ganglia, Frontal cortex, Olfactory nuclei (OLF), Thalamus, Midbrain, Hippocampus, Cerebellum, Lateral hypothalamus (LHA), GRN (Medulla*), Medulla: $p < 0.001$ (Binomial test), Medulla: $p < 0.01$ (Binomial test). **c**, Percentage TF Responsive units: Early visual, Higher visual, Basal ganglia, Frontal cortex, Thalamus, Midbrain, Hippocampus, Cerebellum, GRN (Medulla*): $p < 0.001$ (Binomial test), Olfactory nuclei (OLF), Lateral hypothalamus (LHA), and Medulla: $p > 0.05$ (Binomial test). Error bars in panels a-c are 95% binomial confidence intervals. Red areas designate chance level. See Supplementary Table 1 for $n$ of each brain area grouping. **d**, Percentage overlap of encoding (estimated from GLM) of TF, lick preparation, and lick execution, in all areas with more than 40 units recorded. y-axis is the source population (i.e., all TF responsive neurons, all lick preparation neurons, or all lick execution neurons).

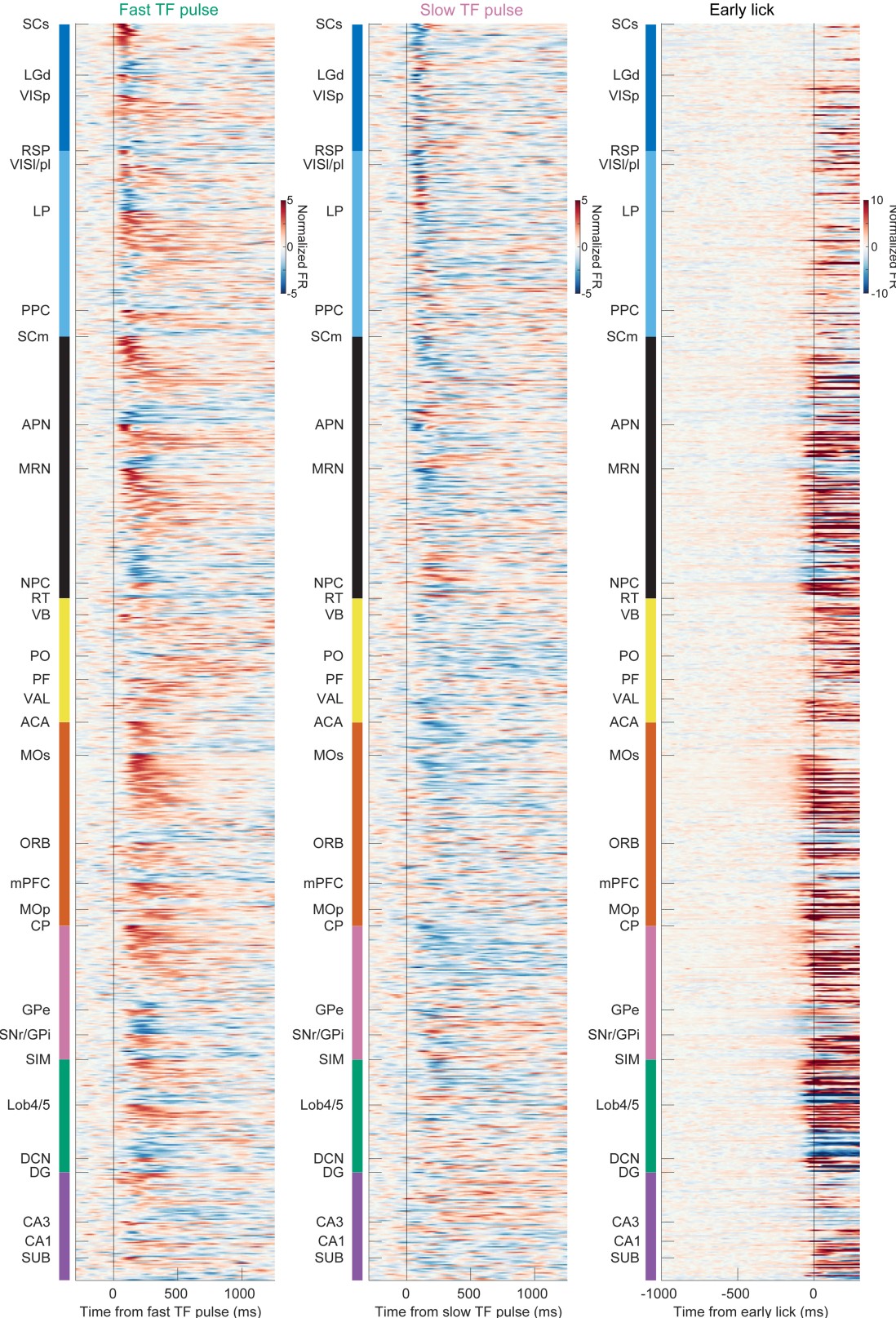

**Extended Data Fig. 4 | Responses of TF responsive neurons across the brain to fast or slow TF pulses and early licks.** Activity (z-scored) of individual neurons around fast TF pulses (*left*), slow TF pulses (*middle*) and early licks (*right*) for all TF responsive units from all areas with 10 or more TF responsive units recorded. Major subdivisions of the brain grouped by colour. Each line represents one neuron.

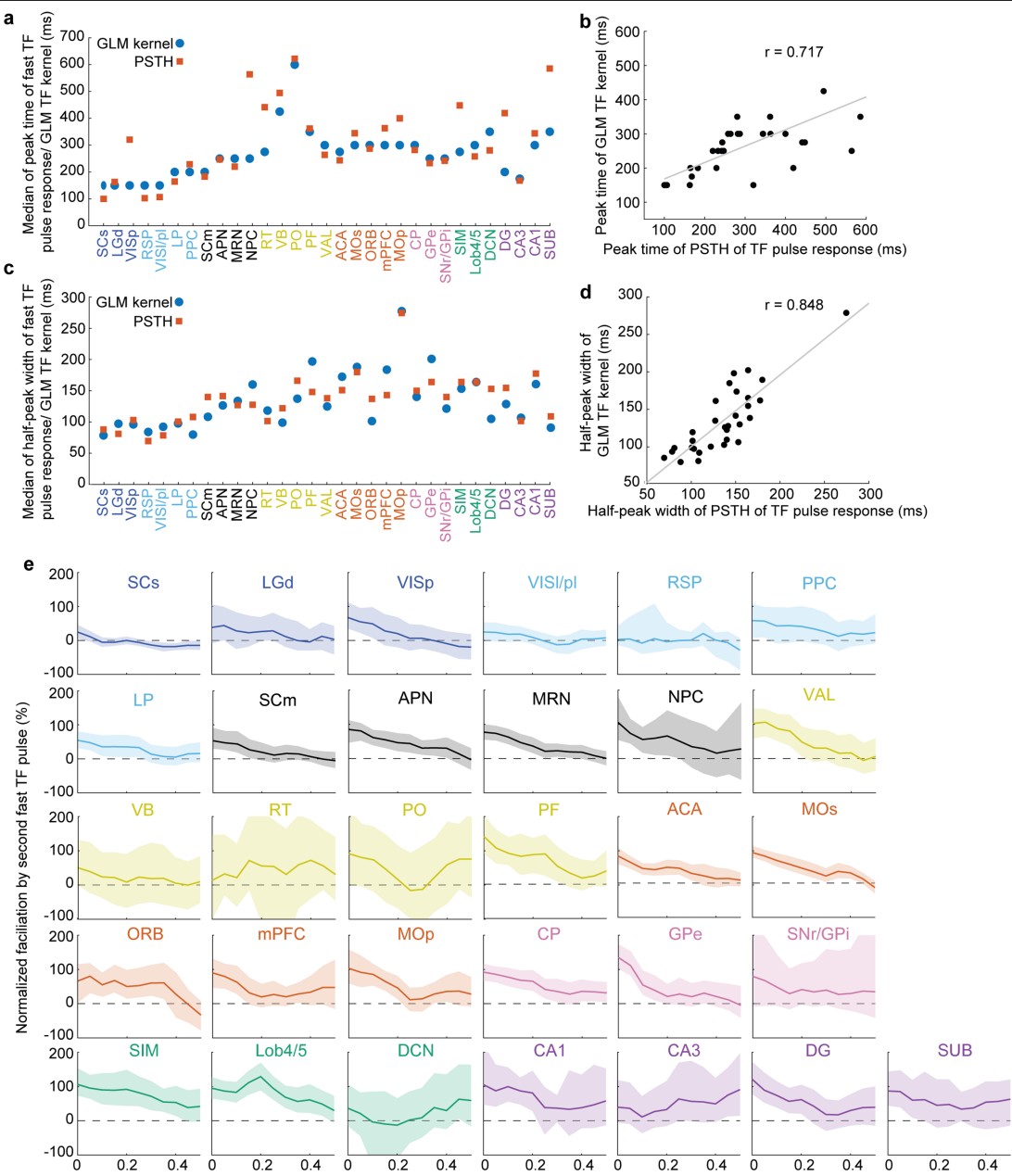

**Extended Data Fig. 5 | Properties of responses to a single fast TF pulse from PSTHs and GLM + Relative facilitation by the second fast TF pulse as a function of delay from the first one. a-d**, Comparison of peak time and response width of PSTHs following a fast TF pulse vs GLM TF kernels. **a**, Median peak time of response to a fast TF pulse estimated from PSTH (red) and median peak time of GLM TF kernel (blue), shown for each brain region. **b**, Correlation across brain regions between median peak time estimated from PSTH and median peak time of GLM TF kernel. **c-d**, Same as a-b, but for fast TF pulse response half-peak width. **e**, Relative facilitation by the second fast TF pulse, normalized by the response to a single fast TF pulse, shown as a function of delay between two fast TF pulses for each brain region with at least 10 TF responsive units (mean and 95% confidence intervals, bootstrap test (see Methods)). Values close to zero imply no facilitation (same size of response to the second fast TF pulse as to the first one), while values close to 100% imply doubling of the response size.

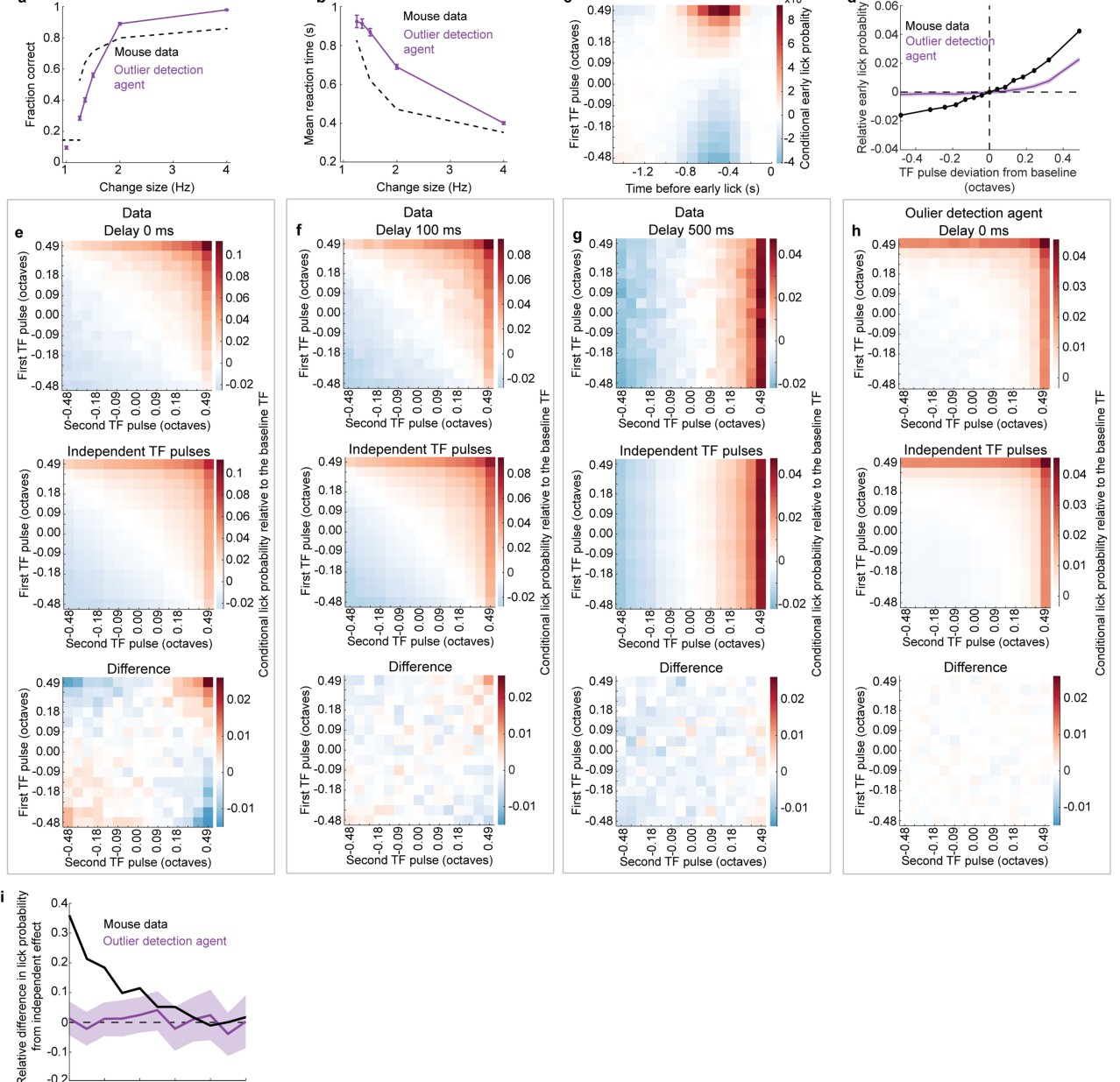

**Extended Data Fig. 6 | Effect of magnitude and timing of TF pulses on probability of early licks. a**, Mean performance (psychometric curves) for mice data (dashed black line, n = 15 mice) and outlier detection agent (purple). **b**, Mean reaction times per change magnitude for outlier detection agent (purple) and mice data (dashed black line, n = 15 mice). Error bars indicate 95% confidence intervals across 4000 synthetic datasets of the model (see Methods). **c**, Conditional probability of early lick at a specific time after a TF pulse of given magnitude. Here and later early lick probability is shown relative to the probability at the mean baseline TF (1 Hz). **d**, Probability of early lick after a TF pulse of given magnitude (here and later cumulatively within [0.2, 1] s window). Mice data is shown in black, outlier detection agent – in purple (mean and non-parametric 95% confidence intervals, see Methods).

**e**, Upper panel: probability of early lick after two sequential TF pulses of given magnitudes; middle panel: expected effect if both pulses influence early lick probability independently; lower panel: difference from the independent effect of TF pulses. **f-g**, The same format as in **e**, but for two TF pulses with 100 ms or 500 ms delay between them. **h**, The same format as in **c**, but shown for data generated by the outlier detection agent (for two sequential TF pulses). **i**, Difference in probability of early lick relative to the independent effect after a sequence of two fast TF pulses (top right corner in lower panels **e-g**), normalized by the expected probability from the effect of independent pulses and shown as a function of delay between fast TF pulses. The results of the same analysis applied to the outlier detection agent data are shown in purple (mean and non-parametric 95% confidence intervals, see Methods).

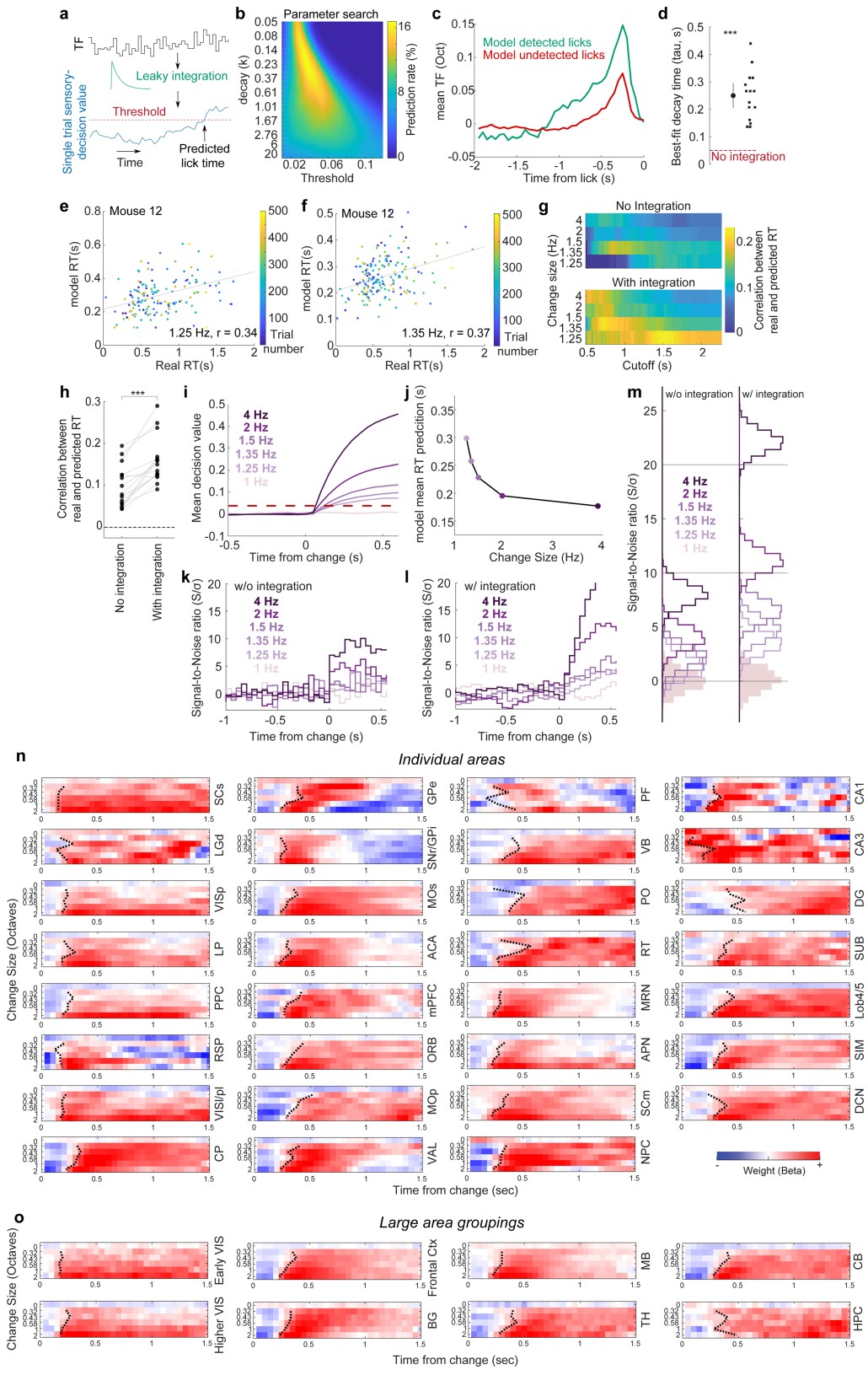

**Extended Data Fig. 7** | See next page for caption.

**Extended Data Fig. 7 | A simple two parameter leaky integrator model supports behavioural evidence integration + GLM change kernels across individual areas and large area groupings. a**, Schematic of the leaky-integrator model. **b**, Parameter search grid identifying which values the integration time and threshold best predicts early licks (i.e., correct predictions of early lick times (on single trials). **c**, Lick triggered stimulus average of early licks detected by the leaky integrator model, and early licks not detected by the model. **d**, Best-fit integration decay time of leaky-integrator model, shown per mouse (black dots) and mean across animals ($n = 15$ mice, error bar is 95% confidence intervals). ***$p < 0.001$, two sided $t$-test. **e**, Relationship between real reaction time and predicted reaction time from leaky integrator model (tau: 0.25 s) for change size 1.25 Hz of example mouse 12. Correlation is calculated across all reaction times. **f**, Same as **f** but for change size 1.35 Hz. **g**, Correlation between observed and predicted reaction times during the change period for outlier detection agent (no integration, *top*) and leaky-integrator model (*bottom*). Threshold parameters corresponding to best-fit were used for each model. The colour along each row corresponds to the correlation value between predicted hit lick reaction times and actual hit lick reaction times on trials with that change magnitude, conditioned by the maximum RT included for this analysis (cutoff time). **h**, Summary of panel **g** with results shown per mouse and RT combined across all change magnitudes (RT cutoff equal to 1 second from change onset). $n = 15$ mice, ***$p < 0.001$, two sided $t$-test. **i**, Mean decision value (integrated TF) after filtering stimulus though a leaky integrator model with a tau of 0.25 s. **j**, Mean reaction time curve for leaky integrator model. **k**, Example trials around change onset when model has no integration. Note the similarity to change kernels of TF responsive units in the SCs in Fig 3l. **l**, Example trials around change onset when model has leaky integration (0.25 s tau). Note the similarity to change kernels of to TF responsive units in the MOs in Fig 3l. **m**, Leaky evidence integration smooths and denoises the noisy sensory input so that the signal-to-noise ratio (S/σ) is considerably larger 0.5 s after change onset, compared to no integration-- making detection of noisy changes easier. **n**, Change size specific GLM change kernels for all areas recorded with 10 or more TF responsive units. **o**, Change size specific change kernels for major area groupings. Dotted line indicates the 50% response crossing for each change size.

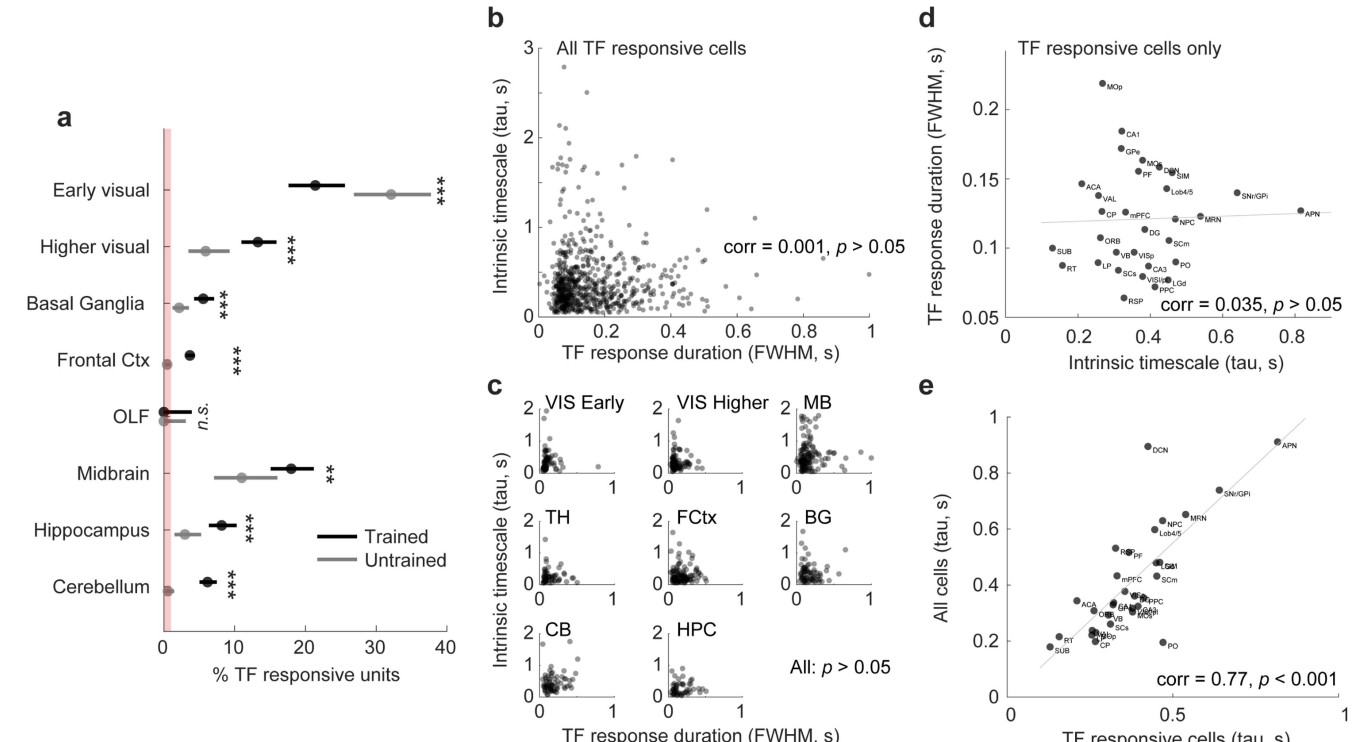

**Extended Data Fig. 8 | Intrinsic vs learned TF pulse response properties.**
**a**, Percentage of units encoding temporal frequency fluctuation during baseline in major area groupings with 95% binomial confidences in untrained and trained mice. Stars designate significance of difference (binomial test) in fractions between naïve and trained mice: *n.s.:* Not significant, ** $p < 0.01$, ***$p < 0.001$, binomial tests. Error bars are 95% binomial confidence intervals. OLF: Olfactory nuclei, Ctx: Cortex. See Supplementary Table 1 for $n$ of each brain area grouping. **b**, Intrinsic timescales (tau) estimated for each TF responsive unit across the brain vs the TF response width for those units. Intrinsic times scales do not correlate with TF response width at a single cell

level ($p > 0.05$, Pearson correlation, p-value is based on t-statistic). **c**, Same as in a but with units divided into major area groups. No area group has significant correlation between intrinsic times scales and TF response width at a single cell level ($p > 0.05$, Pearson correlation, p-value is based on t-statistic). **d**, Same as Fig. 4g, but here areal intrinsic time scale is extracted from TF responsive units only. In agreement with Fig. 4g, there is no correlation (Pearson correlation, p-value is based on t-statistic) between areal intrinsic timescales and median TF response width. **e**, intrinsic timescales of TF responsive units are similar to the intrinsic timescales as areas as a whole (Pearson correlation, p-value is based on t-statistic).

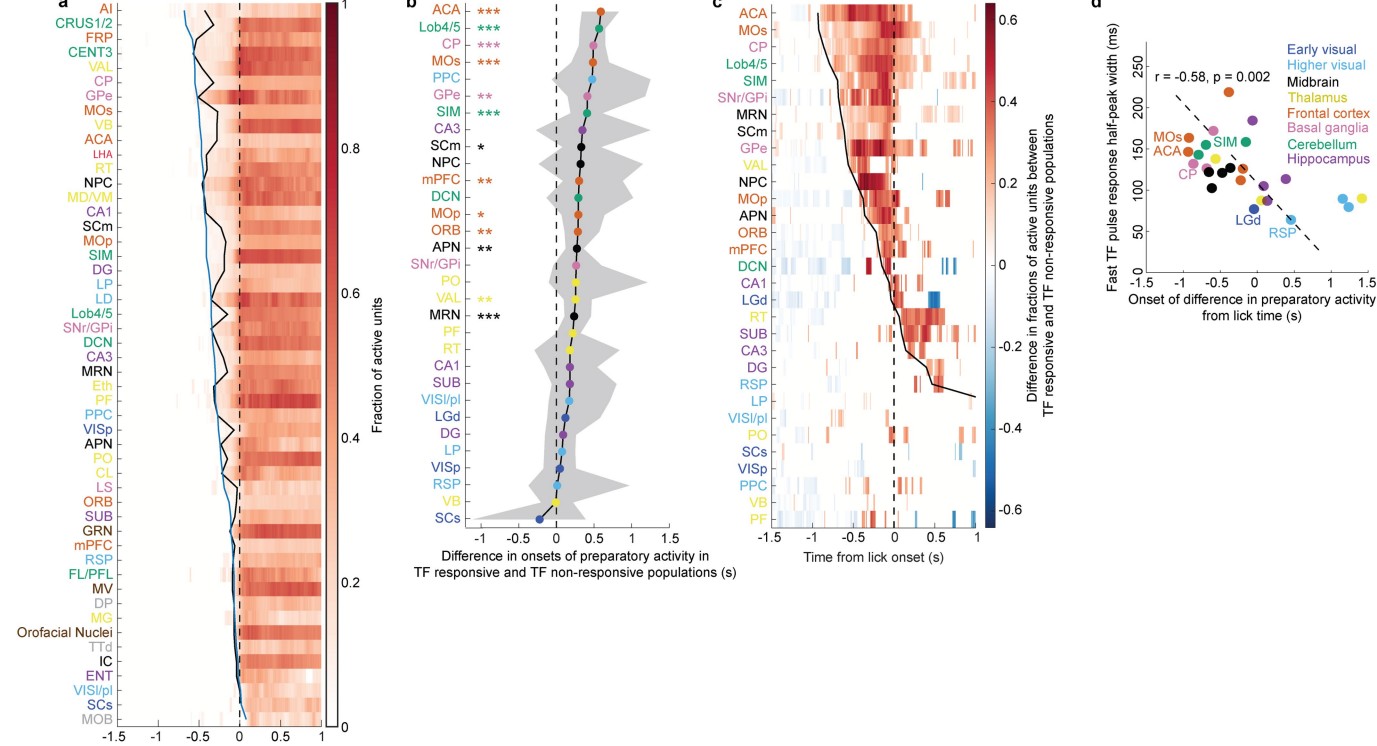

**Extended Data Fig. 9 | Differences in timing of preparatory activity between TF responsive and TF non-responsive populations. a**, Fraction of active units (combined across TF responsive and TF non-responsive units) as a function of time from the hit lick onset, shown across brain regions. Shades of red indicate a higher level of activity. Time points with lower 95% confidence interval (bootstrap test, see Methods) smaller than zero are shown as white. Brain regions are sorted according to the time of the first significant activation (blue line, see Methods). Black line shows the time of first significant activation using the same criterion as for Fig. 5f,g. **b**, Difference in onsets of preparatory activity across TF responsive and TF non-responsive subpopulations. Positive values indicate that TF responsive subpopulation has an earlier preparatory activity. Significant differences from zero are indicated by number of stars

and area shaded in grey indicates 95% confidence intervals (bootstrap test, see Methods). * $p < 0.05$, ** $p < 0.01$, *** $p < 0.001$. **c**, Difference in levels of activity between TF responsive and TF non-responsive subpopulations within each brain region. Shades of red indicate a higher level of activity across TF responsive subpopulation. Time points with non-significant differences ($p \geq 0.05$, bootstrap test) in activity are shown as white. Brain regions are sorted according to the latency of the first significant difference in activation between TF responsive and non-responsive subpopulations (black line). **d**, Pearson correlation (p-value is based on t-statistic) across brain regions between the latency of the first significant difference in activation between TF responsive and TF non-responsive subpopulations and the median half-peak width of response to a fast pulse.

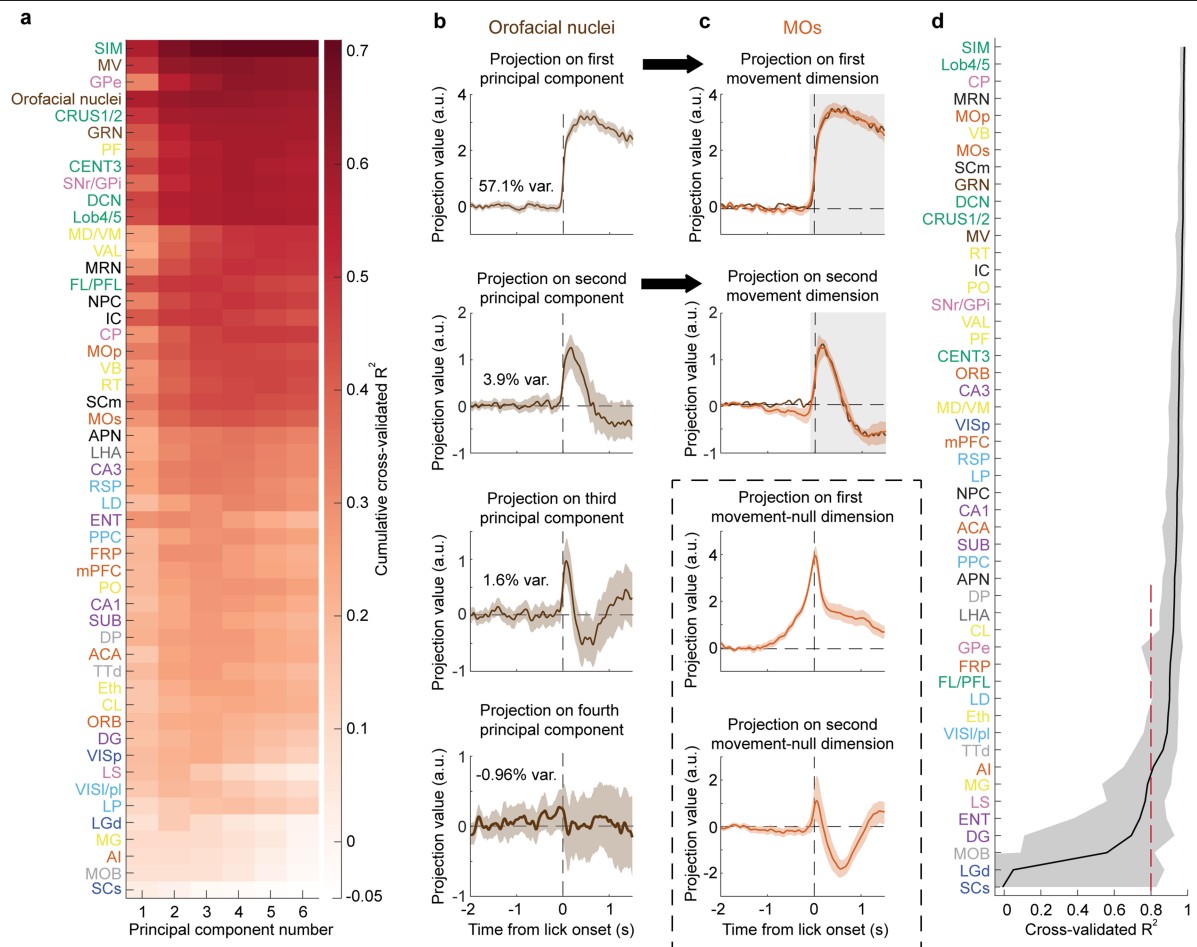

**Extended Data Fig. 10 | Definition of movement and movement-null subspaces. a**, Cross-validated cumulative R-squared coefficient of activity aligned to the hit lick onset shown across first six principal components for each brain region. Brain regions are sorted by the maximum cumulative R-squared value. **b**, Projections onto first four principal components of orofacial nuclei activity aligned to the hit lick onset. Projections on the first two principal components define the temporal profiles of activity within the two-dimensional movement subspace. The amount of cross-validated variance (average across draws) captured by each principal component is indicated on each panel. **c**, Projections of MOs activity (orange) aligned to the hit lick onset onto two movement (top) and two movement-null (bottom) dimensions. Projections of orofacial nuclei activity onto movement dimensions are shown in brown. **d**, Average cross-validated R-squared coefficient of mapping onto the movement subspace, with brain regions ordered from the best to worst mapping accuracy. The minimal value of R-squared coefficient for a brain region to be considered to have a good mapping onto a movement subspace is shown as a dashed red line (0.8). In all panels shaded regions indicate non-parametric 95% confidence intervals (see Methods).

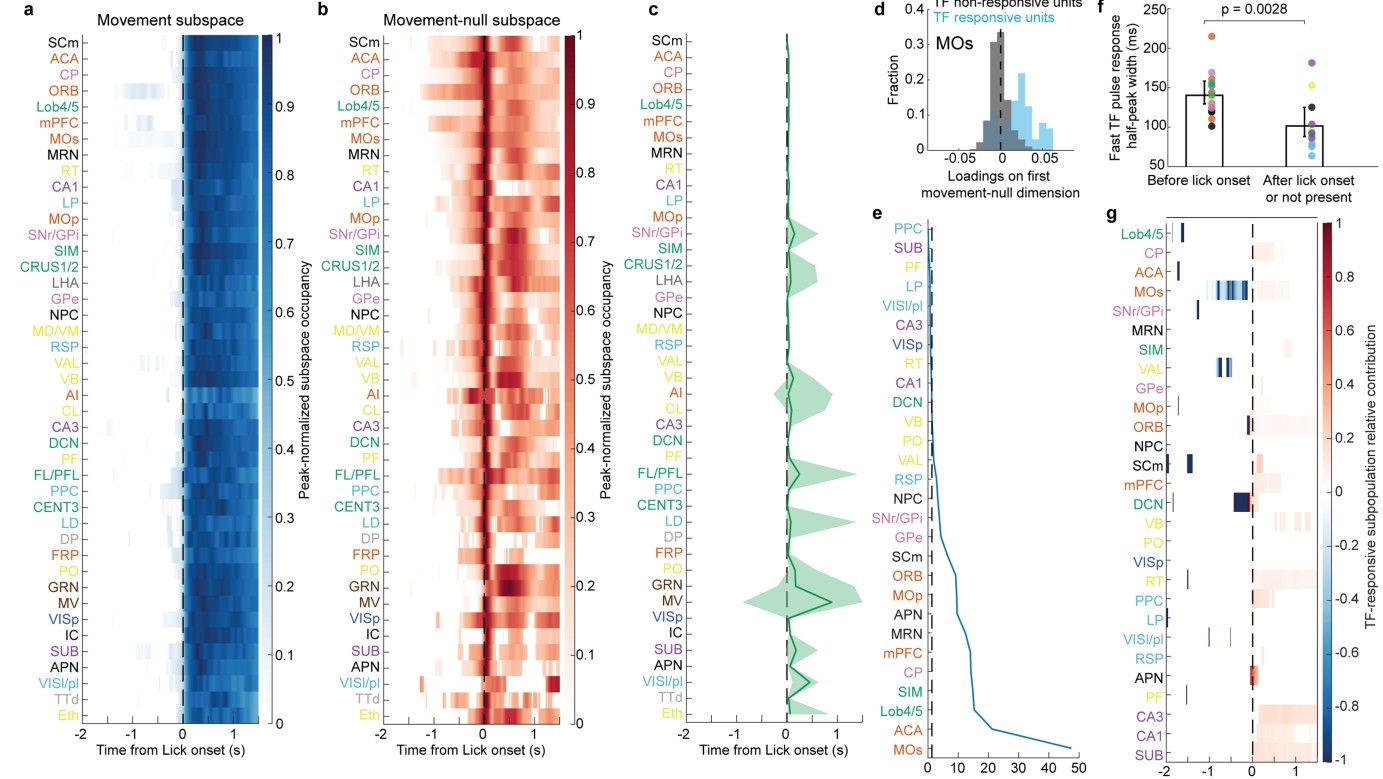

**Extended Data Fig. 11 | Occupancy of movement and movement-null subspaces and contribution of TF-responsive subpopulation within them.** **a**, Peak-normalized occupancy of movement subspace as a function of time for each brain region, relative to the hit lick onset time. Here and on panels b,c the order of brain regions is the same as on Fig. 6f. **b**, Peak-normalized occupancy of movement-null subspace as a function of time for each brain region. **c**, Average time of the peak occupancy within the movement-null subspace (green line), shown for each brain region. Shading indicates 95% confidence intervals. **d**, Distribution of loadings values along the first movement-null dimension that correspond to TF responsive (blue) and TF non-responsive (black) units in MOs. **e**, Minus log of p-value (blue line) for a paired 2-sided t-test between

absolute values of loadings along the first movement-null dimension that correspond to TF responsive and TF non-responsive units. Dashed grey line indicates p = 0.05 level. **c**, Related to Fig. 6h. Comparison (Wilcoxon signed-rank test) of half-peak width of response to fast TF pulse between brain regions that had a disproportionate contribution of TF responsive subpopulation to preparatory activity in movement-null subspace (left bar, n = 16 brain regions) and the rest of brain regions (right bar, n = 12 brain regions). Bars indicate the mean across brain regions, error bars – 95% confidence intervals of the mean (bootstrap test, 2000 times). **f**, Relative contribution of TF responsive subpopulation within the movement subspace as a function of time for each brain region. Brain regions are shown in the same order as on Fig. 6h.

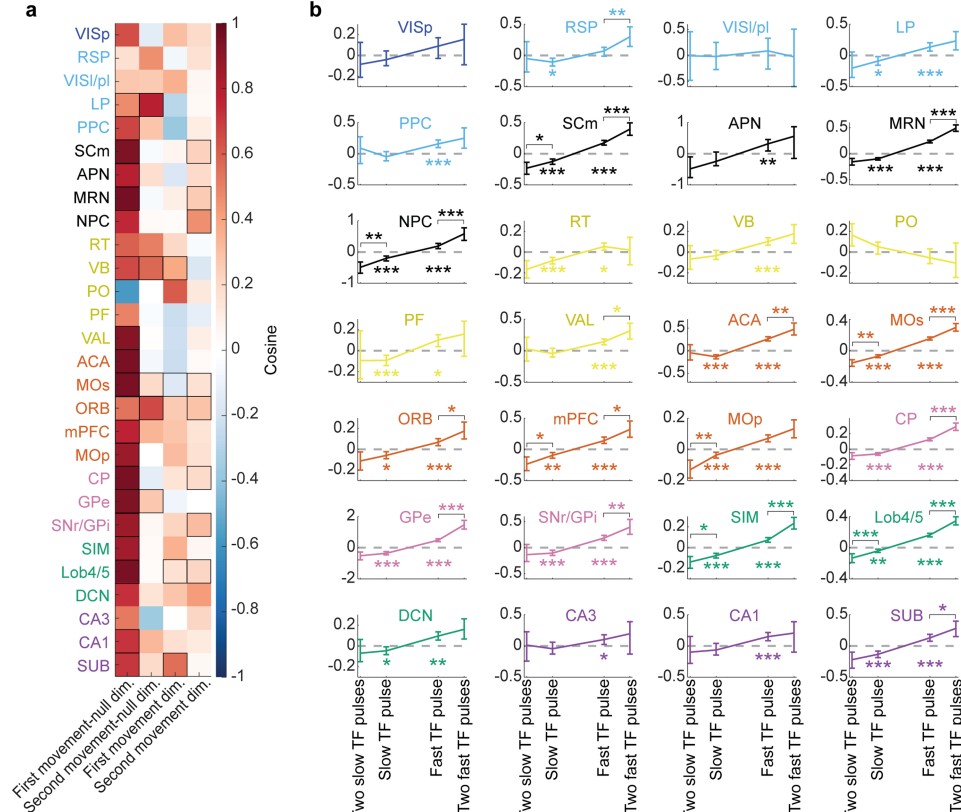

**Extended Data Fig. 12 | Alignment and scaling of TF pulse response projections on movement and movement-null dimensions. a**, Cosine of the angle between population response to a fast TF pulse and movement or movement-null dimensions. Values that are significantly different from zero (p < 0.05, 2-sided bootstrap test) are indicated by the black outline. **b**, Peak value of projections onto the first movement-null dimension of responses to a slow, fast, two sequential slow, and two sequential fast TF pulses. Results are

shown for each brain region as the mean and 95% confidence intervals over 2000 cross-validations (see Methods). See Supplementary Table 1 for number of neurons in each brain region. Number of starts indicates a 2-sided bootstrap test p-value of difference from zero for population response to a single fast or slow TF pulse, or a significance of a difference between responses to one or two sequential TF pulses. * p < 0.05, ** p < 0.01, *** p < 0.001. Non-significant effects are not indicated.

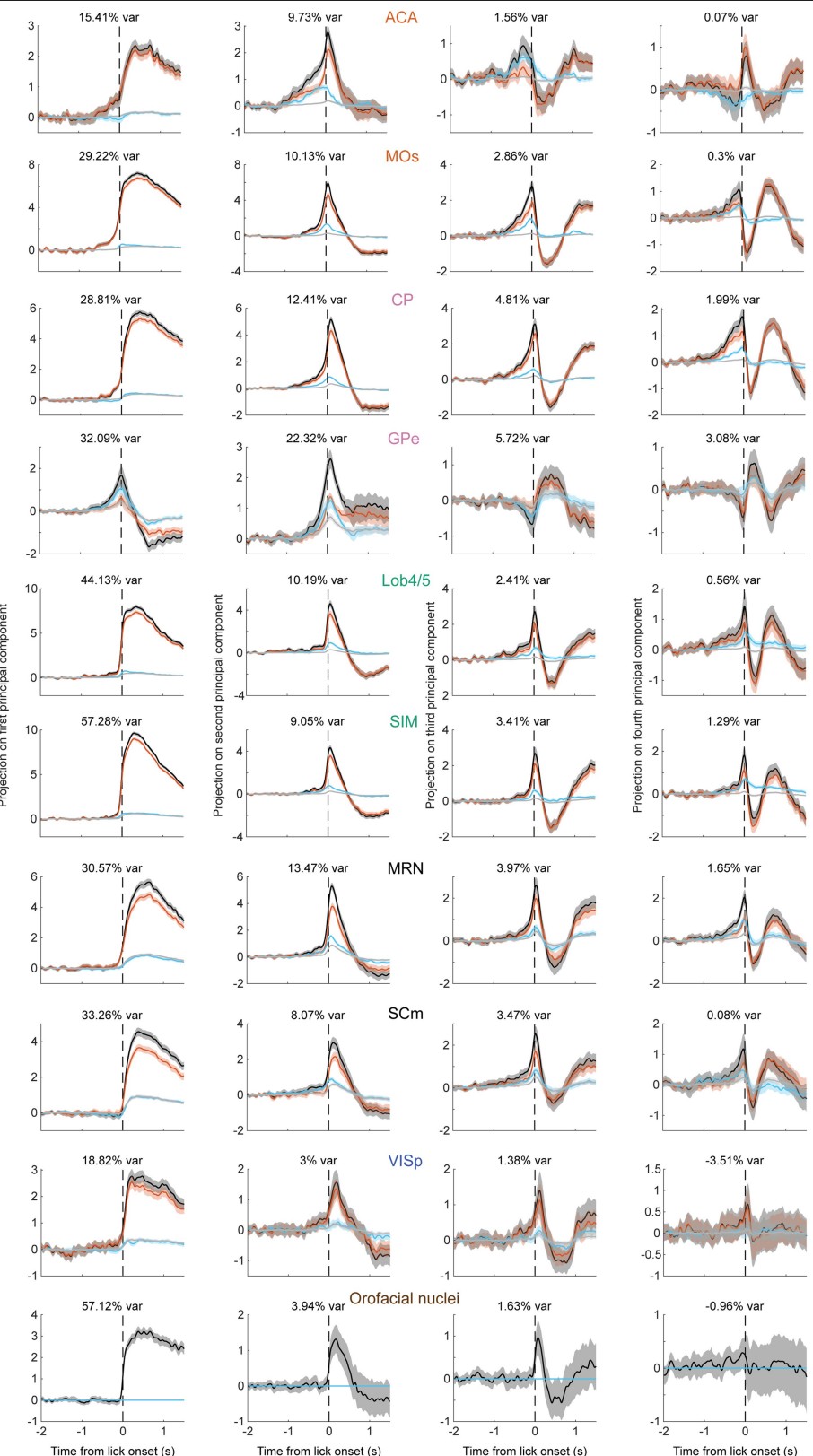

**Extended Data Fig. 13 | Breakdown of projections on main principal components by contributions from TF responsive and TF-nonresponsive units.** Each row shows projections on four main principal components of population activity within a given brain region aligned to the hit lick onset (same as on Fig. 6). The time course of each projection (black) was decomposed into a sum of contributions from TF responsive (blue) and TF non-responsive (red) units. Grey line indicates projection expected from a random sample of the same size as there were TF responsive units, taken randomly (with replacement) from the full population. Data is shown as mean and 95% confidence intervals across 2000 cross-validations (see Methods). The amount of cross-validated variance captured by each principal component is indicated on top of each panel.

# Reporting Summary

## Statistics

For all statistical analyses, confirm that the following items are present in the figure legend, table legend, main text, or Methods section.

| n/a | Confirmed | |
|---|---|---|
| ☐ | ☒ | The exact sample size (*n*) for each experimental group/condition, given as a discrete number and unit of measurement |
| ☐ | ☒ | A statement on whether measurements were taken from distinct samples or whether the same sample was measured repeatedly |
| ☐ | ☒ | The statistical test(s) used AND whether they are one- or two-sided *Only common tests should be described solely by name; describe more complex techniques in the Methods section.* |
| ☐ | ☒ | A description of all covariates tested |
| ☐ | ☒ | A description of any assumptions or corrections, such as tests of normality and adjustment for multiple comparisons |
| ☐ | ☒ | A full description of the statistical parameters including central tendency (e.g. means) or other basic estimates (e.g. regression coefficient) AND variation (e.g. standard deviation) or associated estimates of uncertainty (e.g. confidence intervals) |
| ☐ | ☒ | For null hypothesis testing, the test statistic (e.g. *F*, *t*, *r*) with confidence intervals, effect sizes, degrees of freedom and *P* value noted *Give P values as exact values whenever suitable.* |
| ☒ | ☐ | For Bayesian analysis, information on the choice of priors and Markov chain Monte Carlo settings |
| ☒ | ☐ | For hierarchical and complex designs, identification of the appropriate level for tests and full reporting of outcomes |
| ☐ | ☒ | Estimates of effect sizes (e.g. Cohen's *d*, Pearson's *r*), indicating how they were calculated |

*Our web collection on statistics for biologists contains articles on many of the points above.*

## Software and code

Policy information about availability of computer code

Data collection: SpikeGLX (v20221212-phase30) was used for extracellular recordings as well as for recordings of behavioral signals and timing of behavioral events. Video acquisition was done via-custom made Matlab (2021a) scripts available at https://github.com/BaselLaserMouse/Khilkevich_Lohse_2024 link

Data analysis: Spike-sorting was done with Kilosort 2.0 and units classified as "good" were further either manually curated using Phy2.0 (for trained mice dataset) or with custom filtering steps that were designed to approximate the results of manual curation (on naïve mice dataset). Pupil size was estimated using DeepLabCut. Registration of 3d stack of histological images to the standardized Allen Common Coordinate Framework (Allen CCF) was done with BrainRegister (https://github.com/stevenjwest/brainregister). Neuropixels probe tracts were manually traced using custom software (Lasagna, https://github.com/SainsburyWellcomeCentre/lasagna). Where needed, we manually adjusted the scaling of brain regions along the probe track to align responses on channels with features associated with anatomical locations using custom software (Ephys alignment tool, https://github.com/int-brain-lab/iblapps/tree/master/atlaselectrophysiology

For manuscripts utilizing custom algorithms or software that are central to the research but not yet described in published literature, software must be made available to editors and reviewers. We strongly encourage code deposition in a community repository (e.g. GitHub). See the Nature Portfolio guidelines for submitting code & software for further information.

## Data

Policy information about availability of data

All manuscripts must include a data availability statement. This statement should provide the following information, where applicable:
- Accession codes, unique identifiers, or web links for publicly available datasets
- A description of any restrictions on data availability
- For clinical datasets or third party data, please ensure that the statement adheres to our policy

Datasets used: Allen Common Coordinate Framework Atlas. Due to large size and complexity of the data, it will be available upon request from corresponding authors.

## Research involving human participants, their data, or biological material

Policy information about studies with human participants or human data. See also policy information about sex, gender (identity/presentation), and sexual orientation and race, ethnicity and racism.

| | |
|---|---|
| Reporting on sex and gender | N/A |
| Reporting on race, ethnicity, or other socially relevant groupings | N/A |
| Population characteristics | N/A |
| Recruitment | N/A |
| Ethics oversight | N/A |

Note that full information on the approval of the study protocol must also be provided in the manuscript.

# Field-specific reporting

Please select the one below that is the best fit for your research. If you are not sure, read the appropriate sections before making your selection.

☒ Life sciences        ☐ Behavioural & social sciences        ☐ Ecological, evolutionary & environmental sciences

For a reference copy of the document with all sections, see nature.com/documents/nr-reporting-summary-flat.pdf

# Life sciences study design

All studies must disclose on these points even when the disclosure is negative.

| | |
|---|---|
| Sample size | Only brain regions with at least 40 units were analyzed. Analyses specific to TF-responsive units were done only for brain regions with ≥ 10 of such units. No further sample size calculations were performed. |
| Data exclusions | Data excluded if mice did not perform well after a recording chamber implant (1 mouse excluded) Data Table 1). Experiments were independently performed by A.K. and M.L. |
| Replication | For vast majority of brain regions, recordings were pulled across multiple recordings sessions from multiple mice (see Extended Data Table 1). Experiments were independently performed by A.K. and M.L. |
| Randomization | Trials with different change sizes were randomly interleaved. Change times were were drawn from two distributions in blocks of 30 correct trials: 3-8 seconds, and 10.5-15.5 seconds. |
| Blinding | Curation of units quality and stability was done without the knowledge from which brain regions the recordings were done. Though the subsequent analyses pipeline was applied in the same manner to data from all applicable brain regions, the custom nature of analyses prevented investigators to remain blind to the identity of brain regions or dataset type (trained vs. naive mice). |

# Behavioural & social sciences study design

All studies must disclose on these points even when the disclosure is negative.

| | |
|---|---|
| Study description | *Briefly describe the study type including whether data are quantitative, qualitative, or mixed-methods (e.g. qualitative cross-sectional, quantitative experimental, mixed-methods case study).* |
| Research sample | *State the research sample (e.g. Harvard university undergraduates, villagers in rural India) and provide relevant demographic* |

| | |
|---|---|
| Research sample | *information (e.g. age, sex) and indicate whether the sample is representative. Provide a rationale for the study sample chosen. For studies involving existing datasets, please describe the dataset and source.* |
| Sampling strategy | *Describe the sampling procedure (e.g. random, snowball, stratified, convenience). Describe the statistical methods that were used to predetermine sample size OR if no sample-size calculation was performed, describe how sample sizes were chosen and provide a rationale for why these sample sizes are sufficient. For qualitative data, please indicate whether data saturation was considered, and what criteria were used to decide that no further sampling was needed.* |
| Data collection | *Provide details about the data collection procedure, including the instruments or devices used to record the data (e.g. pen and paper, computer, eye tracker, video or audio equipment) whether anyone was present besides the participant(s) and the researcher, and whether the researcher was blind to experimental condition and/or the study hypothesis during data collection.* |
| Timing | *Indicate the start and stop dates of data collection. If there is a gap between collection periods, state the dates for each sample cohort.* |
| Data exclusions | *If no data were excluded from the analyses, state so OR if data were excluded, provide the exact number of exclusions and the rationale behind them, indicating whether exclusion criteria were pre-established.* |
| Non-participation | *State how many participants dropped out/declined participation and the reason(s) given OR provide response rate OR state that no participants dropped out/declined participation.* |
| Randomization | *If participants were not allocated into experimental groups, state so OR describe how participants were allocated to groups, and if allocation was not random, describe how covariates were controlled.* |

# Ecological, evolutionary & environmental sciences study design

All studies must disclose on these points even when the disclosure is negative.

| | |
|---|---|
| Study description | *Briefly describe the study. For quantitative data include treatment factors and interactions, design structure (e.g. factorial, nested, hierarchical), nature and number of experimental units and replicates.* |
| Research sample | *Describe the research sample (e.g. a group of tagged Passer domesticus, all Stenocereus thurberi within Organ Pipe Cactus National Monument), and provide a rationale for the sample choice. When relevant, describe the organism taxa, source, sex, age range and any manipulations. State what population the sample is meant to represent when applicable. For studies involving existing datasets, describe the data and its source.* |
| Sampling strategy | *Note the sampling procedure. Describe the statistical methods that were used to predetermine sample size OR if no sample-size calculation was performed, describe how sample sizes were chosen and provide a rationale for why these sample sizes are sufficient.* |
| Data collection | *Describe the data collection procedure, including who recorded the data and how.* |
| Timing and spatial scale | *Indicate the start and stop dates of data collection, noting the frequency and periodicity of sampling and providing a rationale for these choices. If there is a gap between collection periods, state the dates for each sample cohort. Specify the spatial scale from which the data are taken* |
| Data exclusions | *If no data were excluded from the analyses, state so OR if data were excluded, describe the exclusions and the rationale behind them, indicating whether exclusion criteria were pre-established.* |
| Reproducibility | *Describe the measures taken to verify the reproducibility of experimental findings. For each experiment, note whether any attempts to repeat the experiment failed OR state that all attempts to repeat the experiment were successful.* |
| Randomization | *Describe how samples/organisms/participants were allocated into groups. If allocation was not random, describe how covariates were controlled. If this is not relevant to your study, explain why.* |
| Blinding | *Describe the extent of blinding used during data acquisition and analysis. If blinding was not possible, describe why OR explain why blinding was not relevant to your study.* |

Did the study involve field work? ☐ Yes ☐ No

# Field work, collection and transport

| | |
|---|---|
| Field conditions | *Describe the study conditions for field work, providing relevant parameters (e.g. temperature, rainfall).* |
| Location | *State the location of the sampling or experiment, providing relevant parameters (e.g. latitude and longitude, elevation, water depth).* |
| Access & import/export | *Describe the efforts you have made to access habitats and to collect and import/export your samples in a responsible manner and in compliance with local, national and international laws, noting any permits that were obtained (give the name of the issuing authority, the date of issue, and any identifying information).* |

| Disturbance | Describe any disturbance caused by the study and how it was minimized. |
|---|---|

# Reporting for specific materials, systems and methods

We require information from authors about some types of materials, experimental systems and methods used in many studies. Here, indicate whether each material, system or method listed is relevant to your study. If you are not sure if a list item applies to your research, read the appropriate section before selecting a response.

## Materials & experimental systems

| n/a | Involved in the study |
|---|---|
| ☒ | ☐ Antibodies |
| ☒ | ☐ Eukaryotic cell lines |
| ☒ | ☐ Palaeontology and archaeology |
| ☐ | ☒ Animals and other organisms |
| ☒ | ☐ Clinical data |
| ☒ | ☐ Dual use research of concern |
| ☒ | ☐ Plants |

## Methods

| n/a | Involved in the study |
|---|---|
| ☒ | ☐ ChIP-seq |
| ☒ | ☐ Flow cytometry |
| ☒ | ☐ MRI-based neuroimaging |

## Antibodies

| Antibodies used | Describe all antibodies used in the study; as applicable, provide supplier name, catalog number, clone name, and lot number. |
|---|---|
| Validation | Describe the validation of each primary antibody for the species and application, noting any validation statements on the manufacturer's website, relevant citations, antibody profiles in online databases, or data provided in the manuscript. |

## Eukaryotic cell lines

Policy information about cell lines and Sex and Gender in Research

| Cell line source(s) | State the source of each cell line used and the sex of all primary cell lines and cells derived from human participants or vertebrate models. |
|---|---|
| Authentication | Describe the authentication procedures for each cell line used OR declare that none of the cell lines used were authenticated. |
| Mycoplasma contamination | Confirm that all cell lines tested negative for mycoplasma contamination OR describe the results of the testing for mycoplasma contamination OR declare that the cell lines were not tested for mycoplasma contamination. |
| Commonly misidentified lines (See ICLAC register) | Name any commonly misidentified cell lines used in the study and provide a rationale for their use. |

## Palaeontology and Archaeology

| Specimen provenance | Provide provenance information for specimens and describe permits that were obtained for the work (including the name of the issuing authority, the date of issue, and any identifying information). Permits should encompass collection and, where applicable, export. |
|---|---|
| Specimen deposition | Indicate where the specimens have been deposited to permit free access by other researchers. |
| Dating methods | If new dates are provided, describe how they were obtained (e.g. collection, storage, sample pretreatment and measurement), where they were obtained (i.e. lab name), the calibration program and the protocol for quality assurance OR state that no new dates are provided. |

☐ Tick this box to confirm that the raw and calibrated dates are available in the paper or in Supplementary Information.

| Ethics oversight | Identify the organization(s) that approved or provided guidance on the study protocol, OR state that no ethical approval or guidance was required and explain why not. |
|---|---|

Note that full information on the approval of the study protocol must also be provided in the manuscript.

# Animals and other research organisms

Policy information about studies involving animals; ARRIVE guidelines recommended for reporting animal research, and Sex and Gender in Research

| | |
|---|---|
| Laboratory animals | Mice of C57BL6 background, see manuscript methods for further information |
| Wild animals | No wild animals were used in the study |
| Reporting on sex | Male mice were used in the study |
| Field-collected samples | No field-collected samples were used in the study |
| Ethics oversight | All experiments were performed under the UK Animals (Scientific Procedures) Act of 1986 (PPL: PD867676F) following local ethical approval by the Sainsbury Wellcome Centre Animal Welfare Ethical Review Body |

Note that full information on the approval of the study protocol must also be provided in the manuscript.

# Clinical data

Policy information about clinical studies
All manuscripts should comply with the ICMJE guidelines for publication of clinical research and a completed CONSORT checklist must be included with all submissions.

| | |
|---|---|
| Clinical trial registration | *Provide the trial registration number from ClinicalTrials.gov or an equivalent agency.* |
| Study protocol | *Note where the full trial protocol can be accessed OR if not available, explain why.* |
| Data collection | *Describe the settings and locales of data collection, noting the time periods of recruitment and data collection.* |
| Outcomes | *Describe how you pre-defined primary and secondary outcome measures and how you assessed these measures.* |

# Dual use research of concern

Policy information about dual use research of concern

## Hazards

Could the accidental, deliberate or reckless misuse of agents or technologies generated in the work, or the application of information presented in the manuscript, pose a threat to:

No | Yes
☐ | ☐ Public health
☐ | ☐ National security
☐ | ☐ Crops and/or livestock
☐ | ☐ Ecosystems
☐ | ☐ Any other significant area

## Experiments of concern

Does the work involve any of these experiments of concern:

No | Yes
☐ | ☐ Demonstrate how to render a vaccine ineffective
☐ | ☐ Confer resistance to therapeutically useful antibiotics or antiviral agents
☐ | ☐ Enhance the virulence of a pathogen or render a nonpathogen virulent
☐ | ☐ Increase transmissibility of a pathogen
☐ | ☐ Alter the host range of a pathogen
☐ | ☐ Enable evasion of diagnostic/detection modalities
☐ | ☐ Enable the weaponization of a biological agent or toxin
☐ | ☐ Any other potentially harmful combination of experiments and agents

# Plants

Seed stocks
*Report on the source of all seed stocks or other plant material used. If applicable, state the seed stock centre and catalogue number. If plant specimens were collected from the field, describe the collection location, date and sampling procedures.*

Novel plant genotypes
*Describe the methods by which all novel plant genotypes were produced. This includes those generated by transgenic approaches, gene editing, chemical/radiation-based mutagenesis and hybridization. For transgenic lines, describe the transformation method, the number of independent lines analyzed and the generation upon which experiments were performed. For gene-edited lines, describe the editor used, the endogenous sequence targeted for editing, the targeting guide RNA sequence (if applicable) and how the editor was applied.*

Authentication
*Describe any authentication procedures for each seed stock used or novel genotype generated. Describe any experiments used to assess the effect of a mutation and, where applicable, how potential secondary effects (e.g. second site T-DNA insertions, mosiacism, off-target gene editing) were examined.*

# ChIP-seq

## Data deposition

☐ Confirm that both raw and final processed data have been deposited in a public database such as GEO.

☐ Confirm that you have deposited or provided access to graph files (e.g. BED files) for the called peaks.

Data access links
*May remain private before publication.*
*For "Initial submission" or "Revised version" documents, provide reviewer access links.  For your "Final submission" document, provide a link to the deposited data.*

Files in database submission
*Provide a list of all files available in the database submission.*

Genome browser session
(e.g. UCSC)
*Provide a link to an anonymized genome browser session for "Initial submission" and "Revised version" documents only, to enable peer review.  Write "no longer applicable" for "Final submission" documents.*

## Methodology

Replicates
*Describe the experimental replicates, specifying number, type and replicate agreement.*

Sequencing depth
*Describe the sequencing depth for each experiment, providing the total number of reads, uniquely mapped reads, length of reads and whether they were paired- or single-end.*

Antibodies
*Describe the antibodies used for the ChIP-seq experiments; as applicable, provide supplier name, catalog number, clone name, and lot number.*

Peak calling parameters
*Specify the command line program and parameters used for read mapping and peak calling, including the ChIP, control and index files used.*

Data quality
*Describe the methods used to ensure data quality in full detail, including how many peaks are at FDR 5% and above 5-fold enrichment.*

Software
*Describe the software used to collect and analyze the ChIP-seq data. For custom code that has been deposited into a community repository, provide accession details.*

# Flow Cytometry

## Plots

Confirm that:

☐ The axis labels state the marker and fluorochrome used (e.g. CD4-FITC).

☐ The axis scales are clearly visible. Include numbers along axes only for bottom left plot of group (a 'group' is an analysis of identical markers).

☐ All plots are contour plots with outliers or pseudocolor plots.

☐ A numerical value for number of cells or percentage (with statistics) is provided.

## Methodology

Sample preparation
*Describe the sample preparation, detailing the biological source of the cells and any tissue processing steps used.*

Instrument
*Identify the instrument used for data collection, specifying make and model number.*

Software
*Describe the software used to collect and analyze the flow cytometry data. For custom code that has been deposited into a community repository, provide accession details.*

| Cell population abundance | *Describe the abundance of the relevant cell populations within post-sort fractions, providing details on the purity of the samples and how it was determined.* |
|---|---|
| Gating strategy | *Describe the gating strategy used for all relevant experiments, specifying the preliminary FSC/SSC gates of the starting cell population, indicating where boundaries between "positive" and "negative" staining cell populations are defined.* |

☐ Tick this box to confirm that a figure exemplifying the gating strategy is provided in the Supplementary Information.

# Magnetic resonance imaging

## Experimental design

| Design type | *Indicate task or resting state; event-related or block design.* |
|---|---|
| Design specifications | *Specify the number of blocks, trials or experimental units per session and/or subject, and specify the length of each trial or block (if trials are blocked) and interval between trials.* |
| Behavioral performance measures | *State number and/or type of variables recorded (e.g. correct button press, response time) and what statistics were used to establish that the subjects were performing the task as expected (e.g. mean, range, and/or standard deviation across subjects).* |

## Acquisition

| Imaging type(s) | *Specify: functional, structural, diffusion, perfusion.* |
|---|---|
| Field strength | *Specify in Tesla* |
| Sequence & imaging parameters | *Specify the pulse sequence type (gradient echo, spin echo, etc.), imaging type (EPI, spiral, etc.), field of view, matrix size, slice thickness, orientation and TE/TR/flip angle.* |
| Area of acquisition | *State whether a whole brain scan was used OR define the area of acquisition, describing how the region was determined.* |

Diffusion MRI    ☐ Used    ☐ Not used

## Preprocessing

| Preprocessing software | *Provide detail on software version and revision number and on specific parameters (model/functions, brain extraction, segmentation, smoothing kernel size, etc.).* |
|---|---|
| Normalization | *If data were normalized/standardized, describe the approach(es): specify linear or non-linear and define image types used for transformation OR indicate that data were not normalized and explain rationale for lack of normalization.* |
| Normalization template | *Describe the template used for normalization/transformation, specifying subject space or group standardized space (e.g. original Talairach, MNI305, ICBM152) OR indicate that the data were not normalized.* |
| Noise and artifact removal | *Describe your procedure(s) for artifact and structured noise removal, specifying motion parameters, tissue signals and physiological signals (heart rate, respiration).* |
| Volume censoring | *Define your software and/or method and criteria for volume censoring, and state the extent of such censoring.* |

## Statistical modeling & inference

| Model type and settings | *Specify type (mass univariate, multivariate, RSA, predictive, etc.) and describe essential details of the model at the first and second levels (e.g. fixed, random or mixed effects; drift or auto-correlation).* |
|---|---|
| Effect(s) tested | *Define precise effect in terms of the task or stimulus conditions instead of psychological concepts and indicate whether ANOVA or factorial designs were used.* |

Specify type of analysis:    ☐ Whole brain    ☐ ROI-based    ☐ Both

| Statistic type for inference<br><br>(See Eklund et al. 2016) | *Specify voxel-wise or cluster-wise and report all relevant parameters for cluster-wise methods.* |
|---|---|
| Correction | *Describe the type of correction and how it is obtained for multiple comparisons (e.g. FWE, FDR, permutation or Monte Carlo).* |

## Models & analysis

| n/a | Involved in the study |
|---|---|
| ☐ ☐ | Functional and/or effective connectivity |
| ☐ ☐ | Graph analysis |
| ☐ ☐ | Multivariate modeling or predictive analysis |

**Functional and/or effective connectivity**

*Report the measures of dependence used and the model details (e.g. Pearson correlation, partial correlation, mutual information).*

**Graph analysis**

*Report the dependent variable and connectivity measure, specifying weighted graph or binarized graph, subject- or group-level, and the global and/or node summaries used (e.g. clustering coefficient, efficiency, etc.).*

**Multivariate modeling and predictive analysis**

*Specify independent variables, features extraction and dimension reduction, model, training and evaluation metrics.*

