## [Peer Review file · Nature]

Manuscript Title: Linking sensation to action via evidence integration throughout the premotor brain

Editorial Notes:

Redactions – unpublished data

Reviewer Comments & Author Rebuttals

Reviewer Reports on the Initial Version:

Referees' comments:

Referee #1 (Remarks to the Author):

This manuscript involves a monumental effort to perform brain-wide single unit neural recordings in mice during a perceptual decision making task. The authors provide comprehensive analyses for how neural activity is associated with sensory, decision, and movement components of the task. This approach allows the authors to demonstrate multiple important organizing principles for these relationships across different brain regions amongst the widespread activity associated with the task. The authors show where activity related to sensory input is present across the brain in naïve mice that are presented with the visual input without having learned the decision task. They contrast this with more widespread activity related to the task in trained mice. They describe how components of this activity are related to the combination of sensory input over time that is used by mice for their decisions and differential participation of brain regions in that processing. They also show how this “evidence integration” alters neural activity in dimensions that are distinct from those engaged in the same circuits during the decision-reporting movement. Among the results reported in this manuscript, no single finding is utterly surprising and instead, the primary beauty of the work rests on how many important open questions are resolved by it and on the coherent picture that emerges for how neural activity is coordinated across brain regions to execute a decision making task. I have no doubt that this will be considered a seminal paper in the fields of decision and systems neuroscience and be important to people from multiple disciplines.

Overall, the approach is top notch and the methods are mostly well described. The analyses build on foundations laid by multiple distinct lines of prior work. They are thorough and well tailored to the questions addressed. The statistics are appropriate. The conclusions drawn are warranted based on the data and analyses. I have two major concerns, both of which I believe to be potentially addressable.

1) The authors repeatedly describe the task as involving “evidence integration”, and that is crucial for many of their results. Multiple compelling analyses show that the animals are doing some form of “integration” across time, although it might be relatively short timescales of integration, perhaps only a couple hundred milliseconds. I think that is fine, although the paper could be improved by more depth in considering the implications of the brevity of the timescale of integration. My bigger concern here is that the task is more complex than Figure 1 suggests, and I worry about what strategies the mice are employing, perhaps in addition to short-timescale evidence integration. There are two block types with

different temporal expectations. In one of the two block types, 90% of the trials have no change possible until more than 10 seconds after the start of the stimulus. One imagines that the mice could use the block type or elapsed time to improve task performance. The authors included probe trials and already have many of the tools at their disposal to assess if mice are adopting any strategies based on these task complexities. If so, are the results robust to these considerations? I would have even greater confidence in the manuscript if the authors could tackle this set of concerns.

2) Related to the previous point, I think the present work falls short in terms of the behavioral model. In a paper otherwise so comprehensive, the modeling seems more piecemeal and selectively applied to limited parts of the data. In reading the paper with care, I'm still left wondering how well does a model involving a single timescale of evidence integration to a fixed decision threshold fully and comprehensively explain the behavior. Beyond correlations between real and predicted RTs, how well does the model capture the actual RTs on average and the range of RTs across different change magnitudes? Can it explain different aspects of behavior (like early lick times and correct RTs) in a unified manner? Can it also explain the psychometric function of performance? Does it have similar parameter values for not just timescale but also threshold in explaining each of these? Where does the model fail? In answering these questions, the authors could also use the modeling as a tool to address the first concern as well.

In addition to these two big-picture concerns, here are some more specific comments:

Figure 1: b) Having a data point at 1 Hz is odd given the labeling of the axes. 1 Hz is the baseline, so I'm not sure what is being calculated as "% hits" for that data point. h) The right side is missing the description of what I presume to be real vs predicted firing rate traces.

Figure 3: b) What are the units of the threshold? I think maybe log-TF. Given the implementation of the model and that log-TF is a monotonic transformation of TF, I'm not sure I understand the motivation for the transformation. Even as log-TF, the threshold strikes me as rather small. I'm worried that the authors may be fitting only the trials with early licks (my worry is related to big-picture point #2 above). If that is the case, then does the model over-predict the number of early licks if it is applied to all trials and by how much? The methods say, "We did this for every trial", so perhaps the authors did this already or maybe that really meant every early lick trial. I also have concerns about the panel's color scale. The range is only from 15-16.75%, but is it even true that all the black area is 15% or is some of it less than 15%? Suffice it to say that there are multiple sources of confusion for this panel.

l/m) I'm confused why the post-peak drops in activity during the change interval for MOs would be consistent with integration, even leaky integration. The change is sustained, so leaky integration would predict a gradual increase to a steady state, but not a drop.

n) This panel was difficult to grasp the details of the how the analysis was done.

Figure 4: The authors do not seem to give any treatment of variability of intrinsic timescales within areas, or whether there were correlations between timescales and measures of neural integration at an individual neuron level.

Figure 6: a) Why is the motion onset mark not at the inflection point for the orthogonal modes hypothesis? f) The green line appears to be showing something different than what is described.

Referee #2 (Remarks to the Author):

This study characterizes encoding of sensory evidence, lick preparation, and lick execution, across many brain structures using large-scale Neuropixels recordings in mice. The study uses state-of-the-art methods and analyses to characterize brain-wide dynamics during behavior. The analyses are technically sound and well performed, and the figures are clear and convincing. However, the results are not novel, and mostly recapitulate previously published work. The main findings are the following.

- Sensory evidence, lick preparation, and lick execution are represented broadly throughout many brain regions.
- There is a hierarchy of timescales, with higher-order areas having more prolonged responses to sensory pulses.
- Neural activity reflects the strength of sensory evidence.
- Responses in higher-order areas, but not sensory areas, require learning.
- Neurons with pulse responses drive preparatory activity (this may be the most novel result), which is in a movement-null space.

Nearly all of these findings have been reported before, in some cases extensively. While the work is nicely done, there is no conceptual advance. Minor novel findings (e.g., integration timescales do not reflect intrinsic timescales, in disagreement with other studies) are niche and more appropriate for a specialized audience. Below, I highlight some things that I hope might improve the manuscript.

Major comment

1. The major weakness, in my view, is the behavioral evidence for accumulation of evidence. The animals could have used an alternative strategy, such as burst detection, instead of the gradual accumulation of sensory evidence over time. A few analyses that might strengthen the claims about an accumulation strategy: showing a relationship between response time and accuracy (controlling for change size). If the animals are accumulating evidence over time, performance should be better for trials with longer integration/response times. They should also compute the psychophysical kernel, to measure the behavioral impact of TFs at different time points relative to the change. A related alternative would be to perform logistic regression using TFs in different time bins relative to the change. The behavioral model was not terribly convincing on its own (e.g., performance seemed generally poor, see color axis in main figure panel). If the authors want to make claims about animals accumulating evidence, it is incumbent on them to demonstrate that.

Minor comments:

-I'm assuming that 1h shows model predictions for held-out test data; that should be clearly stated in

the legend.

-Is it possible to include the acronyms for brain regions in a figure legend instead of the supplement?

-In Figure 4, can they put binomial confidence intervals on the bar plots?

-4f: it's difficult to visually appreciate the difference in τ s. Would it be more clear to overlay the curves?

-For Figure 6, it doesn't appear that neural activity anywhere, including MOs, transitions into the movement subspace before the actual lick. How do the authors interpret the lack of dynamics preceding the action?

Typo: line 548 should be "during memory-guided movements"

Referee #3 (Remarks to the Author):

This paper studies how sensory evidence about the temporal frequency of a visual stimulus is accumulated over time and converted into a licking output. The main results of this paper are that longer sensory integration timescales arise after learning and that the accumulation of sensory evidence occurs in a distributed collection of premotor structures. This manuscript is excellent in its scope and methods. The behavioral task is interesting. The number of recordings and number of brain areas investigated is impressive. Equally impressive is how the manuscript explores in reasonable depth all the stages of processing from sensation to action. I like how the paper takes an exploratory, unbiased approach, whereas other work in this field has often started with targeted investigations. The experiments and analysis appear to be well done. The writing and presentation of complex data and analyses are clear and succinct. While there are many questions that come to mind regarding more analyses and experiments with these data and approaches, I do not feel it is justified to ask because the authors have already done a tremendous amount of work and presented clear results.

I was excited by this paper for the reasons mentioned above. However, my enthusiasm was dampened because, despite the impressive methods and data, I did not find a major advance that goes beyond what is already known in the field. For this journal, I was expecting such an advance. Instead, I find most of the main results to be largely confirmatory of what has already been presented in the field in recent years.

The conclusion that decision making happens in a premotor circuit is the topic of many old and new lines of decision making research. Many studies from Shadlen and colleagues have emphasized how evidence accumulation and decision making happen in a motor-relevant context in LIP in primates. More recent, Svoboda and colleagues have a long line of papers describing how decision making occurs in ALM, which is a premotor structure for tongue movements. Another example, Shadlen in mice reported decision making in a premotor circuit recently (Wu et al 2020 "Context-dependent decision making in a premotor circuit"). I was surprised that some of this literature was not cited, in particular the Wu et al. paper because its main finding (and title) regards the focus on premotor circuits.

The fact that the relevant circuit is distributed is perhaps new relative to the lines of work I mentioned above. But, it is somewhat expected because it is consistent with many papers showing distributed processing in the mouse brain. The authors already cite a large number of these papers, and there are many more.

The difference in the timescales of evidence accumulation across brain structures has been reported multiple times previously (one example, Pinto et al. "Multiple timescales of sensory-evidence accumulation across dorsal cortex"). The main difference here is that the authors find that these timescales are learned instead of intrinsic, whereas previous papers suggest they might be intrinsic. This is an interesting finding, but a modulation of an established theme.

The separation of movement-null and movement-potent dimensions follows closely from the work that Shenoy and colleagues have performed over the years.

I very much wanted to find a conceptual advance here because I liked a lot about this paper, but I struggled to find it. I wonder if the authors had similar challenges because the final, summative sentence of the abstract is "We propose that learning establishes the link between the external input and relevant premotor circuits throughout the brain, which integrate sensory evidence to drive activity dynamics that lead to the initiation of appropriate actions." This sentence, in my opinion, does not offer much insight beyond what has to be the case for any decision making task. All tasks of this type will necessitate learning to associate the sensory with the motor and brain dynamics will always precede actions.

If the other reviewers find interesting and novel major advances, then I am happy to be convinced. Perhaps one suggestion that might help would be to write out in the text what the alternative hypotheses or outcomes are, which might reveal what new has been learned.

One small thing: for some of the areas, the unit count is low, around tens of neurons. As we know, representations in many of these brain structures is sparse, so it might be challenging to make strong claims with limited sampling. Perhaps the authors could clarify which areas this might affect or present statistical arguments for why they have enough power with the current sample size in these low-count areas.

In summary, I am impressed by the scope and quality of this work. It is a study that puts together many previous findings and approaches into a single study. There is great value in this, and this leads to my considerable excitement about this paper. However, I was disappointed that I was not able to find a significant conceptual advance beyond what is already known in this field, which is something that I expect for this journal.

Author Rebuttals to Initial Comments:

Referee #1 (Remarks to the Author):

This manuscript involves a monumental effort to perform brain-wide single unit neural recordings in mice during a perceptual decision making task. The authors provide comprehensive analyses for how neural activity is associated with sensory, decision, and movement components of the task. This approach allows the authors to demonstrate multiple important organizing principles for these relationships across different brain regions amongst the widespread activity associated with the task. The authors show where activity related to sensory input is present across the brain in naïve mice that are presented with the visual input without having learned the decision task. They contrast this with more widespread activity related to the task in trained mice. They describe how components of this activity are related to the combination of sensory input over time that is used by mice for their decisions and differential participation of brain regions in that processing. They also show how this “evidence integration” alters neural activity in dimensions that are distinct from those engaged in the same circuits during the decision-reporting movement. Among the results reported in this manuscript, no single finding is utterly surprising and instead, the primary beauty of the work rests on how many important open questions are resolved by it and on the coherent picture that emerges for how neural activity is coordinated across brain regions to execute a decision making task. I have no doubt that this will be considered a seminal paper in the fields of decision and systems neuroscience and be important to people from multiple disciplines.

Thank you for your enthusiastic assessment!

Overall, the approach is top notch and the methods are mostly well described. The analyses build on foundations laid by multiple distinct lines of prior work. They are thorough and well tailored to the questions addressed. The statistics are appropriate. The conclusions drawn are warranted based on the data and analyses. I have two major concerns, both of which I believe to be potentially addressable.

1) The authors repeatedly describe the task as involving “evidence integration”, and that is crucial for many of their results. Multiple compelling analyses show that the animals are doing some form of “integration” across time, although it might be relatively short timescales of integration, perhaps only a couple hundred milliseconds. I think that is fine, although the paper could be improved by more depth in considering the implications of the brevity of the timescale of integration. Refer to paper (<https://elifesciences.org/articles/82823>), in a dynamic world it is no longer necessarily optimal to be a perfect integrator, but allow for decay to account for the dynamic nature.

This is an interesting point. As you rightly point out, in dynamic environments a perfect integrator is not necessarily optimal (Ruesseler et al. is now cited). We also see similar timescales of integration around ~ 0.25 seconds (See newly added panels to **Fig 3**, and **Rebuttal Figs 6-9** or **Extended Data Figs 8-9**). Short timescales of integration can be beneficial for discounting evidence from an irrelevant past, while still integrating potentially relevant recent stimuli. Indeed, leaky integration therefore makes sense in the context of our reaction time task in which the animals must monitor the dynamically fluctuating sensory evidence (updated every 50 ms) for up to 15.5s. This type of short-scale integration is also akin to ‘evidence smoothing’ rather than accumulation of noisy stimuli in order to denoise a noisy input. However, we refer to this process as integration because at the level of neural implementation ‘smoothing’ and ‘accumulation’ are two particular uses of neural integration - where the divide between these two strategies is determined by the ‘leakiness’ of the integration process. Indeed, the interpretation that animals may be using integration to smooth the evidence can be highlighted by showing that mice with these short integration kernels increase the signal-to-noise ratio during changes in a way that would be predictive of their lick times compared to if they were using a no integration strategy (**Rebuttal Fig 1; Extended Data Fig 9**). We have added this point to the discussion and **Extended Data Fig. 9**.

Rebuttal figure 1. Short time scale leaky evidence integration smooths and denoises visual evidence.

Signal to noise ratio (signal (S) / σ) increases during the change period when stimuli are transformed though leaky integration – effectively denoising through smoothing the evidence over time - aiding detection of small changes in noise, and is predictive of faster reaction times for larger changes compared to smaller changes.

My bigger concern here is that the task is more complex than Figure 1 suggests, and I worry about what strategies the mice are employing, perhaps in addition to short-timescale evidence integration. There are two block types with different temporal expectations. In one of the two block types, 90% of the trials have no change possible until more than 10 seconds after the start of the stimulus. One imagines that the mice could use the block type or elapsed time to improve task performance. The authors included probe trials and already have many of the tools at their disposal to assess if mice are adopting any strategies based on these task complexities. If so, are the results robust to these

considerations? I would have even greater confidence in the manuscript if the authors could tackle this set of concerns.

Thank you for raising this point. We are currently analysing these data for another paper that will address the issue of temporal expectation and time on mouse behaviour and neural responses. But to address your point here in the rebuttal, we indeed find that mice use their knowledge of being in 'early' or 'late' expectation blocks to alter the timing of early licks (**Rebuttal Fig 2a,b**). Despite these differences in timing of behaviour, mice use sensory information in a similar manner across blocks to inform their decision to lick. First, early licks are preceded by an increase in stimulus TF with a similar temporal profile during both early and late blocks (**Rebuttal Fig 2c**), even if we constrain the timing of early licks to the same interval on both blocks (**Rebuttal Fig 2d**).

[REDACTED]

Second, we find that the overall psychometric and chronometric performances are similar between early and late blocks, with some minor differences (**Rebuttal Fig 2e,f**). These differences, however, are much more subtle than the overall range of states that mice go through while performing this task (**Rebuttal Fig 5**). Furthermore, exploring the probe trials (on 10% of trials where the change times are taken from the alternative blocks change time distribution), we can see that if a trial happens unexpectedly early, the mice have a harder time detecting the changes, and respond to small changes slightly slower than if they were expected (**Rebuttal Fig 2g,h**). It is worth noting that this effect even runs contra to what would be expected from the small changes in psychometric and chronometric

curves of early and late blocks (**Rebuttal Fig 2c,d**), where we can see that mice generally perform slightly better when they do not have to wait as long for the change.

Together, these results suggest that mice also make use of temporal expectation as an additional factor to determine when to lick, but that in all cases, when they lick, they are guided by an integrated stimulus, which is the main phenomenon studied in the current paper. Consistent with behavioural results above, we find that TF-responsive neurons integrate sensory evidence similarly in early and late blocks, as the width of the responses to pulses of sensory evidence is not obviously different (**Rebuttal Fig 3**).

[REDACTED]

We believe that these results corroborate the finding that leaky temporal integration on a scale of several hundred milliseconds is a key factor in shaping both the behaviour and neural responses irrespective of the temporal expectation block. The main results presented in this paper are therefore robust to these task variations. Because behavioural performance and neural responses are qualitatively similar across early and late blocks for the questions covered in this paper, here we chose to combine the data across blocks. We have now added a statement to the methods section that reflects this point.

2) Related to the previous point, I think the present work falls short in terms of the behavioral model. In a paper otherwise so comprehensive, the modeling seems more piecemeal and selectively applied to limited parts of the data. In reading the paper with care, I'm still left wondering how well does a model involving a single timescale of evidence integration to a fixed decision threshold fully and comprehensively explain the behavior. Beyond correlations between real and predict RTs, how well does the model capture the actual RTs on average and the range of RTs across different change magnitudes? Can it explain different aspects of behavior (like early lick times and correct RTs) in a unified manner? Can it also explain the psychometric function of performance? Does it have similar parameter values for not just timescale but also threshold in explaining each of these? Where does

the model fail? In answering these questions, the authors could also use the modeling as a tool to address the first concern as well.

The leaky integrator model was intentionally simplistic, and not meant to capture the full extent of mouse behaviour in the task. Rather it was made to test whether a model with leaky integration performs better than models with either perfect integration (no leak) or without integration at all (outlier detection) in predicting single trial lick times. And indeed, this is what we observed. Of course, any such simplistic model with a minimum set of tuneable parameters is not expected to capture the richness of mouse behaviour, especially as it doesn't model the animals' engagement, impulsivity, or temporal expectation, which can vary over shorter or longer timescales (15-s long trials or ~1.5 hour long sessions). Yet, the model does clearly show that compared to an outlier detection strategy, a leaky integration of evidence can explain significantly more of the behaviour. Moreover, the optimal timescale of integration is similar to what we observe in neurons in premotor areas of the brain, and also to multiple new lines of evidence for behavioural integration which we have now included in the paper and this rebuttal (see **Rebuttal Figs 6-9**).

Nonetheless, to directly address your more specific questions about the model, we have made a **Rebuttal Fig 4** (and **Extended Data Fig 9**) to show you more in detail how the summaries in the figure are constructed, and what aspects the model performs well, and what aspects of behaviour it is not capturing (**Rebuttal Fig 4**). Firstly, the lick-triggered average of model-detected licks are both more likely to be stimulus driven with the whole lick-triggered stimulus average, including the integrative tail, scaling when isolating model-detected licks (**Rebuttal Fig 4b**). This is in accordance with the idea that the model identifies integrated stimulus driven licks (or the licks preceded by most stimulus evidence). However, it is also clear from the lick triggered stimulus average of the model-undetected licks that the model does not capture all stimulus driven early licks. Furthermore, the mean reaction time to different change sizes scale in a similar way to mouse behaviour (albeit with overall faster reaction times) (**Rebuttal Fig 4c,d**).

Rebuttal figure 4 (part of Extended Data Fig 10): A simple two parameter leaky integrator model supports behavioural evidence integration

a, Parameter search grid identifying which values the integration time and threshold best predicts early licks. **b**, Lick-triggered stimulus average of early licks detected by the leaky integrator model, and early licks not detected by the model. **c**, Mean decision value (integrated TF) after filtering stimulus through a leaky integrator model with a τ of 0.25 s. The dashed red line shows the model threshold value. **d**, Mean reaction time curve for leaky integrator model (τ : 0.25 s). **e**, relationship between real reaction time and predicted reaction time from leaky integrator model (τ : 0.25 s) for change size 1.25 Hz of

example mouse 12. **f**, Same as in **e**, but separating out early and late block to show the robustness of this relationship across blocks. **g,h** Same as **e,f**, but for change size 1.35 Hz.

Finally, we have included example scatter plots from which the single trial reaction time correlations shown in the **original Fig 3a-e** (now **Extended Data Fig 9**) are calculated (**Rebuttal Fig 4e,g**). Here you can see that for change sizes 1.25 Hz and 1.35 Hz of example mouse #12 the single trial reaction time scales with the model predicted reaction time, but the model reaction time is overall faster than mouse behaviour. We also show here that this relationship is independent of which temporal expectation block the mouse is in (**Rebuttal Fig 4f,h**).

Rebuttal Fig 4c shows this model effectively never misses a trial with a change >1Hz . It is important to note that this is not unlikely for a fully engaged animal. When accounting for behavioural state using a Hidden Markov Model (HMM) approach performing state inference over trial outcomes (Ashwood et al., 2022, *Nature Neurosci*), it becomes evident that when mice are in a fully engaged state (State 1, **Rebuttal Fig 5**) - which they are in the majority of the time (62 % of the time) - they almost never miss a change and respond very quickly (**Rebuttal Fig 5**). Still, the model reacts faster than the animals' behaviour overall, which may be caused by additional factors such as temporal expectation, which our simple model does not take into account. But the finding that the reaction times of the model correlate with real reaction times, and that the reaction times are better predicted when the model integrates evidence over ~0.25 s, means that integration is a stronger determinant of single trial reaction times than an outlier strategy would be. So, although incompletely capturing all behaviour, the strength of this model comes from the fact that we can provide this evidence with a minimal model with only two parameters (integration time and threshold).

Rebuttal figure 5: Behavioural performance during different behavioural states identified with a Hidden Markov Model (HMM)

Performing state identification using an HMM based on trial outcome (and identifying optimal number of states using a combined HMM and GLM (logistic regression) approach) we can see when mice are in a fully engaged state (state 1) the majority of the time (62% of trials). In this state behaviour is near perfect, and reaction times are fast. State 2 is defined as a state of low engagement with more misses and slower reaction times (as well as fewer early licks and aborts). Finally state 3 - which is the least occupied state- is defined by less engaged behaviour compared to state 1 and is marked by large fractions of aborts.

Nevertheless, we also agree with you that capturing behaviour comprehensively needs a more complex model. We have developed such a model for this task in a previous paper from our lab

(Orsolich et al., 2021, *Neuron*), which models the lick probability across time as a function of time-in-trial and stimulus preceding licks (“Gaussian process model”). Please see **point #3** in the section below, and Orsolich et al., 2021 for a detailed description.

In response to your query, however, we have we have now added several lines of new evidence, which empirically demonstrate that mice use temporal integration of the stimulus to drive their behaviour. These are:

1. A recent paper about evidence integration in a dynamic sensory environment used the tau of an “integration kernel” - the stimulus preceding a response (equivalent to our lick-triggered stimulus average) to measure the window over which humans integrated evidence (Ruesseler et al., 2023, *eLife*). We have now applied a similar measure by estimating the tau of the lick-triggered stimulus average to estimate the time course over which sensory evidence is used to guide the mice’s sensory decision. We find that in our task the tau of the lick-triggered stimulus average is 0.27 s and very similar to the tau derived from our simple leaky integrator model of 0.25 s (**Rebuttal Fig 6; Fig 3a-b; Extended Data Fig 9**).

The shape of lick-triggered stimulus average however partially depends on the distribution of trial-to-trial decision-to-motor delays, which could in principle create a false impression that mice integrate sensory evidence. We therefore constructed an artificial agent with internal noise and decision-to-motor delay distribution derived from data, and trained it on mice data (see methods). This agent lacked the ability to integrate sensory evidence and instead used an outlier detection strategy. This artificial outlier detection agent performing the same trials had a much shorter decay time constant of the lick-triggered stimulus average (**Rebuttal Fig 6b**; outlier detection agent tau = 0.17 s), demonstrating that trial-to-trial variance in decision-motor delays cannot account for the width of the lick triggered stimulus average.

Rebuttal figure 6 (New Fig 3a): Estimation of integration time from lick triggered stimulus average (i.e., integration kernel)
a, Lick triggered stimulus average with fitted decay time (τ) of the mice performing the task and in an artificial outlier detection agent performing the same task (and same trials). **b** mean and 95% range of decay times from 10,000 iteration of the outlier detection agent (purple) compared to the tau of the mouse data (black). The τ s are estimated from fits of an exponential decay function starting at the peak mean TF value.

2. To empirically validate that mice use multiple pulses of sensory evidence (i.e., integration) to influence their decision to lick during the baseline period, we analysed how early lick probability is influenced by magnitudes and timing of preceding TF pulses. First, we show that the deviation of a TF pulse relative to the mean baseline 1Hz makes mice correspondingly more or less likely to make an early lick within the subsequent 0.2-1.0 s (**Rebuttal Fig 7a,b**). The outlier detection agent (**Rebuttal Fig 7b**, purple line) was however influenced only by fast TF pulses that represent positive evidence. We next looked at how combinations of two TF pulses of given magnitudes and time between them affect the early lick probability (see upper panels in **Rebuttal Fig 7c and d** for cases of consecutive pulses and with 100 ms delay between pulses for mouse data, and the upper panel in **Rebuttal Fig 7e** for outlier detection agent data). We then compared the observed effects to the predicted probability of mice licking if the two pulses were independently influencing the lick probability (**Rebuttal Fig 7c-e**, middle panels; see a new section in Methods). By exploring the difference between the lick probabilities after two TF pulses in the mouse data from the

independent effect of pulses, we find that two fast pulses occurring within a window of ~ 0.25 s significantly increase the probability of licking, relative to what would be predicted from two pulses being treated independently by the mice (**Rebuttal Fig 7c,d** lower panels and **f**). Critically, the independent effect of TF pulses was sufficient to explain the observed lick probability for any combination of TF pulse magnitudes and timing between them in outlier detection agent data (**Rebuttal Fig 7d,f** purple line). This provides a purely empirical basis for demonstrating that mice behaviourally integrate multiple pulses of evidence on a trial-to-trial basis over ~ 0.25 s during the baseline period (**Rebuttal Fig 7f**). Moreover, we observe similar results even if data are analysed separately for early and late blocks (**Rebuttal Fig 7f**, red and blue lines respectively). Note that this time-scale is very similar to what we found from the tau of the lick-triggered stimulus average and in our leaky integrator model. We have added these analyses to the paper on **Extended Data Fig 8**.

Rebuttal figure 7 (Extended Data Fig 8): Effect of magnitude and timing of TF pulse on probability of early licks.

a, Conditional probability of early lick at a specific time after a TF pulse of given magnitude. Here and later early lick probability is shown relative to the probability at the mean baseline TF (1 Hz) **b**, Probability of early lick after a TF pulse of given magnitude (here and later cumulatively within [0.2, 1] s window). Mice data is shown in black, outlier detection agent in purple. **c**, Upper panel: probability of early lick after two sequential TF pulses of given magnitudes; middle panel: expected effect if both pulses influence early lick probability independently; lower panel: difference from the independent effect of TF pulses. **d**, The same format as in **c**, but for two TF pulses with 100 ms delay between them. **e**, The same format as in **c**, but shown for data generated by the outlier detection agent (for two sequential TF pulses). **f**, Difference between observed probability of early lick after a sequence of two fast TF pulses and the one predicted from their independent effect (top right corner in lower panel **c**, **d**), normalized by the expected probability from the effect of independent pulses and shown as a function of delay between them.

3. A recent paper from our lab (Orsolich et al., 2021, *Neuron*; <https://doi.org/10.1016/j.neuron.2021.03.031>), using the same behavioural paradigm as ours, introduced a Gaussian Process model which estimates lick probability at every point in time of a trial as a function of elapsed time and fitted stimulus filters (**Rebuttal Fig. 8a**), to predict when

mice lick during our task. We applied this model to mice in our study (**Rebuttal Fig. 8**). This revealed that the timing of early licks (**Rebuttal Fig. 8b**), the psychometric curves (**Rebuttal Fig. 6c**), and reaction time curves (**Rebuttal Fig. 8d**) are well captured by this model. The model was trained on both early licks and hits/misses during the change period and showed that the stimulus filter best accounting for when mice lick was a filter with an integration time constant of ~ 230 ms (Stim filter #1; **Rebuttal Fig. 8a**). Importantly, inactivating this leaky integration filter caused the model performance to collapse, while inactivating stimulus filter #2 – the derivative filter – lead only to partial reduction in performance (**Rebuttal Fig. 8e,f**). This suggests that integration of the stimulus over several hundred milliseconds contributes to the animals' decision to lick.

Rebuttal figure 8: Gaussian process modelling of behaviour

a, Top 3 stimulus filters (in order of contribution to model) driving licks from the Gaussian Process model. b, Early lick density for early (blue) and late (orange) block, for 10% held out (cross-validated) dataset. c, Predicted psychometric curve (grey – error bars are constructed from 200 reiterations of the model fit) vs real psychometric (red) curve, on held out (cross validated) data set. d, Predicted mean reaction time curve (grey – error bars are constructed from 200 reiterations of the model fit) vs real reaction time curve (red), on held out (cross validated) data set. e, Predicted psychometric curve with all filters active (black), filter 1 (integration filter) made inactive (mauve), or filter 2 made inactive (green), and real data (red) on held out (cross validated) data set. f, Predicted mean reaction time curve with all filters active (black), filter 1 (integration filter) made inactive (mauve), or filter 2 made inactive (green), and real data (red) on held out (cross validated) data set.

Consistent with this, note the close correspondence between the model's stimulus filter #1 and the early lick triggered stimulus average (psychophysical kernel), which have similar timescales (**Rebuttal Fig 9c**).

Rebuttal figure 9: Estimation of integration time from GP model, and comparison to lick triggered stimulus average
a, Early lick triggered stimulus average with fitted decay time (τ). The τ is estimated from a fit of an exponential decay function starting at the peak mean TF value. b, First stimulus filter from GP model (see **Rebuttal Fig. 8**) with fitted decay time (τ). The τ is estimated from a fit of an exponential decay function starting at the peak mean filter value. c, Normalized lick triggered stimulus average and the first filter of GP model overlaid to highlight the similarity in integration times.

Finally, we want to highlight the observation that at a neuronal level we find strong evidence for leaky temporal integration of sensory evidence, with decay time constants similar to what we extract from behaviour (**Fig 3 f-g**). Unlike most studies investigating evidence integration having to rely on the speed at which activity ramps as a function of strength of evidence during decisions (where the ramp is assumed to mean the accumulation of evidence), we provide direct empirical evidence that individual brief pulses of evidence are combined across time at a single cell level (**Fig 3**), and that these give rise to ramping of activity during the change period (**Fig 3**).

These new analyses demonstrate that mice use integration as a key strategy to guide their decision. In the original paper, we tried to condense this argument into a simple model that was not meant to describe the behaviour in its entirety, but instead specifically capture that licks were more likely to be driven by leaky integration rather than no integration (and the integration times found with that model matched these expanded analyses). We did this in order to make room (without flooding the paper) for the expanded neuronal analyses capturing the single-cell and population level neural dynamics that translate sensory evidence into actions. In the paper, we have now expanded this simple model of behaviour with the series of evidence described above which we believe now convincingly demonstrate that integration is the key behavioural strategy that mice rely on to make their decision in this task.

In addition to these two big-picture concerns, here are some more specific comments:

Figure 1: b) Having a data point at 1 Hz is odd given the labeling of the axes. 1 Hz is the baseline, so I'm not sure what is being calculated as "% hits" for that data point. h) The right side is missing the description of what I presume to be real vs predicted firing rate traces.

You're right, 1Hz data point represents the baseline (i.e., no change). This gives the baseline lick rate given no change (in the same response window as changes would occur in), to illustrate the rate of false alarms. We have now revised this panel in **Fig 1b** to highlight that 1 Hz is capturing the false alarm rate.

Figure 3: b) What are the units of the threshold? I think maybe log-TF. Given the implementation of the model and that log-TF is a monotonic transformation of TF, I'm not sure I understand the motivation for the transformation. Even as log-TF, the threshold strikes me as rather small. I'm worried that the authors may be fitting only the trials with early licks (my worry is related to big-picture point #2 above). If that is the case, then does the model over-predict the number of early licks if it is applied to all trials and by how much? The methods say, "We did this for every trial", so perhaps the authors did this already or maybe that really meant every early lick trial. I also have concerns about the panel's color scale. The range is only from 15-16.75%, but is it even true that all the black area is 15% or is some of it less than 15%? Suffice it to say that there are multiple sources of confusion for this panel.

Similar to most psychophysical judgements, the perception of speed has been found to follow Weber's law – scaling of perceptible change with absolute values (Zanker, 1995; Nover et al., 2005). The justification of log₂ transforming the TF values – and the threshold being in log space - is therefore to allow the signal and noise to follow Weber's law in our task. For the same reason, the noise of the stimulus in our task is also in log₂ space, so in order for the input to be gaussian in the model, the input to the model needs to also be log₂ transformed.

In terms of threshold, this value is in integrator-transformed sample space. This means that a threshold value 0.038 is equivalent to an integrated TF threshold crossing of >1 octave above baseline. In other words, if integration was perfect, it would take more than 3 fast pulses 1 std away from the baseline to cause a predicted lick. In order for a single TF pulse to trigger a lick with this threshold would mean a TF with a value >3 std deviations away from the baseline (which will happen less than ~0.3% of the time).

In terms of the ‘we did this for every trial’, you are right that we estimated the model parameters on early lick trials only (this has now been clarified in the methods). However, the prediction of hit trial reaction times (during the change period) is based on the parameters identified on early lick trials, which means that we are predicting not just on a held-out dataset, but on a different part of the trial.

Finally, in terms of the black parts of the parameter search grid, we apologise for the confusion and thank you for bringing this to our attention – it was not stated clearly that this grid was thresholded. We thresholded this grid by whether the values were significantly better (using t-tests) than the no integration model, to highlight that the values shown are statistically significantly better than an outlier model. We have now changed this to a non-thresholded panel in **Extended Data Fig 9**.

l/m) I'm confused why the post-peak drops in activity during the change interval for MOs would be consistent with integration, even leaky integration. The change is sustained, so leaky integration would predict a gradual increase to a steady state, but not a drop.

We were also struck by the drop in neural activity during the change after the mice committed to a choice. Note that this drop of activity is present only in TF-responsive subpopulation (in MOs and other premotor regions). We believe this is a hallmark of neurons that *integrate* evidence rather than simply respond to sensory input. Initially, neurons ramp their activity in response to a change in stimulus speed. Once mice decide to lick, the integration process terminates, even though the stimulus remains on the screen for another 1s. This is particularly apparent in **Fig6c,h,j**, where activity of the TF responsive subpopulation in movement-null subspace peaks at the time of the lick onset. We agree that if those neurons would simply represent integrated evidence regardless of whether such evidence is useful for a mouse at a given moment of time, one would expect their activity not to decrease even after mouse commits to a decision. However, since this effect is prominent across all brain regions that integrate sensory evidence (**Extended Data Fig 11, Extended Data Fig 16b**) and also a qualitatively similar effect is seen in non-human primate literature following a saccade onset (Shadlen, & Newsome 2001; Roitman and Shadlen, 2002), though the stimulus doesn't last after the saccade there). We think this is a hallmark and what differentiates decision-related versus sensory activity in the brain, and now emphasise this point in the discussion.

n) This panel was difficult to grasp the details of the how the analysis was done.

We appreciate that this is a bit of a convoluted measure, but it is a succinct way to capture the relationship between ramping of neural kernel activity and amount of stimulus evidence (i.e., change size). To aid the understanding of this panel, we have now added an additional panel (**Fig 3m**) above it, which shows the variables from which these summaries are constructed. To capture the ramping speed, we measure the time to which the kernel for a given change size reaches 50% of its peak. We then perform a regression between these 50% peak times and the change sizes (**Fig 3m**). The slope of this correlation is what is summarised in **Fig 3n**, and shows that the time it takes to reach the 50% of peak is longer for smaller changes (i.e., a negative relationship).

Figure 4: The authors do not seem to give any treatment of variability of intrinsic timescales within areas, or whether there were correlations between timescales and measures of neural integration at an individual neuron level.

We have explored this possibility, and we do not find that intrinsic timescales are predictive of integration times at a single cell level, further highlighting the dissociation between integration times and intrinsic timescales. See **Rebuttal Fig 10 (Extended Data Fig 13)**. This figure has been added as **Extended Data Fig 13** to the manuscript.

Rebuttal figure 10 (Extended Data Fig 13): Intrinsic timescales and TF integration at a single neuron level

a, Intrinsic timescales (tau) estimated for each TF responsive unit across the brain vs the TF response width for those units. Intrinsic times scales do not correlate with TF response width at a single cell level ($p > 0.05$). **b**, Same as in **a** but with units divided into major area groups. No area group has significant correlation between intrinsic times scales and TF response width at a single cell level ($p > 0.05$). **c**, Same as figure 4g, but here areal intrinsic time scale is extracted from TF responsive units only. In agreement with figure 4g, there is no correlation between areal intrinsic timescales and median TF response width. **d**, intrinsic timescales of TF responsive units are similar to the intrinsic timescales as areas as a whole.

Figure 6: a) Why is the motion onset mark not at the inflection point for the orthogonal modes hypothesis?

The transition of activity from movement-null to movement subspace will likely happen at the moment of decision, which can also be thought as a time of commitment to an action. The observable onset of movement however will happen later in time and requires at least some elevation in activity of motor neurons, which should correspond to some positive value along the movement dimension in **Fig. 6a** and also parallels what we see in data (**Fig. 6b,d**).

f) The green line appears to be showing something different than what is described.

Perhaps what was confusing is that while panel f shows a relative occupancy between movement and movement-null subspaces, the green line shows a time of peak occupancy in movement-null subspace (and not a peak of relative occupancy). Our aim was to show an important metric from **Extended Data Figure 16** on the main figure. This is now made clear in the legend.

Thank you.

Referee #2 (Remarks to the Author):

This study characterizes encoding of sensory evidence, lick preparation, and lick execution, across many brain structures using large-scale Neuropixels recordings in mice. The study uses state-of-the-art methods and analyses to characterize brain-wide dynamics during behavior. The analyses are technically sound and well performed, and the figures are clear and convincing. However, the results are not novel, and mostly recapitulate previously published work. The main findings are the following.

-Sensory evidence, lick preparation, and lick execution are represented broadly throughout many brain regions.

-There is a hierarchy of timescales, with higher-order areas having more prolonged responses to sensory pulses.

-Neural activity reflects the strength of sensory evidence.

-Responses in higher-order areas, but not sensory areas, require learning.

-Neurons with pulse responses drive preparatory activity (this may be the most novel result), which is in a movement-null space.

Nearly all of these findings have been reported before, in some cases extensively. While the work is nicely done, there is no conceptual advance. Minor novel findings (e.g., integration timescales do not reflect intrinsic timescales, in disagreement with other studies) are niche and more appropriate for a specialized audience. Below, I highlight some things that I hope might improve the manuscript.

Thank you for your comments. Before addressing your specific points, we believe it's important to consider the central question: What advances can be made by obtaining data from the entire brain, as opposed to sampling a subset of individual brain areas, towards understanding perceptual decision-making? We think they are:

- (1) We can show definitively that the evidence integration is distributed across almost the entire brain, as opposed to guessing that this is true. This therefore provides the ground truth data that should underlie future models of perceptual decision making.
- (2) We can show that – at the level of neural dynamics - the evidence integration subspace is aligned with the motor preparation subspace, but orthogonal to the movement subspace, across the entire brain. Therefore, the transformation of accumulated evidence into movement planning and execution takes place within and across subspaces of neural activity that are shared across multi-regional circuits, rather than proceeding successively across a subset of specialized brain areas. This reveals the universality of the transitions in neural dynamics surrounding movement, and allows integration of influential ideas from perceptual decision making and motor control fields into a unified framework.

- (3) We can discern which brain areas have dynamics that are sculpted by learning. The answer is essentially most of them apart from early visual areas. If this was known, we are not aware of it. For example, we might have guessed that integration would only take place in a limited subset of areas, or that the evidence integration would have emerged along a predefined hierarchy, or even that visual regions would also have learnt to integrate evidence. Instead, we show that the longest timescales are present in a distributed set of structures from cortex to basal ganglia to cerebellum to midbrain, indicating that learning turns evidence integration into a highly parallelised process.
- (4) We can relate different properties of neural dynamics across brain areas in the same task. This allows us, for example, to show that the learned dynamics do not reflect the intrinsic dynamics across regions, contradicting several influential models of cortical computation (Murray et. al 2014, Siegle et al., 2021; Pinto et. al 2022, Manea et al., 2022; Cosyne 2022/2024 workshops).

We are proud of these findings as they demonstrate the benefits of brain-wide investigations of behaviour. They provide clear answers to major questions in the fields of perceptual decision making and motor control, and they offer a foundation for understanding many learnt behaviours.

Below we are expanded answers to your comments above:

We agree that our work builds on numerous studies over the past three decades that have focused on neural computations taking place in the brain during decision making. But we hope that you also agree that we as a field still lack a unified understanding of how the brain decides when to act based on ambiguous sensory evidence. Why? On one hand, the majority of previous studies in the field have studied one or several brain areas in isolation in well-controlled tasks that require temporal integration of sensory evidence, but which mostly focused on cortical areas or their immediate targets such as striatum and superior colliculus (e.g. beautiful work from Shadlen, Gold, Brody, Krauzlis labs, etc), thus neglecting the contributions of the majority of other brain regions. On the other hand, other studies had the necessary coverage via brain-wide recordings but in tasks that were not designed to disambiguate sensory evidence integration and motor preparation (Steinmetz et al, International Brain Laboratory, Svoboda lab). More generally, there is a general disconnect between studies that address motor preparation and execution (e.g. Svoboda, Shenoy, Churchland, Hantman, Dudman, Olfveky, Costa labs, etc) and those addressing how and where evidence integration takes place.

We believe that the impact of our study resides in bridging these two fields at a brain-wide scale. We were able to do this by taking advantage of a task designed to separate decision-relevant computations (neural representations of sensory encoding, sensory integration, motor preparation and execution) combined with brain-wide recordings before and after learning. This enabled us to reveal how these processes map onto single-cell and population level neural dynamics in different brain areas within a unified task and common analysis frameworks, and make several conceptually novel insights, which go beyond the findings that you listed above:

(i) Our results address the importance of learning in defining how sensory information propagates and is transformed throughout the brain. We show that learning establishes the coupling of sensory input and preparatory activity in premotor areas across all major brain's subdivisions. Importantly, the timescales of integration in these premotor areas are sculpted by learning since their intrinsic timescales are decoupled from integration timescales.

In relation to this, you state that a main finding is: *‘There is a hierarchy of timescales, with higher-order areas having more prolonged responses to sensory pulses.’* However, the main point that **learning can decouple the timescales required for evidence integration from intrinsic timescales** of each area is something that challenges a major assumption in the field (i.e. that intrinsic timescales determine information processing timescales). This is neither trivial nor incremental, because it suggests that learning and task requirements can mould the timescales of neural dynamics used to integrate information and guide decisions, rather than these timescales being strictly constrained by intrinsic circuit properties.

You also state that a main finding is: *‘Responses in higher-order areas, but not sensory areas, require learning’*. However, the main point is to untangle where responses are learned and where they are inherent to the system across the brain, and we show that **learning establishes decision-relevant (i.e., integrating) sensory responses in dozens (premotor) areas across all major subdivisions of the brain** – a finding that highlights the dramatic impact learning has on the penetrance of relevant input in shaping neural activity across the brain and raises new avenues for research to examine distributed neural mechanisms of learning.

(ii) Evidence integration is a wide-spread phenomenon in the brain, taking place in a sparse population of neurons in most premotor areas from which we measured activity. We believe this contrasts the impression from influential studies in the field which shaped the prevailing view that integration is restricted to several areas of neocortex (parietal and frontal cortex) and their immediate downstream targets (such as striatum and superior colliculus). Moreover, we show which of these distributed premotor areas start integrating earlier than others, revealing the successive transformation of sensory input into preparatory activity across the brain.

In relation to this, you state that a main finding is: *‘Sensory evidence, lick preparation, and lick execution are represented broadly throughout many brain regions.’* However, while **integration of evidence and preparatory activity are widespread, the key point is that these two processes are directly linked across the premotor brain** – thus revealing a global coupling between decision variable (integrated evidence) and motor preparation processes. Importantly, sensory input is more broadly encoded than integrated sensory evidence, but only integrated sensory evidence outside the visual system is coupled to motor preparation.

(iii) Preparatory activity always occupies a movement-null subspace across all premotor areas measured, thus demonstrating that the original findings by Kaufman, Churchland and Shenoy in motor cortex generalise to most brain structures. Crucially, **neurons that integrate evidence drive preparatory activity** in movement-null subspace (to the best of our knowledge, this has not been demonstrated in any area of the brain).

In relation to this, you state that a main finding is: *‘Neurons with pulse responses drive preparatory activity (this may be the most novel result), which is in a movement-null space.’* We agree with this statement, but want to emphasize two major implications. First, with this we demonstrate how accumulation of evidence and preparatory activity are linked throughout the premotor brain. Second, because the preparatory activity occupies movement-null subspace, we show how evidence can accumulate without it triggering the premature action.

In this way, we link evidence accumulation onto motor-related dynamics on a brain-wide scale for the first time. Moreover, these dynamics are widespread across the premotor brain regions and not localized to a few key areas. This contrasts with the current literature that tries to explain decision

making as the interaction between a small subset of cortical regions integrating information and their communication with downstream subcortical regions that threshold this input to drive movement (e.g. Stine et al., 2023). Our results redefine the problem conceptually, as we are now faced with a brain-wide multiregional challenge if we are to understand the mechanisms of where and how the commitment to choice is made.

Taken together, we regard these insights as not just conceptually novel but foundational for the field. We do agree with you that evidence accumulation has been observed in several brain areas, and that movement preparation or execution signals have been observed in many others. The questions we address in this paper, however, are how these computations are orchestrated at the brain-wide scale – both in terms of contribution of neurons that integrate information and in terms of neural dynamics that link the processes of evidence integration, movement preparation and execution. We believe our study unifies two fields of neuroscience (decision-making and motor control) that have largely been studied in isolation, and reveal a comprehensive brain-wide picture of the processes that unfold during as mice make a decision and, importantly, how they are shaped by learning.

Given your comments, we do recognise that some of the above insights were not adequately emphasised in the manuscript, and have rewritten several sections accordingly.

Major comment

1. The major weakness, in my view, is the behavioral evidence for accumulation of evidence. The animals could have used an alternative strategy, such as burst detection, instead of the gradual accumulation of sensory evidence over time. A few analyses that might strengthen the claims about an accumulation strategy: showing a relationship between response time and accuracy (controlling for change size). If the animals are accumulating evidence over time, performance should be better for trials with longer integration/response times.

They should also compute the psychophysical kernel, to measure the behavioral impact of TFs at different time points relative to the change. A related alternative would be to perform logistic regression using TFs in different time bins relative to the change.

The behavioral model was not terribly convincing on its own (e.g., performance seemed generally poor, see color axis in main figure panel). If the authors want to make claims about animals accumulating evidence, it is incumbent on them to demonstrate that.

Thank you for suggesting these behavioural analyses, which have previously been used to demonstrate the integration of sensory evidence (Brunton et. al 2013, Levi et al., 2018 and Stine et al., 2023). These analyses however ideally require a different design of behavioural task than ours, including having two types of behavioural choices, such that it is possible to make incorrect responses that are not a miss. Moreover, ideally the duration of the change epoch should be the same and not dependent on the reaction time. In our task, by design, responses are self-timed but mice make only one type of response (lick straight within a 2.15s window during the change period; otherwise it's a miss). This means it is not possible to calculate the relationship between response time and accuracy. For the same reasons, it is also not possible to calculate the psychophysical kernel aligned to a hit lick reaction time. While it is possible to calculate the psychophysical kernel aligned to the change onset, it will not be a sensitive measure (due to one type of behavioural choice) and it is not clear to us how we would interpret that psychophysical kernel in a meaningful way.

Instead, to address your comments directly, we opted to apply a different set of analyses that are better suited to our task design, demonstrating that mice principally use leaky integration of evidence as a behavioural strategy. These are (also mentioned to reviewer #1):

1. A recent paper about evidence integration in a dynamic sensory environment used the tau of an “integration kernel” - the stimulus preceding a response (equivalent to our lick triggered stimulus average) to measure the window over which humans integrated evidence (Ruesseler et al., 2023, *eLife*). We have now applied a similar measure by estimating the tau of the lick-triggered stimulus average to estimate the time course over which sensory evidence is used to guide the mice’s sensory decision. We find that in our task the tau of the lick-triggered stimulus average is 0.27 s and very similar to the tau derived from our simple leaky integrator model of 0.25 s (**Rebuttal Fig 6; Fig 3a; Extended Data Fig 9**).

The shape of lick-triggered stimulus average however partially depends on the distribution of trial-to-trial decision-to-motor delays, which could in principle create a false impression that mice integrate sensory evidence. We therefore constructed an artificial agent with internal noise and decision-to-motor delay distribution derived from data, and trained it on mice data (see Methods). This agent lacked the ability to integrate sensory evidence and instead used an outlier detection strategy. This artificial outlier detection agent performing the same trials had a much shorter decay time constant of the lick triggered stimulus average (**Rebuttal Fig 6b**; outlier detection agent $\tau = 0.17$ s), demonstrating that trial-to-trial variance in decision-motor delays cannot account for the width of the lick triggered stimulus average.

2. To empirically validate that mice use multiple pulses of sensory evidence (i.e., integration) to influence their decision to lick during the baseline period, we analysed how early lick probability is influenced by magnitudes and timing of preceding TF pulses. First, we show that the deviation of a TF pulse relative to the mean baseline 1Hz makes mice correspondingly more or less likely to make an early lick within the subsequent 0.2-1.0 s (**Rebuttal Fig 7a,b**). The outlier detection agent (**Rebuttal Fig 7b**, purple line) was however influenced only by fast TF pulses that represent positive evidence. We next looked at how combinations of two TF pulses of given magnitudes and time between them affect the early lick probability (see upper panels in **Rebuttal Fig 7c and d** for cases of consecutive pulses and with 100 ms delay in-between for mouse data, and the upper panel in **Rebuttal Fig 7e** for a case of sequential pulses TF pulses in outlier detection agent data). We then compared the observed effects to the predicted probability of mice licking if the two pulses were independently influencing the lick probability (**Rebuttal Fig 7c-e**, middle panels; see a new section in Methods). By exploring the difference between the lick probabilities after two TF pulses in the mouse data from the independent treatment of pulses, we find that two fast pulses occurring within a window of ~ 0.25 s significantly increase the probability of licking, relative to what would be predicted from two pulses being treated independently by the mice (**Rebuttal Fig 7c,d** lower

panels and **f**). Critically, the independent effect of TF pulses was sufficient to explain the observed lick probability for any combination of TF pulse magnitudes and timing between them in outlier detection agent data (**Rebuttal Fig 7d,f purple line**). This provides a purely empirical basis for demonstrating that mice behaviourally integrate multiple pulses of evidence on a trial-to-trial basis over ~ 0.25 s during the baseline period (**Rebuttal Fig 7f**). Moreover, we observe similar results even if data are analysed separately for early and late blocks (**Rebuttal Fig 7f, red and blue lines respectively**). Note that this time-scale is very similar to what we found from the tau of the lick-triggered stimulus average and our leaky integrator model. We have added these analyses to the paper on **Extended Data Fig 8**.

Rebuttal figure 7 (Extended Data Fig 8): Effect of magnitude and timing of TF pulse on probability of early licks.

a, Conditional probability of early lick at a specific time after a TF pulse of given magnitude. Here and later early lick probability is shown relative to the probability at the mean baseline TF (1 Hz) **b**, Probability of early lick after a TF pulse of given magnitude (here and later cumulatively within [0.2, 1] s window). Mice data is shown in black, outlier detection agent in purple. **c**, Upper panel: probability of early lick after two sequential TF pulses of given magnitudes; middle panel: expected effect if both pulses influence early lick probability independently; lower panel: difference from the independent effect of TF pulses. **d**, The same format as in **c**, but for two TF pulses with 100 ms delay between them. **e**, The same format as in **c**, but shown for data generated by the outlier detection agent (for two sequential TF pulses). **f**, Difference between observed probability of early lick after a sequence of two fast TF pulses and the one predicted from their independent effect (top right corner in lower panel **c**, **d**), normalized by the expected probability from the effect of independent pulses and shown as a function of delay between them.

3. A recent paper from our lab (Orsolich et al., 2021, *Neuron*; <https://doi.org/10.1016/j.neuron.2021.03.031>), using the same behavioural paradigm as ours, introduced a Gaussian Process model which estimates lick probability at every point in time of a trial as a function of elapsed time and fitted stimulus filters (**Rebuttal Fig. 8a**), to predict when mice lick during our task. We applied this model to mice in our study (**Rebuttal Fig. 8**). This revealed that the timing of early licks (**Rebuttal Fig. 8b**), the psychometric curves (**Rebuttal Fig.**

8c), and reaction time curves (**Rebuttal Fig. 8d**) are well captured by this model. The model was trained on both early licks and hits/misses during the change period and showed that the stimulus filter best accounting for when mice lick was a filter with an integration time constant of ~ 230 ms (Stim filter #1; **Rebuttal Fig. 8a**). Importantly, inactivating this leaky integration filter caused the model performance to collapse, while inactivating stimulus filter #2 – the derivative filter – lead only to partial reduction in performance (**Rebuttal Fig. 8e,f**). This suggests that integration of the stimulus over several hundred milliseconds contributes to the animals' decision to lick.

Rebuttal figure 8: Gaussian process modelling of behaviour

a, Top 3 stimulus filters (in order of contribution to model) driving licks from the Gaussian Process model. **b**, Early lick density for early (blue) and late (orange) block, for 10% held out (cross-validated) dataset. **c**, Predicted psychometric curve (grey – error bars are constructed from 200 reiterations of the model fit) vs real psychometric (red) curve, on held out (cross validated) data set. **d**, Predicted mean reaction time curve (grey – error bars are constructed from 200 reiterations of the model fit) vs real reaction time curve (red), on held out (cross validated) data set. **e**, Predicted psychometric curve with all filters active (black), filter 1 (integration filter) made inactive (mauve), or filter 2 made inactive (green), and real data (red) on held out (cross validated) data set. **f**, Predicted mean reaction time curve with all filters active (black), filter 1 (integration filter) made inactive (mauve), or filter 2 made inactive (green), and real data (red) on held out (cross validated) data set.

Consistent with this, note the close correspondence between the model's stimulus filter #1 and the early lick triggered stimulus average (psychophysical kernel), which have similar timescales (**Rebuttal Fig 9c**).

Rebuttal figure 9: Estimation of integration time from GP model, and comparison to lick triggered stimulus average

a, Early lick triggered stimulus average with fitted decay time (τ). The τ is estimated from a fit of an exponential decay function starting at the peak mean TF value. **b**, First stimulus filter from GP model (see **Rebuttal Fig. 8**) with fitted decay time (τ). The τ is estimated from a fit of an exponential decay function starting at the peak mean filter value. **c**, Normalized lick triggered stimulus average and the first filter of GP model overlaid to highlight the similarity in integration times.

Regarding your point about burst detection. Burst detection is assumed to happen on very short timescales (i.e. ~50ms), with animals detecting a burst of sequential pulses of evidence to drive their decision (Brunton et al., 2013). Our results above demonstrate that animals use evidence over 200 ms, and the **new Fig 3 (see also Rebuttal Fig 7f)** specifically shows that two pulses of evidence spaced apart by even 200 ms still increase the probability of licking – demonstrating that evidence sampled need not be consecutive but can be spaced by more than 150ms and still be integrated to influence behavioural responses. This does not align with the idea of burst detection, but is better explained by gradual, leaky evidence integration.

Importantly, these behavioural timescales are in agreement with the integration timescales we have observed at a neuronal level across premotor brain areas (**Fig 3 f-g**). Unlike most studies investigating evidence integration having to rely on the speed at which activity ramps as a function of strength of evidence during decisions (where the ramp is assumed to mean the accumulation of evidence), we provide direct empirical evidence that individual brief pulses of evidence are combined across time at a single cell level (**Fig 3**), and that these give rise to ramping of activity during the change period (**Fig 3**).

The addition of these new results further supports the case that mice use leaky evidence integration rather than outlier detection as a principal strategy to make their decisions, and the study is now considerably stronger as a consequence. We thank you for bringing this to our attention!

Minor comments:

-I'm assuming that 1h shows model predictions for held-out test data; that should be clearly stated in the legend.

That is correct. These are predictions on a 10% held-out data set, averaged across a 10-fold cross-validation procedure. This is now stated in the legend.

-Is it possible to include the acronyms for brain regions in a figure legend instead of the supplement?
Yes, it's possible. But we believe the figure legend would inflate too much, and have therefore kept them in supplementary information.

-In Figure 4, can they put binomial confidence intervals on the bar plots?

We did not change the main Figure, as stats are performed on major area groupings and corresponding confidence intervals are shown in **Extended Data Fig 3 and 12**.

-4f: it's difficult to visually appreciate the difference in taus. Would it be more clear to overlay the curves?

Thank you. We have implemented your suggestion.

-For Figure 6, it doesn't appear that neural activity anywhere, including MOs, transitions into the movement subspace before the actual lick. How do the authors interpret the lack of dynamics preceding the action?

The activity starts to transition into the movement subspace with a relatively short, but positive latency around 100ms before the lick onset (see **Figure 6 b** and especially **d**, where MOs state is plotted at

10ms resolution), paralleling the low-dimensional dynamics of orofacial nuclei. We changed the label on **Figure 6 a** and **d** to “Lick onset” to make it clearer.

Typo: line 548 should be “during memory-guided movements”

This has been corrected.

Thank you.

Referee #3 (Remarks to the Author):

This paper studies how sensory evidence about the temporal frequency of a visual stimulus is accumulated over time and converted into a licking output. The main results of this paper are that longer sensory integration timescales arise after learning and that the accumulation of sensory evidence occurs in a distributed collection of premotor structures. This manuscript is excellent in its scope and methods. The behavioral task is interesting. The number of recordings and number of brain areas investigated is impressive. Equally impressive is how the manuscript explores in reasonable depth all the stages of processing from sensation to action. I like how the paper takes an exploratory, unbiased approach, whereas other work in this field has often started with targeted investigations. The experiments and analysis appear to be well done. The writing and presentation of complex data and analyses are clear and succinct. While there are many questions that come to mind regarding more analyses and experiments with these data and approaches, I do not feel it is justified to ask because the authors have already done a tremendous amount of work and presented clear results.

Thank you for your positive feedback.

I was excited by this paper for the reasons mentioned above. However, my enthusiasm was dampened because, despite the impressive methods and data, I did not find a major advance that goes beyond what is already known in the field. For this journal, I was expecting such an advance. Instead, I find most of the main results to be largely confirmatory of what has already been presented in the field in recent years.

Thank you. The central question is: What advances can be made by obtaining data from the entire brain, as opposed to sampling a subset of individual brain areas, towards understanding perceptual decision-making? We think they are:

- (1) We can show definitively that the evidence integration is distributed across almost the entire brain, as opposed to guessing that this is true. This therefore provides the ground truth data that should underlie future models of perceptual decision making.
- (2) We can show that – at the level of neural dynamics - the evidence integration subspace is aligned with the motor preparation subspace, but orthogonal to the movement subspace, across the entire brain. Therefore, the transformation of accumulated evidence into movement planning and execution takes place within and across subspaces of neural activity that are shared across multi-regional circuits, rather than proceeding successively across a subset of specialized brain areas. This reveals the universality of the transitions in neural dynamics surrounding movement, and allows integration of influential ideas from perceptual decision making and motor control fields into a unified framework.
- (3) We can discern which brain areas have dynamics that are sculpted by learning. The answer is essentially most of them apart from early visual areas. If this was known, we are not aware of it. For example, we might have guessed that integration would only take place in a limited subset of areas, or that the evidence integration would have emerged along a predefined hierarchy, or even that visual regions would also have learnt to integrate evidence. Instead, we show that the longest timescales are present in a distributed set of structures from cortex to basal ganglia to cerebellum to midbrain, indicating that learning turns evidence integration into a highly parallelised process.
- (4) We can relate different properties of neural dynamics across brain areas in the same task. This allows us, for example, to show that the learned dynamics do not reflect the intrinsic

dynamics across regions, contradicting several influential models of cortical computation (Murray et. al 2014, Siegle et al., 2021; Pinto et. al 2022, Manea et al., 2022; Cosyne 2022/2024 workshops).

We are proud of these findings as they demonstrate the benefits of brain-wide investigations of behaviour. They provide clear answers to major questions in the fields of perceptual decision making and motor control, and they offer a foundation for understanding learnt behaviours.

More specifically, the study provides several novel foundational insights about how the brain as a whole *learns* to use of sensory evidence to guide action. These are:

- As animals learn to respond to changes in a noisy stimulus stream, a sparse population of neurons distributed across the entire premotor brain acquires the ability to transform a relevant stimulus feature into an integrated sensory representation. In other words, unlike in sensory brain areas, neurons in the premotor structures develop the capacity to integrate the noisy stimulus stream, essentially performing a low-pass filtering to denoise the stimulus, making sensory change detection easier.

We believe this contrasts the interpretation arising from influential studies in the field that evidence accumulation is restricted to several 'cognitive/associative' areas in the neocortex (such as parietal and frontal cortex) and their immediate downstream targets (such as striatum and superior colliculus).

- Learning can decouple sensory processing timescales required for evidence integration from intrinsic activity timescales inherent to each area. This is something that challenges a major assumption in the field - that intrinsic timescales determine information processing timescales (Murray et. al 2014, Siegle et al., 2021; Pinto et. al 2022, Manea et al., 2022; Cosyne 2022/2024 workshops).
- The brain's solution that enables sensory input to drive behavioural responses is to *allow evidence integration to become motor preparation* within the same population of neurons distributed across the brain. In other words, learning aligns these computations within the same subspace of population activity, concurrently and globally. Importantly, neurons that integrate evidence drive preparatory activity in movement-null subspace, which allows evidence to accumulate without prematurely triggering the action. In this way, we link evidence accumulation onto motor dynamics on a brain-wide scale for the first time.

This provides an alternative to the search for a specific decision-making locus (such as superior colliculus or striatum) converting integrated sensory evidence into behavioural responses (e.g. see Stine et al., 2023 as a recent example). Instead, we highlight that this transformation is not restricted to a particular structure, but resides within neural dynamics of a sparse subpopulation of neurons distributed across most premotor structures in the brain.

We are very grateful for your comments, because they highlight a mismatch between the insights we were trying to convey, and the challenge we faced when providing a comprehensive description of decision processes on a brain wide scale within a single paper. Based on your comments, we believe our writing did not get this balance right, and in response, have re-written key parts of the manuscript, including the entire abstract, to emphasise more clearly the above conceptual advances (see sections highlighted in red).

The conclusion that decision making happens in a premotor circuit is the topic of many old and new lines of decision making research. Many studies from Shadlen and colleagues have emphasized how evidence accumulation and decision making happen in a motor-relevant context in LIP in primates. More recent, Svoboda and colleagues have a long line of papers describing how decision making occurs in ALM, which is a premotor structure for tongue movements. Another example, Shadlen in mice reported decision making in a premotor circuit recently (Wu et al 2020 “Context-dependent decision making in a premotor circuit”). I was surprised that some of this literature was not cited, in particular the Wu et al. paper because its main finding (and title) regards the focus on premotor circuits.

The fact that the relevant circuit is distributed is perhaps new relative to the lines of work I mentioned above. But, it is somewhat expected because it is consistent with many papers showing distributed processing in the mouse brain. The authors already cite a large number of these papers, and there are many more.

Thank you for mentioning the Wu et al 2020 paper, we added it to the references.

We agree that studies from Shadlen, Gold and colleagues are incredibly insightful and have pioneered the field of decision making, but they have focused only on individual (or a few) – and predominantly cortical – areas in isolation, including Wu et al. Moreover, the Svoboda papers beautifully examine the representation and circuit underpinnings of preparatory activity and motor execution in ALM cortex and connected circuits, but do not address the relationship between sensory evidence and its integration and how that relates to commitment to a decision (in the Svoboda task, it is not clear *when* the animal makes a decision – for that, reaction time tasks are needed).

As you indeed point out, our study reveals that the decision making process concurrently engages a widely distributed network of (premotor) areas across frontal cortex, basal ganglia, cerebellum, thalamus, and midbrain – from integration of visual evidence to the preparation and execution of an action. Importantly, we reveal the underlying single-cell and population level neural dynamics within a unified task and common analysis framework at the brain-wide level. This is a very distinct type of insight from saying that evidence integration happens in a particular premotor brain area, as it underscores the global and orchestrated nature of these processes. Perhaps more importantly, it emphasizes that the underlying mechanisms giving rise to decision-making processes (particularly integration of sensory evidence) must be explored at the level of concurrent activity in widely distributed multi-regional circuits.

The fact that the relevant circuit is distributed is perhaps new relative to the lines of work I mentioned above. But, it is somewhat expected because it is consistent with many papers showing distributed processing in the mouse brain. The authors already cite a large number of these papers, and there are many more.

Thank you for raising this important point. While recent mouse studies have indeed demonstrated brain-wide representations of various task-related variables, they do not provide direct evidence of where and how evidence integration occurs in the brain, nor how this integrated evidence is transformed into an appropriate action plan—two computations that are central to perceptual decision-making. Our study addresses these critical questions by leveraging brain-wide recordings in a task

designed to disambiguate sensory and motor components of neural activity, an aspect that was not the focus of the brain-wide recording studies you mentioned.

We reveal that, after learning, evidence integration and motor planning become tightly coupled, essentially merging into a single process. Specifically, we found that sensory evidence drives preparatory movement dynamics within sparse neural subpopulations distributed widely across most premotor structures in the brain. We believe that these insights represent a conceptual shift in the field of decision-making as it requires us to think differently about the mechanisms that underpin how and where perceptual decisions are made - pointing to collective neural dynamics across a multi-regional neural population that concurrently implements the sensory denoising and sensorimotor transformation.

Furthermore, our study sheds light on a crucial aspect that has been largely overlooked in previous work, in distinguishing which task-relevant representations across the brain are inherent versus which are shaped by learning. The papers looking at distributed representation of task variables across the brain (e.g. Steinmetz et al., 2019; IBL et al., 2023; Allen et al., 2019; Stringer et al., 2019; Chen et al., 2024) did not investigate the effects of learning on these representations at the brain wide scale. By comparing responses in naïve and trained mice we reveal which areas across the brain become recruited by learning to integrate evidence and, importantly, show that learning shapes the timescales over which sensory evidence is integrated across the brain, rather than it being determined by intrinsic timescales inherent to each area.

The difference in the timescales of evidence accumulation across brain structures has been reported multiple times previously (one example, Pinto et al. “Multiple timescales of sensory-evidence accumulation across dorsal cortex”). The main difference here is that the authors find that these timescales are learned instead of intrinsic, whereas previous papers suggest they might be intrinsic. This is an interesting finding, but a modulation of an established theme.

We politely disagree with the notion that our findings are merely a modulation of an established theme. We are in an era wherein there is a major drive to understand how neural dynamics in biological and artificial systems explain behaviour. A fundamental question in this pursuit is whether and how the intrinsic properties of neural circuits constrain their operational dynamics, such as the timescales over which they integrate information or maintain memories of past stimuli (e.g. Murray et. al 2014, Siegle et al., 2021; Pinto et. al 2022, Manea et al., 2022; Cosyne 2022/2024 workshops).

The prevailing view in the field is that intrinsic properties of brain areas determine their timescales of information processing, such as evidence integration (e.g. Murray et. al 2014; Pinto et. al 2022). Our study challenges this notion by demonstrating that the timescales of evidence integration are not predetermined by intrinsic timescales but are instead likely shaped by learning. Crucially, we verified the intrinsic nature of these timescales (i.e., their stability across learning) for each area by independently measuring them across two datasets (naive versus trained mice). This approach allowed us to disentangle the effects of learning from intrinsic properties. Our findings have significant implications for understanding the flexibility with which learning can sculpt neural circuit dynamics on a brain-wide scale.

Thus, we feel this insight goes beyond a mere modulation of an established theme, as it challenges a central assumption in the field, by providing evidence that learning shapes the timescales of

evidence integration independently of intrinsic timescales, a key factor governing neural dynamics during decision-making.

The separation of movement-null and movement-potent dimensions follows closely from the work that Shenoy and colleagues have performed over the years.

Our aim was precisely that – to apply a population analysis framework that was established specifically for the analysis of primary and premotor cortex activity during a motor control task – onto brain-wide recordings during a decision-making task requiring evidence integration. In doing so, we reveal that the findings of Shenoy, Churchland and colleagues is a general principle that extends across most premotor areas of the brain (but not in sensory areas or purely motor areas). Moreover, this approach enabled us - for the first time - to bridge the motor control and decision-making fields in a unified neural dynamics framework at a brain-wide scale.

Within this framework, we show that the neurons responsible for integrating sensory evidence are the same ones that predominantly drive preparatory activity and that evidence integration occurs in the movement-null subspace across all major brain subdivisions. This finding provides an explanation for why deliberation, i.e., the accumulation or evaluation of sensory evidence, can occur without inadvertently driving actions. Specifically, our results suggest that sensory and decision-making processes can unfold within a population subspace that is decoupled from the movement-generating dimensions of neural activity.

Thus, by applying the movement-null and movement-potent dimensions framework at a brain-wide scale, we reveal a general organizing principle for how premotor areas across the brain coordinate decision-making and motor control. Our findings demonstrate that the separation of these dimensions is not unique to motor cortex but rather a common feature of most premotor regions. This discovery underscores the importance of studying decision-making and motor control within a unified framework and highlights the power of brain-wide recordings in uncovering general principles of neural computation.

I very much wanted to find a conceptual advance here because I liked a lot about this paper, but I struggled to find it. I wonder if the authors had similar challenges because the final, summative sentence of the abstract is “We propose that learning establishes the link between the external input and relevant premotor circuits throughout the brain, which integrate sensory evidence to drive activity dynamics that lead to the initiation of appropriate actions.” This sentence, in my opinion, does not offer much insight beyond what has to be the case for any decision making task. All tasks of this type will necessitate learning to associate the sensory with the motor and brain dynamics will always precede actions.

We appreciate your feedback and acknowledge that the final sentence of the abstract did not effectively capture the novel insights of our study. Our intention was to highlight the unique combination of findings that demonstrate how learning establishes a link between sensory evidence and motor preparation specifically in regions capable of integrating evidence, which are also the premotor regions. While it may seem obvious that such a link must exist somewhere in the brain, our study reveals that this linkage occurs in a remarkably widespread manner, spanning multiple premotor regions across all major brain subdivisions.

In light of your feedback, we have revised the abstract and made changes throughout the text to emphasize the key takeaways more clearly. These modifications better convey the conceptual advances of our work, which include:

- Revealing the widespread coupling of sensory evidence and motor preparation across multiple premotor regions in all major brain subdivisions, challenging the notion of a more localized decision-making process.
- Demonstrating that learning plays a crucial role in establishing this distributed linkage between sensory integration and motor preparation, shaping the neural dynamics that drive action initiation.
- Uniting the fields of decision-making and motor control at a brain-wide scale, providing a comprehensive framework for understanding how sensory evidence is transformed into actions through coordinated neural dynamics across premotor regions.

We hope this convinces you of the importance and conceptually novelty of our study

If the other reviewers find interesting and novel major advances, then I am happy to be convinced. Perhaps one suggestion that might help would be to write out in the text what the alternative hypotheses or outcomes are, which might reveal what new has been learned.

Please see above, as well as the opening comments of Reviewer #1.

One small thing: for some of the areas, the unit count is low, around tens of neurons. As we know, representations in many of these brain structures is sparse, so it might be challenging to make strong claims with limited sampling. Perhaps the authors could clarify which areas this might affect or present statistical arguments for why they have enough power with the current sample size in these low-count areas.

We share your concern and for this reason we chose a minimum threshold of 40 stable single units for a brain region to be included in any analysis. For analyses of TF-responsive units the threshold of their count was 10. To ascertain the robustness of results, we show a 95% confidence intervals of bootstrapped distribution sampled across TF-responsive units (for example **Fig 3f,g**). Moreover, for these reasons, for the majority of analyses we included the comparison across major groups of brain regions, where individual TF-responsive units of all brain regions within the group (e.g. early visual areas or frontal cortex) were pulled together (see **Fig 2h,i**; **Fig 3n**; **Fig 5d**; **Fig 6l,o**).

In summary, I am impressed by the scope and quality of this work. It is a study that puts together many previous findings and approaches into a single study. There is great value in this, and this leads to my considerable excitement about this paper. However, I was disappointed that I was not able to find a significant conceptual advance beyond what is already known in this field, which is something that I expect for this journal.

Again, we are very grateful for your comments. In response, we have re-written the abstract and made changes to the rest of the text to emphasize the key conceptual insights and believe the paper is considerably stronger as a result. Thank you.

Reviewer Reports on the First Revision:

Referees' comments:

Referee #1 (Remarks to the Author):

After now having the chance to read the revised manuscript, the thoughtful comments from the other reviewers and the author rebuttal, my positive view of this work remains the same as I described in the first round of review. Without repeating myself unnecessarily, I would like to highlight a part of my perspective that relates to the main concerns of the other reviewers. I agree that no single finding reported in this manuscript is utterly surprising, but the vast scope of the work resolves many very important open questions in a way that would not be possible without the monumental effort that was needed to perform these experiments. On the whole, I assess this work to be even more important than many other (highly important) papers on neural mechanisms of decision making that have published in the highest profile journals, including work that I have been involved in performing. The authors do an excellent job in describing the conceptual advances and their impact in the revised abstract and in multiple places in the rebuttal document.

In addition to those big-picture considerations, the authors also addressed all of my specific comments with care and thoroughness in a manner that addresses my concerns. They have made a large number of revisions and additions to the manuscript that will be helpful to readers who might have shared my concerns. My one regret is that I am not sure if there is a way for the rebuttal to be accessible to general readers, which would be a shame considering the depth of thought the authors put into it.

In short, I give a strong recommendation in favor of publication.

Referee #2 (Remarks to the Author):

The reviewers addressed my major technical comment, by including new analyses that support the claim that mice integrate sensory evidence over time. I think these additional analyses have improved the paper. However, my view of it remains essentially the same. I think it is technically well-performed, the approaches are state of the art, and the conceptual advances are incremental and mostly replicate previous findings. The other two reviewers also seemed to share this core perspective (albeit with differing levels of enthusiasm about the work overall). I think this is a good paper! I just don't think I have learned anything surprising or profound from it beyond previous studies.

Referee #3 (Remarks to the Author):

I appreciate the thoughtful responses from the authors. My overall assessment remains unchanged from the first round of review because the main results and conclusions are largely unchanged from the first submission. I find the work to be technically impressive. The abstract is improved and better highlights the findings. However, I still do not see a major conceptual advance from the work. I am largely in agreement with the comments from Reviewer 2. I do not have any concerns about the technical rigor of the work.